# Symmetric Basis Convolutions for Learning Lagrangian Fluid Mechanics

**Rene Winchenbach & Nils Thuerey**
Physics-based Simulation Group
Technical University Munich
{rene.winchenbach,nils.thuerey}@tum.de

## Abstract

Learning physical simulations has been an essential and central aspect of many recent research efforts in machine learning, particularly for Navier-Stokes-based fluid mechanics. Classic numerical solvers have traditionally been computationally expensive and challenging to use in inverse problems, whereas Neural solvers aim to address both concerns through machine learning. We propose a general formulation for continuous convolutions using separable basis functions as a superset of existing methods and evaluate a large set of basis functions in the context of (a) a compressible 1D SPH simulation, (b) a weakly compressible 2D SPH simulation, and (c) an incompressible 2D SPH Simulation. We demonstrate that even and odd symmetries included in the basis functions are key aspects of stability and accuracy. Our broad evaluation shows that Fourier-based continuous convolutions outperform all other architectures regarding accuracy and generalization. Finally, using these Fourier-based networks, we show that prior inductive biases, such as window functions, are no longer necessary. An implementation of our approach, as well as complete datasets and solver implementations, is available at https://github.com/tum-pbs/SFBC.

## 1 Introduction

Physical simulations play an essential role in many fields of science, e.g., Computational Fluid Dynamics (CFD) (Zhang et al., 2021), with a long history of classical numerical research aimed at improving the performance and accuracy of solvers (Vacondio et al., 2021). Nonetheless, numerical solvers typically rely on many insights and intuitions, e.g., which solver and time-stepping scheme to use and which conservation laws to enforce explicitly and implicitly (Pope, 2000). While neural solvers traditionally focus on learning from data (Ummenhofer et al., 2019; Morton et al., 2018) there is often a clear benefit to the integration of inductive biases to enforce difficult-to-learn aspects, e.g., rotational invariance (Satorras et al., 2021; Lino et al., 2022; Thomas et al., 2018), momentum conservation (Prantl et al., 2022) and symmetry (Wang et al., 2021).

Within CFD, the underlying Partial Differential Equations (PDEs) are either solved on grids with Eulerian approaches or using particles with Lagrangian methods. For Eulerian approaches on structured grids, Convolutional Neural Networks (CNNs) have found broad adoption (Tompson et al., 2017; Bar-Sinai et al., 2019; Thuerey et al., 2020); however, applying CNNs to unstructured grids is not directly possible. Graph Neural Networks (GNNs) (Pfaff et al., 2021; Zhou et al., 2020) approaches have been popular for unstructured grids, e.g., *GNS* (Sanchez-Gonzalez et al., 2020), where cell centers are treated as vertices and cell faces as graph edges. These GNNs have found broad application for various PDEs, e.g., Burger's equation, for unstructured grids and particle-based systems (Brandstetter et al., 2022b; Allen et al., 2022). *Continuous Convolutions* (CConvs) (Wang et al., 2018) are a subset of GNNs, where interactions are based solely on the relative position of cells or particles. These continuous approaches are especially suited for Lagrangian simulations, as particle positions change continuously over time. CConvs often use convolutional filters built using interpolation approaches instead of using Multilayer-Perceptrons (MLPs), as these filters have inherently useful properties, such as smoothness (Fey et al., 2018) and antisymmetry (Prantl et al., 2022). Consequently, CConv approaches learn more accurate solutions than general GNNs for an equivalent number of parameters in physical problems (Prantl et al., 2022).

In this context, we propose (a) a generalized formulation of CConvs using separable basis functions, (b) a Fourier-based architecture with even and odd symmetry for improved inference accuracy and (c) a novel dataset consisting of four test cases designed to be challenging while exhibiting quantifiable physical behavior. We perform an exhaustive set of ablation studies to investigate which classes of basis functions and inductive biases are beneficial for accurately learning Lagrangian flow physics.

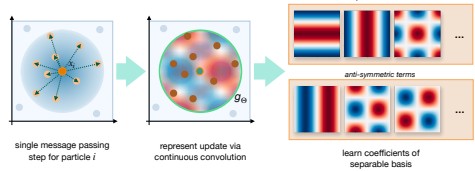

Figure 1: Visual overview of SFBC.

## 2 BACKGROUND AND RELATED WORK

We will first outline our definition of neural solvers in Section 2.1, followed by an overview of classical solvers in Section 2.2 and prior work regarding machine learning for simulations in Section 2.3.

### 2.1 NEURAL NETWORK TIME INTEGRATION

At a given time $t$ with particles at position $\mathbf{x}^t \in \mathbb{R}^d$, with $d$ spatial dimensions, and corresponding velocities $\mathbf{v}^t \in \mathbb{R}^d$, the goal of a physics simulation is to compute a new set of positions and velocities at a new time-point $t^\star$, i.e., $\mathbf{x}^{t^\star}$ and $\mathbf{v}^{t^\star}$, subject to a PDE $\delta^t \mathbf{u} = F(t, \mathbf{x}, \mathbf{u}, \delta^x \mathbf{u}, \delta^{xx} \mathbf{u}, \dots)$, where $\mathbf{u}$ is being solved for. Based on the PDE, a solver now computes an update for the particle positions, $\Delta \mathbf{x}^t$, and particle velocities, $\Delta \mathbf{v}^t$, for a given discrete time-step $\Delta t$, which yields:

$$\mathbf{x}^{t+\Delta t} = \mathbf{x}^t + \Delta \mathbf{x}^t; \quad \mathbf{v}^{t+\Delta t} = \mathbf{v}^t + \Delta \mathbf{v}^t. \tag{1}$$

Existing neural solvers (Ummenhofer et al., 2019; Prantl et al., 2022) often simplify this problem by inferring the velocity update from the position update, i.e., $\Delta \mathbf{v}^t = \Delta \mathbf{x}^t / \Delta t$. Including an initial velocity $\mathbf{v}^t$ and external constant forces $\mathbf{a}^t$, such as gravity, yields our general learning task as

$$\mathbf{x}^{t+\Delta t} = \mathbf{x}^{t,\prime} + G(\mathbf{x}^t, \mathbf{f}^t, \Theta); \quad \mathbf{x}^{t,\prime} = \mathbf{x}^t + \Delta t \mathbf{v}^t + \Delta t^2 \mathbf{a}^t, \tag{2}$$

where $G$ is a Neural Network with parameters $\Theta$ and a feature vector $\mathbf{f}$. Note that the initial velocity $\mathbf{v}^t$, due to the inference of velocity from position updates, is not the actual instantaneous velocity of the particles but is defined via $\mathbf{v}^t = \frac{\mathbf{x}^t - \mathbf{x}^{t-\Delta t}}{\Delta t}$. The minimization task is then given as

$$\min_{\Theta} L(\mathbf{x}^{t,\prime} + G(\mathbf{x}^t, \mathbf{f}^t, \Theta), \mathbf{y}^{t+\Delta t}), \tag{3}$$

where $\mathbf{y}^{t+\Delta t}$ are known, ground-truth positions from a given dataset and $L$ being a loss function, commonly chosen as $L^2$. In the following, we refer to solvers of this form as Neural Network Time Integrators (NNTIs). Note that the time-step $\Delta t$ and associated time-stepping scheme do not need to be the same for ground-truth simulation and NNTI. For example, it is possible to predict multiple ground truth-simulation steps simultaneously using multi-stepping or temporal-bundling (Ummenhofer et al., 2019; Brandstetter et al., 2022b).

### 2.2 CLASSICAL SOLVERS

We utilize the Smoothed Particle Hydrodynamics (SPH) method as the basis for our test cases and to inform some of our inductive biases. SPH is a Lagrangian simulation technique initially introduced in an astrophysics context (Monaghan, 1994) but has found broad application in various fields, e.g., in CFD (Monaghan, 2005) and Computer Graphics (Ihmsen et al., 2013; Macklin & Müller, 2013). At the heart of SPH are interpolation operations $\langle A \rangle$ of quantities $A$ carried by particles, with positions $\mathbf{x}$, mass $m$ and density $\rho$, using a Gaussian-like kernel function $W$ as

$$\langle A_i \rangle = \sum_{j \in \mathcal{N}_i} A_j \frac{m_j}{\rho_j} W(\mathbf{x}_j - \mathbf{x}_i, h), \tag{4}$$

where $h$ is the support radius and $\mathcal{N}_i$ being the neighbors of $i$, including itself, which are all particles closer than $h$, see Koschier et al. (2019) for a broader overview. This can be seen as a message-passing step using the edge lengths and features of the adjacent vertices with a summation message-gathering operation. Within SPH many solvers exist for a broad variety of problems and choosing

the correct one for a respective problem can be challenging as they are vastly different. As our test cases involve compressible, weakly-compressible and incompressible SPH solvers to generate the data, understanding their background and requirements is valuable.

For compressible simulations, SPH utilizes either an explicit pressure formulation using a compressible Equation of State, as do we, or using Riemannian solvers (Puri & Ramachandran, 2014), which would be an important direction for future research due to their complexity. For weakly compressible simulations the most commonly utilized technique is the $\delta$-SPH method (Marrone et al., 2011), using explicit pressure forces combined with diffusion models for velocity and density terms using very small timesteps, with the $\delta^+$-SPH (Sun et al., 2018) variant that we use for our data generation further expanding this approach. Finally, for incompressible SPH (Ihmsen et al., 2013; Bender & Koschier, 2015) the simulation revolves around solving a Pressure Poisson Equation using an implicit solver using many iterations per timestep. Overall, the SPH solvers we used vary from straightforward explicit integration over numerically challenging explicit integration with very small timesteps to requiring large linear systems to be solved per timestep.

## 2.3 NEURAL NETWORKS

Many methods have been proposed to solve PDEs using machine learning techniques Battaglia et al. (2016); Morton et al. (2018); Um et al. (2020), where Physics-Informed Neural Networks (PINNs) are among the most prominent approaches to solving such PDEs (Cai et al., 2021). PINNS are coordinate networks that learn the solution of a PDE without first requiring a discretization using continuous residuals. While powerful, these methods have many drawbacks, including difficulties in training and cannot be applied to discrete simulation data. Another prominent machine learning technique is point-cloud classification and segmentation, e.g., *PointNet* (Qi et al., 2017). However, these approaches are not well-suited to Lagrangian simulations as particles are much more interconnected than point clouds. Neural Operators have recently also found research interest in solving PDEs (Li et al., 2021b; Guibas et al., 2021; Li et al., 2021a). However, applying these approaches to irregular grids or particle-based simulation is not easily possible. On the other hand, Graph Neural Networks (GNNs) (Sanchez-Gonzalez et al., 2020) naturally map to simulations as most simulations can be interpreted as a form of message passing on a graph.

**Graph Neural Networks** can be directly applied to SPH simulations, where each particle is a vertex, and the particle neighborhoods describe the graph connectivity. GNNs use the graph to generate messages using just the vertex information (Qi et al., 2017), edge information (Wang et al., 2018), edge and vertex (Sanchez-Gonzalez et al., 2020), or edge, vertex, and additional feed-through features collected on vertices using pooling operations (Brandstetter et al., 2022a). These collected messages are then either used directly, as new features on the vertices, or combined with existing vertex features. Further operations, such as input encoding and output decoding, have also been proposed (Sanchez-Gonzalez et al., 2020). The message processing performed using MLPs with relatively shallow but broad hidden architectures, e.g., 2 hidden layers with 128 neurons.

**Continuous Convolutions** are a subset of GNNs and utilize only coordinate distances as inputs to the filter functions, similar to SPH kernel functions (Ummenhofer et al., 2019). These filter functions are then combined with the features of adjacent vertices to form the messages. While this imposes an inductive bias, it can make learning the problem more manageable. The coordinate distances can then be processed using an MLP (Wang et al., 2018) or other function interpolation techniques, e.g., linear interpolation (Ummenhofer et al., 2019) or spline-based interpolation (Fey et al., 2018). Several extensions of these approaches have been proposed for physical simulations, e.g., using antisymmetry (Prantl et al., 2022) to conserve particle momentum.

**Fourier and Chebyshev Methodes:** Fourier Neural Operators (FNOs) (Li et al., 2021b) learn physical simulations by transforming a given spatial discretization into a spectral representation using an FFT. By applying the learning task in the spectral domain, these operators can learn spatially invariant behavior but are limited to regularly sampled data due to their reliance on FFTs. Fourier encodings have also been applied in image classification and segmentation to make networks less spatially dependent (Li et al., 2021a). Chebyshev basis polynomials have also been used in Graph classification tasks using CConvs as higher-order interpolants (Defferrard et al., 2016; He et al., 2022), as well as other polynomial bases, e.g., Bernstein polynomials (He et al., 2021).

## 3 METHOD

Our work builds on a generalized formulation of continuous convolutions that acts as a superset of existing methods, such as *LinCConv* (Ummenhofer et al., 2019) and SplineConv (Fey et al., 2018). Based on this model, we describe parameterizations of convolutional filter functions using separable basis functions and explain how prior work fits into this concept. Furthermore, we will describe how symmetries can be built into this model and construct a Fourier-based convolutional architecture that uses both even and odd symmetry. Finally, we will discuss window functions, an important inductive bias in many existing CConv approaches.

**Formulation**: In general, convolution is a mathematical operation that combines an input function, $f : \mathbb{R} \to \mathbb{R}$, and a filter function, $g : \mathbb{R} \to \mathbb{R}$, through an operation defined as (Wang et al., 2018)

$$(f \circ g)(\mathbf{x}) = \int_{-\infty}^{\infty} f(x - \tau) \cdot g(\tau) d\tau. \tag{5}$$

We then limit $g$ to be compact, i.e., $g(\tau) = 0 : \forall |\tau| \geq h$, with $h$ being a cutoff distance, also referred to as support radius within SPH contexts. We can then sample $f$ on the positions of vertices, $\mathbf{x}$, and base $\tau$ on the coordinate distances between connected vertices to discretize the convolution as

$$(f \circ g)(\mathbf{x}_i) = \sum_{j \in \mathcal{N}_i} f(\mathbf{x}_j) \cdot g\left(\frac{\mathbf{x}_i - \mathbf{x}_j}{h}\right), \tag{6}$$

where $i$ and $j$ are the indices of two vertices, see Appendix A.1. This formulation is then expanded by the inclusion of a normalization term $\phi(x)$, typically only used in classification tasks, a window function $w$ similar in shape to an SPH kernel function, see Appendix A.2, and a coordinate mapping function $\Lambda : \mathbb{R}^d \to \mathbb{R}^d$, see Appendix A.3. Denoting the trainable weights of $g$ by $\Theta$ this yields

$$(f \circ g)(\mathbf{x}_i) = \frac{1}{\phi(\mathbf{x}_i)} \sum_{j \in \mathcal{N}_i} f(\mathbf{x}_j) \cdot g_\Theta\left(\Lambda\left(\frac{\mathbf{x}_i - \mathbf{x}_j}{h}\right)\right) \cdot w\left(\frac{|\mathbf{x}_i - \mathbf{x}_j|}{h}\right). \tag{7}$$

The machine learning task then is to find a set of weights $\Theta$ such that $(f \circ g_\Theta)(\mathbf{x}_i) = y(\mathbf{x}_i)$, where $y$ represents the supervised ground truth result. We now propose to parametrize $g_\Theta$ using a set of $n$ one-dimensional basis functions $b_i(x)$, with $i \in [0, n)$, such that for a one-dimensional convolution we get $g_\Theta(q) = \langle \mathbf{b}(q), \Theta \rangle$, where $\langle \cdot, \cdot \rangle$ denotes the inner product. A direct choice for $b_i$ would be a piece-wise constant function that results in a Nearest Neighbor interpolation or a piece-wise linear function that results in the *LinCConv* approach; see Appendix A.4. For a two-dimensional convolution, we construct a matrix of basis terms as the outer product of a set of separable basis terms in $x$ and $y$, i.e., $b_i^x(\mathbf{q}_x)$, with $i \in [0, u)$, and $b_j^y(\mathbf{q}_y)$, with $j \in [0, v)$, such that

$$g_\Theta(\mathbf{q}) = \langle [\mathbf{b}^x(\mathbf{q}_x) \otimes \mathbf{b}^y(\mathbf{q}_y)], \Theta \rangle. \tag{8}$$

While the imposed restriction of the basis terms being separable limits the potential choices for basis terms, it is also in line with prior work, e.g., by Fey et al. (2018). A key benefit of such a separable formulation is that gradients can be computed straightforwardly through a transpose of the weight matrix instead of requiring more expensive steps or even involving non-linear optimization, as is required for traditional RBF Networks as proposed by Broomhead & Lowe (1988). It is important to keep in mind that, as continuous convolutions are matrix multiplications, updating $\Theta$ at learning time is still a linear operation even if the basis terms are non-linear; see Appendix A.1.

Several basis terms $\mathbf{b}$ have already been considered in prior work, and we will summarize some of them here in a two-dimensional context. A simple choice is using a bi-linear interpolation, e.g., as in the *LinCConv* approach (Ummenhofer et al., 2019). Here, each entry of the weight matrix $\Theta$ only has a very local influence, and the shape of the learned convolution operator $g_\Theta$ is a combination of piece-wise linear functions, and thus also piece-wise linear. While such a formulation is straight-forward to implement, it gives no guarantees regarding symmetry, the learned function cannot be smooth, and all learned convolutional filters in a larger network will have discontinuities located at the same relative positions. Using cubic B-Splines (Fey et al., 2018) results in smoother learned filters without discontinuities but still does not guarantee any symmetries.

**Incorporating Symmetry**: The *DMCF* approach (Prantl et al., 2022) used an antisymmetric basis to enforce conservation of momentum, which was implemented through explicit mirroring of weights.

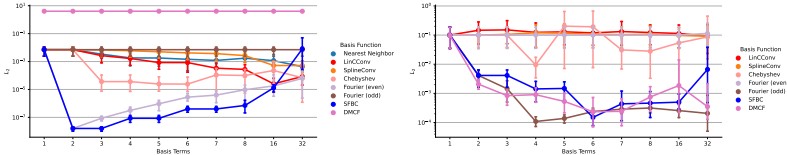

Figure 2: Quantitative evaluation of the one-dimensional kernel function (left) and kernel gradient (right) toy problem. Evaluations are based on the $L_2$ error related to the number of base terms.

This had the side effect of reducing the effective number of parameters by a factor of two, and symmetry was neglected during backpropagation. Our framework allows us to directly implement the antisymmetry constraint, which enables us to instead modify the basis formulation from Eq. 8 such that it can be applied to any basis function and results in an antisymmetric basis

$$g_\Theta^{\text{asymm}}(\mathbf{q}) = \langle [\text{sgn}(\mathbf{q}_x)\mathbf{b}^x(2|\mathbf{q}_x| - 1) \otimes \mathbf{b}^y(\text{sgn}(\mathbf{q}_x) \cdot \mathbf{q}_y)], \Theta \rangle, \tag{9}$$

which can be modified to lead to a symmetric formulation by excluding the sgn term:

$$g_\Theta^{\text{symm}}(\mathbf{q}) = \langle [\mathbf{b}^x(2|\mathbf{q}_x| - 1) \otimes \mathbf{b}^y(\text{sgn}(\mathbf{q}_x) \cdot \mathbf{q}_y)], \Theta \rangle. \tag{10}$$

While ensuring that all basis functions are antisymmetric is useful for conserving momentum, not all target functions are necessarily antisymmetric, e.g., a density interpolation in SPH is rotationally invariant and can not be learned with such a basis. Furthermore, the resulting filter function is not smooth, and the resolution along different coordinate axes is not identical. To resolve these shortcomings, we propose a set of smooth basis functions with either even or odd symmetry, where all basis terms influence the outcome for any value $q$. Our primary choice for such a basis is a Fourier series; see Appendix A.4 for visualization and definition of the two-dimensional Fourier series. Note that this fundamentally differs from applying a Fourier transform, e.g., as done in the *FNO* approach (Li et al., 2021b). Architectures like *FNO* transform an input signal into frequency space through an explicit Fourier transform and learn in the frequency domain. We instead use the Fourier basis to construct a convolutional kernel. The input signal keeps its spatial representation, but the learning task becomes finding the best possible coefficients for a Fourier series. Finally, we also include Chebyshev polynomials, which are popular for approximating non-periodic compact functions and have inherent symmetries. Additional basis constructions are discussed in Appendix A.4.

**Window Functions** are an important inductive bias in many prior works regarding CConvs, especially within learning physical simulations. The general motivation behind a window function is that interactions should be (a) compact, (b) smooth, and (c) behave like SPH interpolations. This inductive bias is primarily informed by SPH methodology and, consequently, prior work generally used SPH kernel functions as window functions such as the Müller (Ummenhofer et al., 2019) or Spiky kernel (Prantl et al., 2022). We evaluate several other functions; details are given in Appendix A.2.

## 4 RESULTS

We now propose the *SFBC* (Symmetric Fourier Basis Convolution) approach using a Fourier basis with no window function and an identity coordinate mapping. We first compare *SFBC* against other CConv approaches in a toy problem in one and three dimensions to compare the capabilities of the interpolation bases; see Section 4.1. We then compare *SFBC* against various baselines in a compressible one-dimensional problem; see Section 4.2. Next, we perform an in-depth evaluation of a two-dimensional closed domain simulation focused on inference stability; see Section 4.3. Finally, we evaluate a fluid blob collision scenario to investigate how different basis terms perform with partially occupied support domains; see Section 4.4. For more details on the training and setup, see Appendix C. We also performed a runtime analysis of the most relevant hyperparameters and found that *SFBC* on average only incurs an increase of $0.5\%$, $8.5\%$ and $12\%$ over *LinCConv* in one, two, and three dimensions, respectively, see Appendix C.6 and Figure 38 for details.

### 4.1 TOY PROBLEMS

To evaluate the capabilities of the different basis functions to learn symmetries, we consider two tasks: (a) a symmetric task to learn an SPH kernel interpolation, and (b) an antisymmetric task

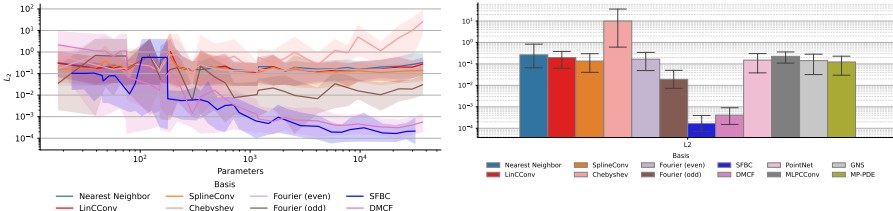

Figure 3: A quantitative evaluation of the relationship between parameter count and test error for basis convolutions(left), and a quantitative evaluation of different networks for a fixed layout with four message-passing steps and 32 features per layer (right). Error bars showing lower to upper $5\%$.

to learn the SPH gradient, see Appendix B.2. This setup is motivated by the concept that if a machine learning method cannot learn the basic components of an SPH simulation, then learning the overarching SPH simulation is made more difficult. As we want to evaluate the abilities of the different basis functions to act as interpolation functions, we utilize a network with a single message-passing step without activation functions, i.e., the learning task is a linear optimization problem.

**Kernel Function**: Based on the results shown in Figure 2 and Appendix C.1.1, we observe a clear difference between different basis terms. The *SFBC* approach performs best but shows a decrease in performance with increasing numbers of base terms as this learning task does not require any higher-order harmonics, and learning a contribution of zero is potentially challenging. The second best-performing method is the Chebyshev basis, although this method performs several orders of magnitude worse than the Fourier basis. The baseline methods show much worse overall performance, only getting closer *SFBC* for a high number of terms. The *DMCF* approach and a Fourier series consisting of odd terms perform the worst as these, by construction, can only learn antisymmetric behavior. The inclusion of symmetry significantly improves performance and reduces the lowest achieved $L_2$ error by a factor of $5217$ when compared to *LinCConv*.

**Kernel Gradient**: For this case, shown in Figure 2 and Appendix C.1.2, we see a notably different result. Only a few of the methods demonstrate any reasonable performance, and all of these methods have inherently antisymmetric terms. Overall, the Fourier series with only odd symmetry terms performed best, followed by a complete Fourier series and *DMCF*, where the latter was impeded primarily by a lack of smoothness for low basis term counts. These results demonstrate that antisymmetric learning tasks are challenging for traditional basis functions, and including symmetry and smoothness as an inductive bias significantly improves the overall learning behavior by a factor of $7.6$ relative to *DMCF* and $853$ relative to *LinCConv* when compared to the odd Fourier series.

**Three Dimensions**: We expanded this evaluation to three dimensions, see Appendix B.5, where we found similar behavior regarding symmetric tasks with *SFBC* outperforming *LinCConv* by a factor of two on average. As the tensor products of antisymmetric bases are not necessarily antisymmetric, we observed much worse performance for the antisymmetric odd Fourier basis. They perform on average three times worse than *SFBC* (see Appendix C.5.1). We also utilized this setup to evaluate how non-ideal learning setups perform and found that *SFBC* was significantly more resilient against superfluous message-passing steps and input features, see Appendix C.5.2. Overall these findings suggest that including symmetry and smoothness into the basis significantly improves learning performance and improves the networks ability to learn in a broad range of conditions.

## 4.2 TEST CASE I: COMPRESSIBLE 1D SPH

Based on the results from the toy problems, we now evaluate how these behaviors translate to a more holistic learning task where the overall physics update should be learned. The learning setup here is a one-dimensional compressible SPH simulation, where the updated velocities per particle should be learned based on the velocity of the current particles. Appendix B.1 provides details of the data generation and underlying simulation model. For the width of the base functions, we chose $n = 6$, based on the results from the toy problems and a hidden architecture for the MLP-based approaches of 2 deep and $128$ wide, in line with Sanchez-Gonzalez et al. (2020).

**Basis Function Methods**: We first consider the influence of the network size on the achievable test error by evaluating a large number of hyperparameters, see Fig. 3 and Appendix C.2. Our

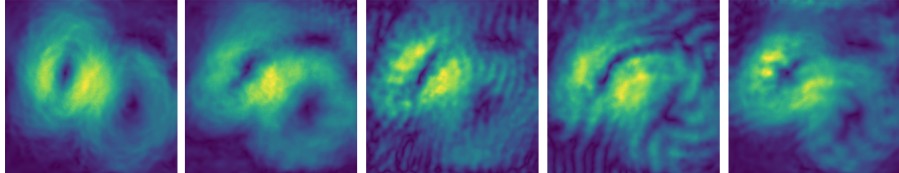

Figure 4: A qualitative comparison after $64$ inference steps with (f.l.t.r.) of the ground truth data, our *SFBC* approach, *LinCConv*, a Fourier basis with window, and Chebyshev with window. The particle data is mapped to a regular grid with color mapping indicating current velocity.

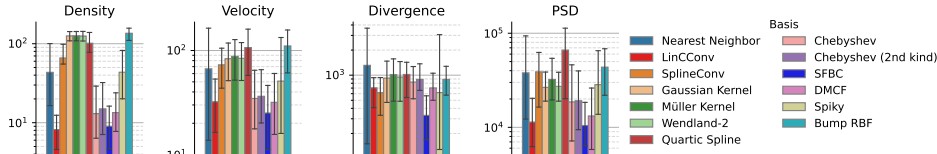

Figure 5: The quantitative results are based on the integrated error over an inference period of $64$ steps regarding four error metrics not included in the training loss.

results demonstrate that there is a clear and noticeable difference between basis functions built upon antisymmetric terms, i.e., *Fourier*, *Fourier (odd)* and *DMCF*, that makes them perform notably better than all other basis functions. Considering the size of the network, we made several observations; see Appendix C.2. On the one hand, we found that increasing the number of message-passing steps, e.g., from 3 to 6, only changed the network performance by $\pm 10\%$, while increasing the features per layer from 4 to 32, with 3 message-passing steps, improved performance by an order of magnitude, see Fig. 21. On the other hand, we also saw a clear benefit of the *SFBC* approach as it outperforms *DMCF* at virtually all sizes, with a more significant difference for smaller layouts.

**MLP-based Methods**: Next we compare to a set of MLP-based GNNs using *PointNet* (Qi et al., 2017), *GNS* (Sanchez-Gonzalez et al., 2020), *MLSConv* (Wang et al., 2018) and *MP-PDE* (Brand-stetter et al., 2022b) architectures with 5 message passing steps and 32 features per vertex and message. Our results show that *GNS* and *MP-PDE* perform as well as most convolutional basis functions but not as well as methods with built-in symmetries, see Fig. 3. However, they require significantly more parameters to reach a comparable accuracy, i.e., MLSConv, GNS, and MP-PDE require 229K, 393K, and 395K parameters, respectively, compared to 30K for the CConv methods. This highlights the importance of basis convolutions as an inductive bias that allows the CConv-based networks to achieve the same performance with fewer resources. Within the collection of MLP baselines, *MLSConv* performs slightly worse but is still comparable to other approaches. This indicates that the inductive bias of including a convolution by itself is not sufficient but that the advantages come from constructing them using basis functions with inherently useful properties.

Overall, the results indicate that the basis function convolutions perform similarly to MLP-based GNNs while requiring significantly fewer parameters. Furthermore, including a bias of symmetry significantly improves the capabilities of a network, e.g., only methods with symmetry were able to accurately learn the gradient function. Overall, our *SFBC* approach shows its robustness by performing better than other methods across a wide range of architectural changes, see Appendix C.2.

## 4.3 TEST CASE II: WEAKLY-COMPRESSIBLE 2D SPH

For the second test case, we focus on the long-term stability of various NNTIs for a closed domain weakly-compressible simulation; see Appendix B.3 for details on the data generation. We utilize a network architecture using $n = 4$ with four rounds of message passing, in line with Ummenhofer et al. (2019), and train the networks with a maximum rollout of $10$ and an evaluation on the test dataset with an inference length of $64$. To quantify the performance, we map the particle density and velocity to a regularly sampled grid spanning the closed simulation domain and then assess the $L_2$ difference of density, velocity, and divergence on the grid. We also compute a Power Spectral Density (PSD) difference based on the velocity field. The divergence is the central metric for this case as it is a derived metric that is challenging to uphold and closely correlated to a lack of smoothness,

e.g., see Fig. 4. Note that we naturally require a low error in all metrics for a truly accurate result. We now evaluate a broad set of choices of basis functions and hyperparameters.

**Baseline Comparison**: The proposed Fourier-based network performs notably better than other baseline methods, i.e., *LinCConv*, *DMCF* and *SplineConv*, and outperforms all other basis functions in almost all metrics, e.g., it outperforms *LinCConv* by a factor of 2.6 regarding divergence error. The only exception is the nearest neighbor basis regarding density and PSD error but, crucially, not regarding divergence error; see Fig. 5 and Appendix C.3.1. Overall, Chebyshev basis functions also performed well, but only when using a window function, and *SplineConv* performs worse than *LinCConv*, in line with prior observations (Ummenhofer et al., 2019). Finally, for B-spline and other SPH kernels, the higher the order, the better the result. In addition to the quantitative evaluations, we also considered the qualitative results, where the Fourier basis with no window shows the, by far, best prediction, see Fig. 4. In contrast, other methods show a superimposed noise on the velocity field, which is a combination of both the choice of basis and window function, adding different noise patterns. This superimposition of noise can easily dominate the overall prediction and make the prediction unstable and unusable. We also performed a study for a more significant number of seeds (32 instead of 4) for the key methods, see Appendix C.3.5, and found no significant difference, i.e., the divergence error for our method changed from 1.29 to 1.30.

**Window Function**: We also evaluated different window functions regarding both the *LinCConv* baseline and our proposed basis, see Fig. 6. We observed that the choice of window function significantly impacts the network performance, e.g., *LinCConv* improved in terms of density error by 2.1 times; see Appendix C.3.2. At the same time, the Fourier basis shows a 2 times increase for the divergence error with the *Müller* window. This points to fundamental differences stemming from the basis construction. Notably, the Fourier basis has the desirable

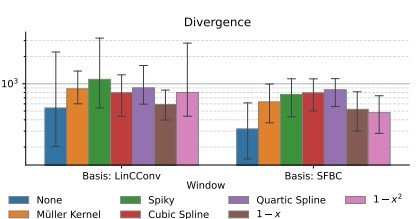

Figure 6: Window Function Study

property of giving a very good performance without requiring a window function. Finally, our evaluations demonstrate that contrary to intuitions in prior work, using window functions that are not Gaussian-shaped, e.g., a linear window, can outperform existing window functions, e.g., using a linear window showed the lowest density, velocity, and PSD errors for *LinCConv*. This highlights the importance of choosing the window, as the network's performance can be fine-tuned for a task.

**Fourier Terms**: We already observed a significantly different behavior for different choices of Fourier series terms in the toy problems and now investigate this more closely. We evaluated different variations of Fourier-based networks; see Fig. 7 and Appendix C.3.4, where we made several crucial observations. Using only even or odd symmetry basis terms did not lead to an overall stable prediction, highlighting that using either symmetry

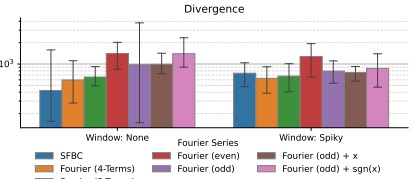

Figure 7: Fourier Series Study

exclusively is not ideal. Furthermore, by changing which harmonics are used for a given number of terms, the behavior can be adjusted to be optimal for using a window function or not. Replacing the first harmonic cosine term with a second harmonic sine term without a window function reduced the divergence error by a factor of 1.4.

**Coordinate Mappings**: Ummenhofer et al. (2019) proposed a volume-preserving mapping in the *LinCConv* approach. To expand on the brief evaluations in previous work, we used our framework to assess the importance of the coordinate mapping. In addition to the volume-preserving mapping, a mapping via the classic choice of polar coordinates serves as a baseline, details for which can be found in Appendix C.3.3. Comparing these two variants to networks trained without any coordinate mapping shows no clear advantage as the divergence error increased by up to 15%.

Overall, our evaluations show a clear and significant improvement over existing baselines for our *SFBC* approach. Not only are quantitative metrics improved, e.g., the divergence error is reduced by 30%, but these results were also achieved with fewer inductive biases.

## 4.4 TEST CASE III: INCOMPRESSIBLE 2D SPH

For our final test case, we focus on the behavior of different basis terms regarding free surfaces for an underlying divergence free solver; see Figure 9 and Appendix C.5. We use the $L_2$ difference of the ground truth and predicted particle positions for training and compute the mean distance of all predicted particle positions to the closest particle in the ground truth data for evaluation.

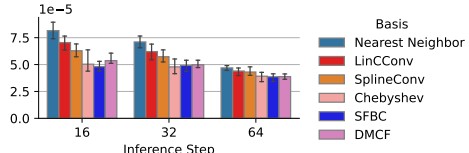

Figure 8: Point-wise distances for different inference lengths as mean values per step.

**Basis Function Comparison**: We use an architecture with 6 basis terms, 6 message-passing steps and 32 features to compare the most relevant baselines. The Fourier-based method performs best during the initial inference period, up to 2 times the training rollout length, e.g., 16 inference steps. Still, after that, most methods perform more similar; see Fig. 8 and 31. Qualitatively, the Fourier basis performed noticeably better than other methods in preserving the shape of the colliding droplets, as shown in Fig. 9. The window function in this scenario, see Appendix C.4.3, has a significant impact on performance for prior approaches, i.e., *LinCConv* showed a $3.3\times$ difference by changing the window function, and *DMCF* was only stable with the *Müller* window. In contrast, Fourier and Chebyshev bases exhibited a $20\%$ difference.

**Varying Network Size**: To verify whether the benefits of the Fourier approach hold across varied architectures, we repeated this experiment across different base term counts and network layouts, see Appendix C.4.1 and C.4.2. This evaluation shows that the same scaling behavior occurs as in test case I, implying that this behavior holds across different dimensions and problems. Furthermore, we found the same benefit for our *SFBC* approach, i.e., there is a notable improvement at virtually all network sizes with a more pronounced benefit of up to $50\%$ for smaller layouts, indicating that *SFBC* provides a robust and accurate basis for learning representations of physical systems.

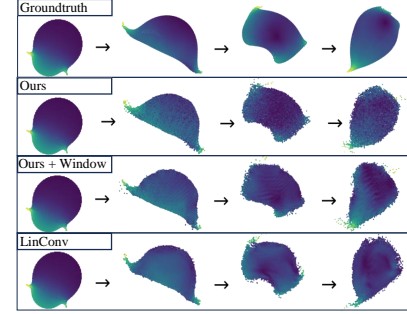

Figure 9: Qualitative evaluation of test case 3 starting on frame 10 for 32, 64 and 96 timesteps, velocity color coded.

Overall, our *SFBC* approach works best in this very challenging scenario and outperforms all baselines. We furthermore observed that some baseline methods, especially *DMCF*, performed vastly differently when using no, or simply different, window functions. In contrast, our approach remains stable in all cases while performing best without a window function.

## 5 CONCLUSIONS

We introduced the Symmetric Fourier Basis Convolution (*SFBC*) approach as a novel inherently symmetric and smooth continuous convolution approach applied to Lagrangian fluid simulations. Using this technique, we found improved performance in our challenging test cases, representing different fluid systems and solvers. Our broad evaluations identified prior inductive biases that are no longer necessary for our Fourier-based approach. At the same time, they confirm the general benefits of learning function-based convolutions in unstructured settings.

However, while we did consider a broad set of parameters, we only considered networks with up to circa 200 thousand parameters, which is a very relevant topic for future work. Our framework allows for easy and flexible explorations of continuous convolutions with new basis functions. We hope our work will inspire future investigations to improve learning methods for unstructured data sets outside of fluid mechanics (Lam et al., 2022; Reiser et al., 2022; Ahmedt-Aristizabal et al., 2021). rephrased: Furthermore, we would like to expand our formulation to encompass more general non-linear versions of Radial Basis Function networks (Broomhead & Lowe, 1988). In addition, algorithms for the initialization of the new types of basis networks, and existing methods such as *LinCConv*, are an area where substantial room for improvement exists. Finally, we would like to explore the larger space of basis functions combining different functions for different tasks, e.g., using a different basis for the radial and angular components for spherical coordinate mapping.

## ACKNOWLEDGMENTS

This work was supported by the DFG Individual Research Grant TH 2034/1-2.

## ETHICS STATEMENT

Numerical solvers for PDEs can be applied in many fields and play an essential role in design and engineering. Numerical solvers can also be utilized in other less beneficial areas, but our research is not connected to any ethically questionable application fields. Predicting the societal impact of more efficient solutions to numerical systems is challenging. However, such methods can play a positive role either directly by reducing the Carbon footprint of running numerical solutions or indirectly by solving inverse problems.

## REPRODUCIBILITY STATEMENT

We generated all datasets in our work by ourselves and using custom solvers. Consequently, it is of vital importance that the datasets and solver implementations are freely available, and, accordingly, we have made the dataset with the solver and neural network implementation available as open source code at `https://github.com/tum-pbs/SFBC`. The open-source code also includes implementations of baselines with configurations for the respective hyperparameters used in our paper.

Outside of the repository, we have also described our evaluations in Section C and provided further implementation details in Appendix Section A. We have also included detailed descriptions of our data generation setups in all test cases in Appendix Section B. We also performed a broad set of hyperparameter studies to ensure that our baseline evaluations are fair and proper, see Section C.3.5.

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

Table 1: Overview of the different Ablation Studies

| Section | Test Case | Hyperparameter | Purpose |
|---------|-----------|----------------|---------|
| C.1.1 | I | Number of Basis Terms | Symmetric learning task performance |
| C.1.2 | I | Number of Basis Terms | Anitsymmetric learning task performance |
| C.2 | I | Basis Function & Layout | Comparison against baseline methods |
| C.3.1 | II | Basis Function | Comparison against baseline methods |
| C.3.4 | II | Fourier Series Terms | Influence of including certain sine/cosine terms |
| C.3.5 | II | Random Seed | Repeatability and statistical significance |
| C.3.3 | II | Coordinate Mapping | Evaluation of prior inductive biases |
| C.3.2 | II | Window Function | Evaluation of prior inductive biases |
| C.4.1 | III | Number of Basis Terms | Validation of one-dimensional results |
| C.4.2 | III | Network Layout | Search for optimal network |
| C.4.3 | III | Basis Function | Comparison against baseline |
| C.5.1 | IV | Basis Function | Comparison against baseline |
| C.5.2 | IV | Network Architecture | Influence of Overparametrization |
| C.6 | - | Computational Performance | Comparison against baseline |

# APPENDIX

In the appendix, we will be providing additional information to the main paper. First we will be discussing additional details regarding the basis convolution model, e.g., defining all choices of basis functions we considered, see Appendix A. Next, we will be discussing details regarding the simulation setup and data generation, e.g., how the random initial conditions were chosen, see Appendix B. Finally, we will be providing additional results for all of our ablation studies, see Table 1 for an overview of the ablation studies, in Appendix C.

## A   SUPPLEMENTARY MODEL DETAILS

Here, we will discuss the mathematical foundations of our convolutional approach and mathematical definitions of all used basis functions, window functions, and coordinate mappings.

### A.1   MATHEMATICAL MODEL

In general, a convolution in 1D is a mathematical operation on two functions $f : \mathbb{R} \to \mathbb{R}$, the input function, and $g : \mathbb{R} \to \mathbb{R}$, the filter function, which are convolved using a convolutional operator $\circ$, i.e., $h(x) = (f \circ g)(x)$. This convolutional operator in a continuous form is defined as

$$(f \circ g)(x) = \int_{-\infty}^{\infty} f(x - \tau)g(\tau)d\tau. \tag{11}$$

In a Machine Learning problem, the goal is now that given an input function $f$, e.g., the input feature vector at a given set of positions, and a target output function $h$, i.e., the ground truth, to find a filter function $g$ such that $h$ is the convolution of $f$ and $g$. To achieve this, $g$ needs to be parametrized into a learnable form $g_\Theta$, where we consider three primary approaches:

- Using a Multilayer Perceptron (MLP)
- Using Radial Basis Functions, e.g., piece-wise linear functions
- Using Traditional approximation techniques, e.g., a Fourier series

Note that for the discussions here, we will only focus on the latter two approaches as they are very similar, and most of our evaluations concentrate on these approaches.

For a Lagrangian simulation, e.g., using SPH, several inductive biases can be included to help a network. As particle simulations only have a finite set of particles, we can reformulate the convolutional operation as one where interactions are based on the edges of a graph, with particles being

the vertices and the edge features being pair-wise particle distances. Accordingly, we can rewrite the convolutional operation for $n$ particles and apply it at the location of a single particle $x_i$ to be

$$(f \circ g_\Theta)(x_i) = \sum_{j=1}^{n} f(x_j) g_\Theta(x_i - x_j), \tag{12}$$

which is a direct discretization. However, this formulation is not very practical as every convolution would require an interaction for every particle-particle pairing, i.e., a computational complexity of $\mathcal{O}(n^2)$. To resolve this issue, we apply an inductive bias motivated by SPH and similar methods in limiting the interactions to a compact domain, i.e., $g_\Theta$ is zero outside of the interval $[-h, h]$, with $h$ being the support radius of the particles. Note that for convenience, we will assume $h = 1$ for further discussions (which can be ensured through appropriate scaling of the positions $x$. This results in a change in the convolutional operation as

$$(f \circ g_\Theta)(x_i) = \sum_{j \in \mathcal{N}_i} f(x_j) g_\Theta(x_i - x_j), \tag{13}$$

where $\mathcal{N}_i$ are all the neighboring particles of $i$ with $|x_i - x_j| \leq h$. Based on this convolution, some approaches introduce a normalization function $\phi(\mathbf{x})$ as

$$(f \circ g_\Theta)(\mathbf{x}_i) = \frac{1}{\phi(\mathbf{x})} \sum_{j \in \mathcal{N}_i} f(\mathbf{x}_j) g_\Theta(\mathbf{x}_i - \mathbf{x}_j), \tag{14}$$

However, an inductive bias we apply is that the filter function $g$ should behave similarly to an SPH interpolation, i.e., fewer neighbors lead to smaller interpolation values. While this, on a numerical level, leads to a kernel deficiency problem in SPH for free surfaces, it is nevertheless a widely used SPH technique. This can be achieved simply by setting $\phi(\mathbf{x}) = 1$, which has been done in prior work, e.g., in the *LinCConv* approach (Ummenhofer et al., 2019).

A further inductive bias we apply is to exclude the particle from interactions with itself and instead use a fully connected layer for self-interactions. This bias has been utilized before (Prantl et al., 2022) and is primarily motivated by gradient interpolations in SPH, which include no contribution from a particle on itself, similar to central difference schemes. If a learning task were to learn such a gradient interpolation, either explicitly or implicitly, as part of a more involved learning problem, any weight describing the self-interaction would have to be zero. This, however, overly restricts the weights, and it is more straightforward to exclude such an interaction, e.g., as

$$(f \circ g_\Theta)(x_i) = \sum_{j \in \mathcal{N}_i \setminus i} f(x_j) g_\Theta(x_i - x_j) + \omega f(x_i), \tag{15}$$

We will exclude this bias from further discussions for convenience and readability.

To parameterize $g_\Theta$, we define $g_\Theta$ as a combination of $n$ basis terms $b_i(\tau)$, with an associated weight $\theta_i$, where the overall filter function is a summation of these terms. For example, for the *LinCConv* approach, the basis terms $b_i$ describe a linear interpolation over the domain $[-1, 1]$ with each basis function being piece-wise linear. For computational efficiency, we can reformulate this summation as a dot product of a basis term vector $\mathbf{b}$ with the weight vector $\Theta$ as

$$g_\Theta(x) = \langle \mathbf{b}(x), \Theta \rangle. \tag{16}$$

We impose a further inductive bias for a two-dimensional convolution in that the filter functions are separable w.r.t. the coordinate axes, e.g., the *LinCConv* approach uses a bi-linear interpolation with piece-wise linear separable basis functions. Treating $\Theta$ as a weight matrix with basis functions $\mathbf{b}_x$, with $u$ components, and $\mathbf{b}_y$, with $v$ components, and $\mathbf{x} = [p_x, p_y]$, we can re-write the convolution:

$$g_\Theta(x) = \sum_{i=1}^{u} \sum_{j=1}^{v} B(x)_{i,j} \cdot \Theta_{i,j}; \quad B(x) = \mathbf{b}_x(p_x) \otimes \mathbf{b}_y(p_y), \tag{17}$$

Which works analogously for three dimensions. At this point an important observation is that if $\mathbf{b}_x$ and $\mathbf{b}_y$ are orthogonal bases then the resulting two dimensional basis will also be orthogonal. To prove this we first consider the definition of orthogonality for a basis in one dimension, which is

defined based on the existence of an inner product such that $\langle f, g \rangle = 0$, which is evaluated through an integral of the form

$$\langle f, g \rangle = \int_{-1}^{1} f(x)g(x)w(x)dx, \tag{18}$$

where $w(x)$ is a weight function. Given some functions, e.g., an orthogonal polynomial basis, the corresponding series can be written as $f^a(x) = \sum_j a_j F_j(x)$, where $F$ is the basis polynomial and $a_j$ is a sequence of coefficients. For orthogonality to hold for this series any inner product of two basis terms needs to be either non zero, if the same term is multiplied with itself, or $0$ otherwise. Accordingly, we can write the orthogonality constraint as

$$\int_{-1}^{1} F_i(x)F_j(x)w(x)dx = 0, \quad \forall i, j \in \mathbb{N} \wedge i \neq j. \tag{19}$$

For example given the Chebyshev polynomials of the first kind $T_n(x)$ defined through the recursion

$$T_0(x) = 1,$$
$$T_1(x) = x,$$
$$T_{n+1}(x) = 2xT_n(x) + T_{n-1}(x),$$

with an according orthogonality constraint of

$$\int_{-1}^{1} T_i(x)T_j(x)w(x)dx = 0, \quad \forall i, j \in \mathbb{N} \wedge i \neq j. \tag{20}$$

As an example, consider $T_4(x)$ and $T_3(x)$, which yields

$$\int_{-1}^{1} \left[ 8x^4 - 8x^2 + 1 \right] \left[ 4x^3 - 3x \right] \left[ \frac{1}{\sqrt{1-x^2}} \right] dx \tag{21}$$

which can be readily integrated and yields $0$ as its definite integral. The orthogonality constraint in a two dimensional function is defined as an inner product over a square region and can be written as

$$\iint_{-1}^{1} f(x,y)g(x,y)w(x,y)dxdy = 0, \tag{22}$$

with a weight function $w$ dependent on both parameters. For our basis convolution approach we now consider two, not necessarily identical, orthogonal bases $a$ and $b$ defined via an outer product as

$$B(x,y) = \sum_i \sum_j a_i(x) \cdot b_j(y), \forall i, j \in \mathbb{N}, \tag{23}$$

which means that for orthogonality to hold any inner product of two basis term $B_{i,j}(x,y) = a_i(x) \cdot b_j(y)$ and $B_{k,l}(x,y) = a_k(x) \cdot b_l(y)$ is either non zero, for $i \neq k \wedge j \neq l$, or $0$. To show that this is true we first consider the orthogonality of the basis terms $B_{i,j}$ along each cardinal direction, i.e., we want to verify that

$$\int_{-1}^{1} f_{i,j}(x,y)g_{k,l}(x,y)w(x)dx = \int_{-1}^{1} a_i(x)b_j(y) \cdot a_k(x)b_l(y)w(x,y)dx = 0, \quad \forall y \in \mathbb{R}, \tag{24}$$

and analogously along the other axes. Considering the integrand, $b_j$ and $b_l$ are independent w.r.t. the variable of integration and can be moved out of the integration to yield

$$\int_{-1}^{1} a_i(x)b_j(y) \cdot a_k(x)b_l(y)dx = [b_j(y) \cdot b_l(y)] \int_{-1}^{1} a_i(x) \cdot a_k(x)w(x,y)dx. \tag{25}$$

The right hand part of this equation is the same as the orthogonality requirement for $a$ itself and, accordingly, the integral is zero if $w(x,y)$ is identical to the weight function $w_a(x)$ for the orthogonality of $a$. The derivation along the other axes proceeds analogously. We now consider the original

orthogonality requirement again and refactor $w(x, y) = w_a(x) \cdot w_b(y)$, which yields:

$$
\begin{aligned}
0 &= \iint_{-1}^{1} f_{i,j}(x,y) g_{k,l}(x,y) w(x,y) dx dy, \\
&= \iint_{-1}^{1} f_{i,j}(x,y) g_{k,l}(x,y) \left[ w_a(x) \cdot w_b(y) \right] dx dy, \\
&= \int_{-1}^{1} \int_{-1}^{1} a_i(x) b_j(y) \cdot a_k(x) b_l(y) \left[ w_a(x) \cdot w_b(y) \right] dx dy \\
&= \int_{-1}^{1} \int_{-1}^{1} \left[ a_i(x) \cdot a_k(x) w_a(x) \right] \left[ b_j(y) \cdot b_l(y) w_b(y) \right] dx dy \\
&= \left[ \int_{-1}^{1} a_i(x) \cdot a_k(x) w_a(x) dx \right] \left[ \int_{-1}^{1} b_j(y) \cdot b_l(y) w_b(y) dy \right].
\end{aligned}
\tag{26}
$$

Thus we have shown that given two orthogonal bases $a$ and $b$, with respective weight functions $w_a$ and $w_b$, the two dimensional basis resulting from an outer product is orthogonal with respect to $w(x, y) = w_a(x) w_b(y)$. Accordingly, using any basis along an axis and using this seperable construction results in an orthogonal combined basis. Note that this can be analogously shown for three-dimensional bases that are the tensor product of three bases.

For many learning problems, $f$ is not a scalar function but describes a multi-dimensional input function $\mathbf{f} : \mathbb{R} \to \mathbb{R}^{\text{input}}$, with input features defined at all locations. Furthermore, the output of a convolution is multi-dimensional, i.e., $f \circ g : \mathbb{R} \to \mathbb{R}^{\text{output}}$, where we impose the bias that each input feature should be connected with each output feature, i.e., the convolution should be fully connected, which is in line with prior work. Accordingly, we can redefine the convolutional operator as

$$
(f \circ g_\Theta)(x_i)_{\text{output}} = \sum_{j \in \mathcal{N}_i \setminus i} \sum_{\text{input}=1}^{\text{inputFeatures}} g_\Theta^{\text{input,output}}(x_i - x_j) f_{\text{input}}(x_j),
\tag{27}
$$

which can be reformulated by summarizing all relevant input feature associations for one output as

$$
\mathbf{g}_\Theta^{\text{output}}(\mathbf{x}) = \left[ \mathbf{g}_\Theta^{1,\text{output}}(\mathbf{x}), \ldots, \mathbf{g}_\Theta^{\text{inputFeatures,output}}(\mathbf{x}) \right]^T,
\tag{28}
$$

$$
(f \circ g_\Theta)(x_i)_{\text{output}} = \sum_{j \in \mathcal{N}_i \setminus i} \langle \mathbf{g}_\Theta^{\text{output}}(x_i - x_j), \mathbf{f}(x_j) \rangle.
\tag{29}
$$

We now compute the messages $\mathbf{M}$, which are an $n \times o$ tensor (with $o$ being the number of output features and $n$ the number of edges of the graph), based on the basis function tensors $\mathbf{B}^\mathbf{x}$ and $\mathbf{B}^y$, of shape $n \times u$ and $n \times v$, respectively, as well as the input feature tensor $\mathbf{F}$, of shape $n \times i$ (with $i$ input features), as well as the weight matrix $W$, of shape $u \times v \times i \times o$, using Einsum notation as

$$
\mathbf{M}_{no} = \mathbf{B}_{nu}^x \cdot \mathbf{B}_{nv}^y \cdot \mathbf{W}_{uvio} \cdot \mathbf{F}_{ni},
\tag{30}
$$

which can be efficiently implemented using either the built-in *einsum* function of PyTorch (Paszke et al., 2019) (and related frameworks) or a more direct implementation such as Nvidia's Cutlass library (Thakkar et al., 2023), as done by Ummenhofer et al. (2019). Computing the gradients of such an operation is, mathematically, straightforward as it is just a sequence of matrix multiplications, and the shape of the actual base functions comprising $\mathbf{B}$ do not affect the shape of the gradients.

However, relying on auto-diff gradients can be impractical for such operations as the intermediate matrices can be large and must be stored for every convolution operation in a Neural Network. We chose to implement this process through a custom forward and backward operation that does not compute $B$ for all edges at once. Instead it performs this operation in batches of size $b$, which limits the memory requirements significantly as there is no need for a large intermediate matrix. Furthermore, we recompute the basis terms $B^x$ and $B^y$ during backpropagation as this allows us to only store the particle distances, which are shared for all layers, and input features, per layer, for backpropagation, instead of large matrices that are potentially different per layer. Whilst this imposes some computational overhead, it can also significantly reduce memory requirements, allowing training on GPUs with 4GByte of VRAM and less, even for three dimensional networks, with the batch size parameter $b$ trading off computational performance and memory requirements. For details on computational requirements see Appendix C.6

## A.2 WINDOW FUNCTIONS

A further inductive bias applied by prior work is including so-called window functions in the convolution operator. These window functions are inspired by the shape of SPH kernel functions, which are zero at the support radius and tend to be shaped like Gaussian functions. Imposing this bias can be achieved straightforwardly by including an additional term in the convolutional operation as

$$(f \circ g_\Theta)(\mathbf{x}_i) = \frac{1}{\phi(\mathbf{x}_i)} \sum_{j \in \mathcal{N}_i \setminus i} g_\Theta(\mathbf{x}_i - \mathbf{x}_j) f(\mathbf{x}_j) \cdot W\left(\frac{|\mathbf{x}_i - \mathbf{x}_j|}{h}\right), \tag{31}$$

where $W$ is the window function, which is zero at 1 and Gaussian-shaped. While many choices exist for these window functions, we limit our evaluations to the following window functions:

**None**: Using no window function is the most straightforward choice as this can be implemented by simply not including the window function. This term could also be defined as

$$W^{\text{None}}(r) = \begin{cases} 1, & r \leq 1, \\ 0, & \text{else.} \end{cases} \tag{32}$$

**Linear**: A naïve choice for a window function is a function that is 1 at the origin and linearly decays towards 0 at $r = 1$. While this function is not very Gaussian shaped, it does not significantly impact the shape of the learned convolutional operation besides tending towards 0 and can be defined as

$$W^{\text{Linear}}(r) = [1 - r]_+, \tag{33}$$

where $[\cdot]_+ = \max(\cdot, 0)$.

**Parabolic**: A slightly more complex variant of the prior *Linear* window function uses a parabolic decay instead of a linear one. This window function can thus be defined simply as

$$W^{\text{Parabolic}}(r) = \left[1 - r^2\right]_+ \tag{34}$$

**Müller**: This window function is based on Müller et al. (2003) and defined as a polynomial series of order 6. The advantage of this window function is that it is Gaussian-shaped but does not require any square root operations as the distance $r$ is only used squared and is defined as

$$W^{\text{Müller}}(r) = \left[1 - r^2\right]_+^3 \tag{35}$$

**Spiky**: This window function is based on Müller et al. (2003) and is purposefully designed to not be Gaussian shaped, i.e., the gradient of the kernel function does not tend towards 0 as $r$ tends to 0. This is imposed to ensure that particles keep repulsing each other in SPH, which avoids particles clumping up unnaturally, also known as the pairing problem in SPH. This function is defined as

$$W^{\text{Spiky}}(r) = [1 - r]_+^3. \tag{36}$$

**B-Splines**: Piece-wise Bezier splines are a popular choice in SPH for kernel functions and find wide usage, especially within Computer Graphics focused research (Ihmsen et al., 2013). For these functions, three popular choices exist based on the degree of the spline, i.e., cubic, quartic, and quintic splines (Dehnen & Aly, 2012). These are defined, respectively, as:

$$W^{\text{Cubic}}(r) = [1 - r]_+^3 - 4\left[\frac{1}{2} - r\right]_+^3 \tag{37}$$

$$W^{\text{Quartic}}(r) = [1 - r]_+^4 - 5\left[\frac{3}{5} - r\right]_+^4 + 10\left[\frac{1}{5} - r\right]_+^4 \tag{38}$$

$$W^{\text{Quintic}}(r) = [1 - r]_+^5 - 6\left[\frac{2}{3} - r\right]_+^5 + 15\left[\frac{1}{3} - r\right]_+^5 \tag{39}$$

A.3    COORDINATE MAPPINGS

So far, we only considered the coordinates to be given as Cartesian; however, it might be helpful to utilize other coordinate systems. Coordinate mappings $\Lambda$ are applied to the filter function $g_\Theta$ as

$$(f \circ g_\Theta)(\mathbf{x}_i) = \frac{1}{\phi(\mathbf{x}_i)} \sum_{j \in \mathcal{N}_i \backslash i} g_\Theta(\Lambda(\mathbf{x}_i - \mathbf{x}_j))f(\mathbf{x}_j) \cdot W\left(\frac{|\mathbf{x}_i - \mathbf{x}_j|}{h}\right),$$

The mapping function $\Lambda$ thus is a function $\Lambda : \mathbb{R}^d \to \mathbb{R}^d$, which could be defined in arbitrary ways, e.g., in one dimension one could utilize $\Lambda(x) = x^2$, however, we only consider commonly used coordinate mappings. Accordingly, no functional mapping exists in one dimension besides an identity mapping, i.e., using $\Lambda(x) = x$. For two-dimensions (which also can be expanded similarly to three-dimensions), we consider the following three mappings:

**Identity**: This mapping serves as the baseline approach of directly using the Cartesian as

$$\Lambda^{\text{Identity}}(\mathbf{x}) = \mathbf{x}$$

**Polar**: As the inputs to $\Lambda$ are distances, limited by a spherical support radius $h$, a direct choice for a coordinate mapping is to map the input to polar coordinates. Note that this implies some necessary changes to how the basis functions are evaluated; however, we will skip the details here for brevity as the results indicate no significant gain in using this mapping. This polar coordinate mapping can be defined straightforwardly using the $\texttt{atan2}$ function as

$$\Lambda^{\text{Polar}}(\mathbf{x}) = \left[2||\mathbf{x}||_2 - 1, \frac{1}{\pi}\operatorname{atan2}(\mathbf{x}_y, \mathbf{x}_x)\right]^T,$$

where the scaling ensures that the domain remains unchanged, i.e., $\Lambda : [-1,1]^2 \to [-1,1]^2$.

**Preserving**: The preserving mapping, proposed by Ummenhofer et al. (2019) and based on Griepentrog et al. (2008), is intended to remap the spherical support volume of the convolutions to a cubic domain to ensure that each weight influences a comparable amount of space. While this becomes more important in three dimensions, as the volume ratio between a cube and sphere is much worse than that of a square and circle, it can still be applied in two dimensions by setting the z component when performing the mapping to zero. This preserving mapping works in a two-stage process where a ball is first mapped to a cylinder and then mapped to a cube. This mapping is defined as:

$$\Lambda_{\text{ball}\to\text{cyl}}(\mathbf{q}) = \begin{cases} (0,0,0), & \text{if}||\mathbf{q}||_2 = 0 \\ \left(x\frac{||q||_2}{||(x,y)||_2}, y\frac{||q||_2}{||(x,y)||_2}, \frac{3}{2}z\right), & \text{if}\frac{5}{4}z^2 \le x^2 + y^2 \\ \left(x\sqrt{\frac{3||q||_2}{||q||_2+|z|}}, y\sqrt{\frac{3||q||_2}{||q||_2+|z|}}, \operatorname{sgn}(z)||q||_2\right), & \text{else} \end{cases} \quad (40)$$

$$\Lambda_{\text{cyl}\to\text{cube}}(\mathbf{q}) = \begin{cases} (0,0,z), & \text{if}x=0, y=0 \\ \left(\operatorname{sgn}(x)||(x,y)||_2, \frac{4}{\pi}\operatorname{sgn}(x)||(x,y)||_2\arctan\frac{y}{x}, z\right), & \text{if}|y| \le |x| \\ \left(\frac{4}{\pi}\operatorname{sgn}(y)||(x,y)||_2\arctan\frac{x}{y}, \operatorname{sgn}(y)||(x,y)||_2, z\right), & \text{else.} \end{cases} \quad (41)$$

A.4    BASE FUNCTIONS

So far, we only considered an arbitrary basis tensor $\mathbf{B}^{x,y}(\mathbf{x})$ and now we would like to discuss all the utilized base functions within this paper with particular emphasis on our proposed Fourier basis. For the versions based on Radial Basis Functions (RBFs) we assume a Cartesian coordinate system with an evenly spaced grid of central points $c_i$ among the $x$ and $y$ axes, as $x_i = -1 + i\frac{2}{u-1}$ and $y_j = -1 + j\frac{2}{v-1}$, respectively with a separation distance of $\Delta x$. Due to our assumption of separability, the functions are defined equally regardless of which axis they are applied to, with the sole exception being the *DMCF* formulation, which requires some further processing. Furthermore, for each of these central points, we can compute the relative signed distance $r_i$ from the input point and the centroid, i.e., $r_i = p - c_i$. Accordingly, each RBF is defined as $b_i\left(q = \frac{p-c_i}{\Delta x}\right)$.

Finally, most of these basis functions are designed to act as partitions of unity, i.e., with a weight vector $\Theta = \mathbf{1}$, the resulting filter function $g_\Theta$ is 1 everywhere. This is essential, as we want these

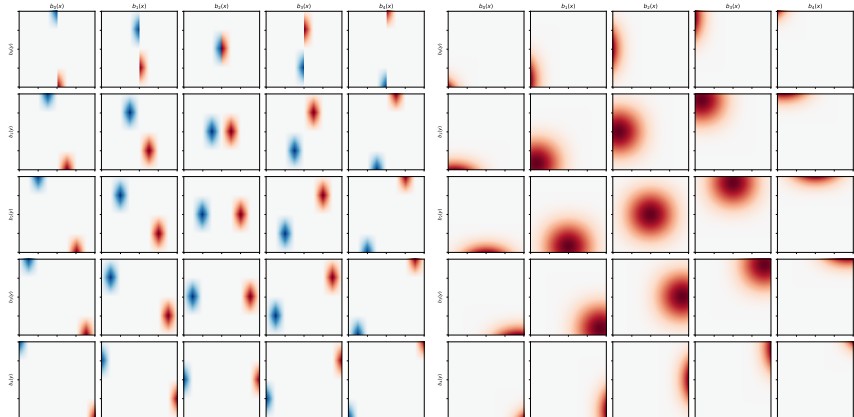

Figure 10: The $5 \times 5$ basis function combinations for the *DMCF* basis (left) and SplineConv (right).

methods to act as interpolation functions rather than approximations. Enforcing this property is possible by either normalizing the output of $g$, i.e., by introducing a corrective term $\frac{1}{\sum_i b_i(\frac{p-c_i}{\Delta x})}$, or by adjusting the definitions of the basis functions such that they do not require this corrective term. We chose the latter option as the former introduces modifications to the shapes of the basis functions, e.g., the basis functions for the corner terms might appear different than the central points, which is undesirable, and this approach has been used by prior work as well (Fey et al., 2018).

**Nearest Neighbor**: Nearest Neighbor interpolation is a simple baseline to compare all other methods against and is constructed simply as a group of piece-wise constant functions. An important note here, however, is that the function needs to be carefully designed such that there is no overlap of the basis terms on the edges of their influence radii, i.e., using $|q| \leq 0.5$ would lead to multiple bases contributing to the same input $p$, which would violate the requirement of a partition of unity. This basis term can then be defined as

$$b^{\text{NN}}(q) = \begin{cases} 1, & -\frac{1}{2} < q \leq \frac{1}{2}, \\ 0, \text{else.} \end{cases} \tag{42}$$

**LinCConv**: A higher-order interpolation scheme is a linear interpolation, also used by (Ummenhofer et al., 2019) and referred to by this name in this paper. Building a linear interpolation as a radial basis is straightforward using a piece-wise linear definition, which can be further simplified by using the $[\cdot]_+$ notation used before as

$$b^{\text{LinCConv}}(q) = [1 - |q|]_+ \tag{43}$$

**DMCF**: In prior work, a modification of the linear basis was proposed that includes only antisymmetric terms, primarily focused on antisymmetries of the combined coordinate-axes. Consequently, these terms have few symmetries along the individual axes, see Figure 10, but are always antisymmetric overall. Prantl et al. (2022) defined these terms as a bilinear basis and then modified the filter weights separately after each weight update to be antisymmetric. However, this results in the network not seeing the correct gradients and losing half of the weights to enforce antisymmetry. Instead, we define the *DMCF* basis using the *LinCConv* basis with a modification as

$$g_\Theta^{\text{DMCF}}(\mathbf{q}) = \langle \left[ \text{sgn}(\mathbf{q}_x) \mathbf{b}^{\text{LinCConv},x}(2|\mathbf{q}_x| - 1) \otimes \mathbf{b}^{\text{LinCConv},y}(\text{sgn}(\mathbf{q}_x) \cdot \mathbf{q}_y) \right], \Theta \rangle, \tag{44}$$

which utilizes all weights but increases the interpolation frequency in $x$ by a factor of 2. For completeness, an alternative formulation that also ignores half the weights but still yields correct gradients without increasing the interpolation frequency could be formulated as

$$g_\Theta^{\text{DMCF/2}}(\mathbf{q}) = \langle \left[ \text{sgn}(\mathbf{q}_x) \mathbf{b}^{\text{LinCConv},x}(|\mathbf{q}_x|) \otimes \mathbf{b}^{\text{LinCConv},y}(\text{sgn}(\mathbf{q}_x) \cdot \mathbf{q}_y) \right], \Theta \rangle. \tag{45}$$

**B-Spline/SplineConv**: A natural extension of linear interpolation is using higher-order spline functions, where the cubic B-Spline function has been used before by Fey et al. (2018). However, ensuring that these methods are partitions of unity is more challenging as it requires adjusting the spacing

of the centroids by scaling them by $1 - \frac{2}{n}$, i.e., the new centroids are $c_i^{\text{spline}} = \left(1 - \frac{2}{n}\right)\left(-1 + i\frac{2}{n-1}\right)$. Furthermore, the width of each basis function needs to be adjusted, compared to their definition as window functions used before. We evaluate these modified widths through an optimization process, i.e., we optimized this parameter such that the interpolation results in a partition of unity with minimal width per basis. This results in three B-Spline basis terms:

$$b^{\text{SplineConv}}(q) = \left[1 - \frac{|q|}{c}\right]_+^3 - 4\left[\frac{1}{2} - \frac{|q|}{c}\right]_+^3, \tag{46}$$

$$b^{\text{Quartic}}(q) = \left[1 - \frac{|q|}{c}\right]_+^4 - 5\left[\frac{3}{5} - \frac{|q|}{c}\right]_+^4 + 10\left[\frac{1}{5} - \frac{|q|}{c}\right]_+^4, \tag{47}$$

$$b^{\text{Quintic}}(q) = \left[1 - \frac{|q|}{c}\right]_+^5 - 6\left[\frac{2}{3} - \frac{|q|}{c}\right]_+^5 + 15\left[\frac{1}{3} - \frac{|q|}{c}\right]_+^5, \tag{48}$$

with normalization constants $c$ of $1.732051$, $1.936492$ and $2.121321$ (Dehnen & Aly, 2012).

**Wendland-2**: An alternative kernel often used within CFD-oriented SPH applications is part of the Wendland series of kernel functions (Dehnen & Aly, 2012; Sun et al., 2018). These functions are also polynomial but do not exhibit some of the numerical disadvantages as the B-Spline kernels and, as such, are an interesting alternative to evaluate for a machine learning context. These terms also require the modified spacing, identical to the B-Splines, and the Wendland-2 basis is defined as

$$b^{\text{Wendland-2}}(q) = \left[1 - \frac{|q|}{c}\right]_+^4 \left(1 + 4\frac{|q|}{c}\right); c = 1.620185. \tag{49}$$

**Gaussian**: A natural extension of the Spline bases is utilizing non-compact Gaussian functions, i.e., using exponential functions. As these are locally defined, i.e., only as part of the filter function and not used to determine the graph edges, non-compact functions do not impose any of the usual drawbacks. These terms also require the same spline centroid and are defined as

$$b^{\text{Gaussian}}(q) = \exp\left(-q^2\right). \tag{50}$$

**Spiky**: The Spiky kernel, discussed before, is primarily included as an additional case for ablation studies as, due to the shape of the function, this basis term cannot be normalized by adjusting the spacing and width of the basis. Nevertheless, this basis term is defined as (Müller et al., 2003)

$$b^{\text{Spiky}}(q) = [1 - |q|]_+^3 \tag{51}$$

**RBF Bump**: There is a rich history of Radial Basis Functions within RBF Interpolation theory, and we chose one of these terms that is different from the classic bases. Note that for an actual RBF basis, and thus RBF network, the centroids would need to be learned as well, which is a non-linear optimization task, to achieve proper interpolation qualities and not doing so, as done here, is unlikely to work but serves as a valuable baseline for ablation studies. Nevertheless, this basis is defined as

$$b^{\text{Bump}}(q) = \begin{cases} \exp\left(\frac{-1}{1-(0.38739618954567656r)^2}\right), & \text{if } r < \frac{1}{0.38739618954567656}, \\ 0, & \text{else.} \end{cases} \tag{52}$$

In addition to this set of radial base terms, we also consider two traditional approximation techniques that operate more globally, i.e., they are directly evaluated on $p$ instead of $q$. While many approximation bases exist, we primarily focus on Fourier series terms and Chebyshev polynomials as they find wide application for interpolating periodic and non-periodic functions.

**Fourier**: This method is the primary focus of our paper and results from using a Fourier-series as the basis. The first term of a Fourier series is always a constant function, i.e., $b_0(p) = 1$; however, for higher order terms, we could either first use the sine or cosine term. We will later include an ablation study of some variations of this ordering, but the following is the standard definition:

$$b_i^{\text{Fourier}}(p) = \begin{cases} 1, & i = 0, \\ \frac{1}{\sqrt{\pi}}\cos\left(\pi\left[\lfloor\frac{i-1}{2}\rfloor + 1\right]p\right), & i \text{ odd}, \\ \frac{1}{\sqrt{\pi}}\sin\left(\pi\left[\lfloor\frac{i-1}{2}\rfloor + 1\right]p\right), & i \text{ even}. \end{cases} \tag{53}$$

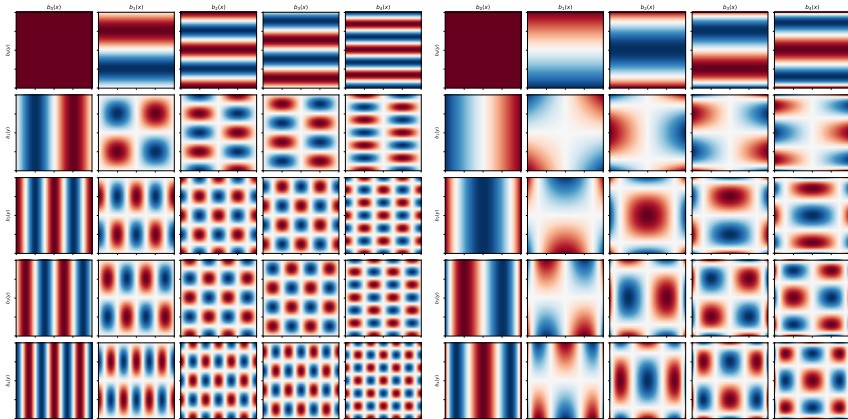

Figure 11: The $5 \times 5$ basis function combinations for a Fourier basis (left) and Chebyshev (right).

For some of the variants we evaluate in our ablation studies, we modify the first term, i.e., $b_0^{\text{Fourier}}$, as this term is inherently symmetric, but we want to evaluate purely asymmetric Fourier terms as well. To achieve this, we either (a) drop the term, (b) modify the term to be based on the sign of x, i.e., $b_0^{\text{Fourier, sgn}}(x) = \text{sgn}(x)$, or (c) use x directly, i.e., $b_0^{\text{Fourier, linear}}(x) = x$. Using the standard basis definition for $n = 5$ in two dimensions, we find symmetries and antisymmetries, see Figure 11, w.r.t. the $x$ and $y$ coordinates as well as the combined coordinates $p = [x, y]$, in a variety of configurations.

**Chebyshev**: Another popular choice in graph convolutions are Chebyshev basis functions, which are smooth, inherently symmetric, and antisymmetric. For these terms, we primarily consider Chebyshev polynomials of the first kind; see Figure 11 for a visualization of this basis, defined as:

$$
\begin{aligned}
b_0^{\text{Chebyshev}}(p) &= 1 \\
b_1^{\text{Chebyshev}}(p) &= p \\
b_i^{\text{Chebyshev}}(p) &= 2p\, b_{i-1}^{\text{Chebyshev}}(p) - b_{i-2}^{\text{Chebyshev}}(p),
\end{aligned}
\tag{54}
$$

where the magnitude of the basis terms is bound by the domain $[-1, 1]$. Furthermore, for some ablation studies, we also considered Chebyshev polynomials of the second kind, defined as

$$
\begin{aligned}
b_0^{\text{Chebyshev2}}(p) &= 1 \\
b_1^{\text{Chebyshev2}}(p) &= 2p \\
b_i^{\text{Chebyshev2}}(p) &= 2p\, b_{i-1}^{\text{Chebyshev2}}(p) - b_{i-2}^{\text{Chebyshev2}}(p),
\end{aligned}
\tag{55}
$$

which are not bound in magnitude by the domain $[-1, 1]$.

## B  EXPERIMENTAL DETAILS

We focus on several datasets with a focus on quantifiable behavior, and describe in th following how these datasets were generated. Existing datasets oftentimes involve behavior that is difficult to numerically quantify or behavior where a loss metric is difficult to relate to the visual perception of the simulation.

We chose to create our datasets to include a wide variety of SPH problems across one, two, and three dimensions to make them as versatile as possible. In this section, we will focus on the setup of the solvers for the generation of our datasets, as well as basic network parameters and data augmentation techniques. Accordingly, we will be discussing each test case in a separate sub-section.

Overall, there are few similarities between our different test cases and datasets; however, some familiarities exist. Notably, all of our datasets, as well as the implementations of our used classical solvers, and the implementation of our network architecture is available as open source code at *https://github.com/tum-pbs/SFBC*. Furthermore, we utilized the Adam (Kingma & Ba, 2015) optimizer in all cases. We built all of our code, including the SPH simulations, using PyTorch and PyTorch Geometric for graph processing, e.g., neighborhood searches.

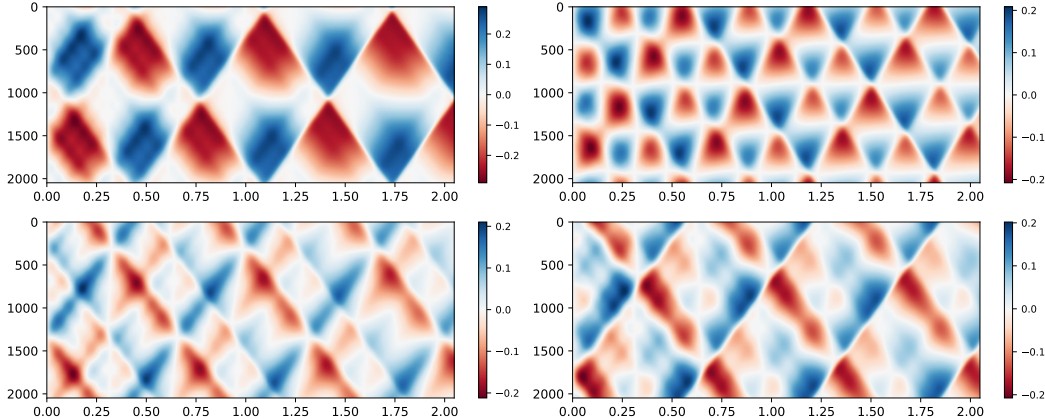

Figure 12: Visualization of the test dataset for test case I consisting of 4 simulations of 2048 timesteps each. Each plot represents a simulation from the dataset, with simulation time on the $x$ (from 0 to 2.048s) axis and particle index on the $y$ axis (from 0 to 2048), with color mapping indicating particle velocity (blue for up and red for down). The initial conditions were randomly generated with identical simulation parameters, e.g., viscosity, across all simulations and not specially selected for good or bad testing performance for some methods.

### B.1  TEST-CASE I: ONE-DIMENSIONAL COMPRESSIBLE SPH

Lagrangian fluid simulations pose several problems for machine learning techniques compared to Eulerian simulations. One primary consideration is the particle spacing and distribution, which, for Lagrangian simulations, changes as the flow evolves. This means that if the predictions of a network lead to errors on one inference step, then the input to the next inference step is out of distribution, compared to the ground truth. Accordingly, it is vital that machine learning techniques can deal with changing and varied particle distributions and that the network solution does not learn behavior that is only useful for a very narrow set of distributions. Consequently, data augmentation techniques are essential during training; however, in this test case, we want to specifically investigate how different particle distributions affect the network performance.

For an incompressible simulation, the spacing between particles is somewhat constrained due to incompressibility limitations; however, for a compressible simulation, we can generate particle distributions that cover a much broader range of inter-particle spacings. Furthermore, by reducing the dimensionality of the simulation, we can perform a much more focused investigation of this relationship. Accordingly, our test case is a compressible one-dimensional SPH simulation that primarily investigates differences in methods' capabilities to handle a broad range of particle distributions.

Our underlying simulation model, in this case, is a simple Equation of State (EoS) based explicit time integrator using a Runge-Kutta integrator of fourth order (Antuono et al., 2012) combined with an explicit diffusion term for the velocity field. As the EoS, we chose an ideal gas equation with no influence of energy or temperature, as the general NNTI approach only predicts changes in position and not in energy. For the diffusion term, we chose the approach of Price (2012), although the exact choice of diffusion term does not matter for our evaluations. Finally, we chose a periodic domain and a simulation region of $\Omega = [-1, 1]$ as boundary conditions.

To set up the initial conditions, we use a two-step process where we first create a random density profile for a fixed domain of $[-1, 1]$ and then place a set number of particles (2048) such that their summation density $\rho_i = \sum_j m_j W_{ij}$ is identical to the random density profile. To produce the initial density profile, we utilize *Perlin* noise (Perlin, 1985), a perceptually isotropic gradient noise commonly used for procedural generation. Perlin noise, in general, is based on a pseudo-random process based on a seed $s$ that generates an $n$-dimensional noise texture of frequency $f$ along each dimension, defined as $P_f : \mathbb{R}^n \rightarrow [-1, 1]$. A common technique is to combine multiple Perlin noise textures of increasing frequency to generate a so-called *Octave* noise using several octaves $n_{\text{Octave}}$,

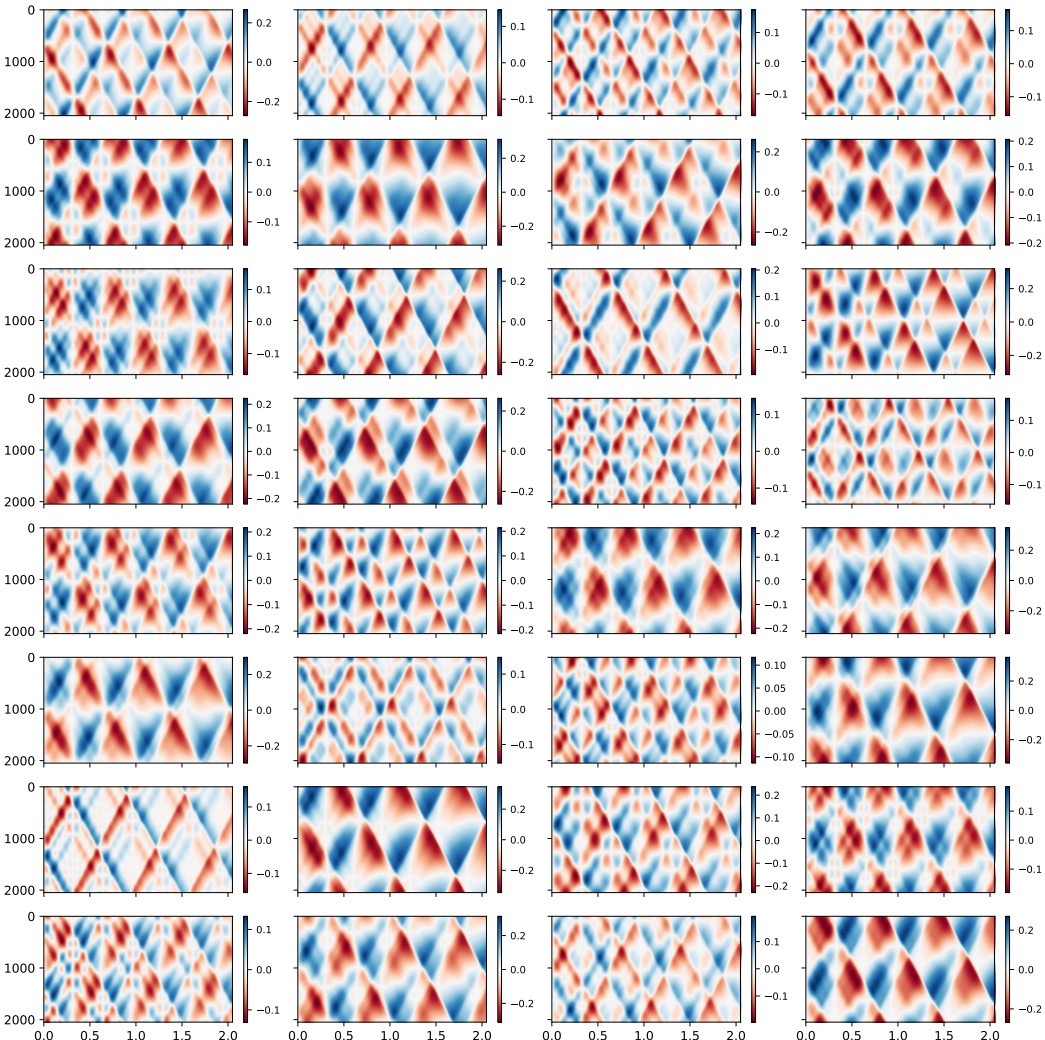

Figure 13: Visualization of the training dataset for test case I consisting of 32 simulations of 2048 timesteps each. Each plot represents a simulation from the dataset, with simulation time on the $x$ (from 0 to 2.048s) axis and particle index on the $y$ axis (from 0 to 2048), with color mapping indicating particle velocity (blue for up and red for down). The initial conditions were randomly generated with identical simulation parameters, e.g., viscosity, across all simulations.

a lacunarity $l$ denoting the increase in frequency per Octave and a mixing factor $\alpha$, as

$$\text{Octave}_f(\mathbf{x}) = \sum_{i=0}^{n_{\text{octave}}-1} \alpha^i P_{f \cdot l^i}(\mathbf{x}). \tag{56}$$

To generate our dataset, we set $n_{\text{octave}} = 4, \alpha = \frac{3}{4}$ and $l = 2$. As the noise is in the range $[-1, 1]$, we need to modify the noise amplitude by scaling the noise by $\frac{1}{4}$ and adding an offset of 2, meaning the density field is a pseudo-random periodic density field in the domain $\left[1 + \frac{3}{4}, 2 + \frac{1}{4}\right]$.

While it is possible to use this octave noise as the density field directly, this would lead to discontinuities at the boundaries. One technique to sample periodic noise is embedding a regular structure into a higher dimensional noise function. By embedding a circle, which has constant curvature, into a two-dimensional Perlin-noise function and then using a parametric description of a circle, i.e., $\mathbf{x} = [r\cos\theta, r\sin\theta]$ for $\theta \in [-\pi, \pi)$, we can generate a periodic one-dimensional noise.

This density field is then sampled on a discrete grid, with 2048 grid points, yielding a discrete Probability Density Function (PDF). We then compute a discretized Cumulative Density Function (CDF) by integrating the discrete PDF. This discrete CDF can then be used to construct an approximate inverse CDF by building a linear interpolation using the discrete CDF as $x$-coordinates and a regularly spaced range from $[-1, 1]$ as the $y$-coordinates. We then regularly sample the domain $[0, 1]$ using $n_{\text{particles}}$ points and sample the inverse CDF using these locations to find the initial particle locations.

After sampling the particles, we initialize the simulation with a constant initial velocity of $0$, a particle size $a = \frac{1}{n_{\text{particles}}}$, a support radius $h = 4 * a$, diffusion coefficients $\alpha = 1$ and $\beta = 2$, a stiffness coefficient $\kappa = 10$, a numerical speed of sound $c_s = 10$, particle rest density $\rho_0 = 1000$ and with a fixed timestep of $\Delta t = 10^{-3}$. We utilize 36 random initial conditions to generate our dataset and evaluate 2048 timesteps each. Out of these, we use 32 (chosen randomly) as the training set, see Fig. 13, and the remaining 4 as the testing set, see Fig. 12.

For the training task, we compute a multi-step update, i.e., for each timepoint in the simulation, we compute the average velocity $v_t^{\text{prior}}$ over the prior $s = 16$ timesteps and the next $s = 16$ timesteps $v_t^{\text{next}}$. The network's goal is to predict $v_t^{\text{next}}$ based on $v_t^{\text{prior}}$ and the particle area, which are the only quantities that influence the particle behavior in the underlying simulation. During training, we compute the $L_2$ difference between ground truth and network prediction for a batch size of 4 without temporal unrolling, where each batch is the result of picking 4 random samples across the entire training dataset where each training timestep is used at-most-once. Our training consists of 5 epochs, each consisting of 1000 weight updates, with an initial learning rate of $10^{-3}$ that is halved after every epoch. To evaluate the test error, we utilize all four test simulations and evaluate the $L_2$ difference between ground truth and prediction for the time points $t = [0 + s, 128 + s, 256 + s, 1024 + s]$

For the simulation and training on this dataset, we utilized a naïve neighbor search that computes the distance of all particles to all particles, with a computational complexity of $\mathcal{O}(n^2)$ and calculated the distance of particles using floating point modulo operations to implement the distance checks. Due to this implementation, the positions of particles may lie outside of the domain $[-1, 1]$, and a modulo operation needs to be applied to find the actual position of a particle within the simulation domain. Furthermore, this means that the number of particles in the simulation stays constant, and no ghost particles are utilized to implement the periodic boundary condition. Accordingly, the position is consistent, i.e., particles are never mapped into or out of existence based on where they are in the simulation, and their position is the direct result of the integrated velocity.

### B.2 TOY PROBLEMS

SPH simulations are generally built on interpolation operations using either the kernel function or the gradient of the kernel function. Accordingly, if a neural network is supposed to learn an entire simulation computed using SPH, then the network should be capable of learning interpolation and gradient interpolation tasks. Conversely, if a network cannot learn either operation, the likelihood of the network learning an entire simulation step is low. Consequently, we devise a simple set of two toy problems to evaluate the network's ability to learn these tasks.

As the basis of these evaluations, we utilize the first test case, i.e., the one-dimensional compressible simulation data set, with a modified learning task. In general, an SPH interpolation is defined as

$$\langle A_i \rangle = \sum_{j \in \mathcal{N}_i} A_j \frac{m_j}{\rho_j} W_{ij}, \tag{57}$$

with $A$ being some quantity, $m$ being the mass of a particle, $\rho$ being the density of a particle computed using an SPH interpolation, and $W_{ij} = W(x_j - x_i, h)$ is a Gaussian-like kernel function (Monaghan, 2005). The most straightforward SPH interpolation, then, is a density estimate that can be computed by setting $A = \rho$, i.e., (Koschier et al., 2019)

$$\rho_i = \sum_{j \in \mathcal{N}_i} m_j W_{ij}, \tag{58}$$

which can be modified to computing the number density $\delta_i = \frac{\rho_i}{\rho_0}$ (Solenthaler & Pajarola, 2008)

$$\delta_i = \sum_{j \in \mathcal{N}_i} a_j W_{ij}, \tag{59}$$

with $a = \frac{m}{\rho_0}$ being the area of a particle. This number density can be interpreted as a quantity that depends on a single message-passing step using a cutoff radius of $h$, a vertex feature $a$, and a symmetric convolutional filter function $W$. Note that in SPH, the magnitude of the kernel function scales inversely with the particle scale. Accordingly, it is convenient to use a vertex feature of $1$ and treat $aW_{ij}$ as a single term. The first learning task is to learn a convolutional filter function $g_\Theta$ as

$$\sum_{j \in \mathcal{N}_i} a_j W_{ij} = \sum_{j \in \mathcal{N}_i} 1 \cdot g_\Theta(x_j - x_i, h). \tag{60}$$

In addition to the SPH interpolation, we also consider an SPH gradient interpolation, which, in its most naïve formulation, can be defined simply as (Koschier et al., 2019)

$$\langle \nabla_i A_i \rangle = \sum_{j \in \mathcal{N}_i} A_j \frac{m_j}{\rho_j} \nabla_i W_i j, \tag{61}$$

where $\nabla_i$ denotes the spatial derivative with respect to the position of particle $i$. Note that many more advanced formulations for gradient interpolations are commonly used in SPH, which, e.g., are accurate for constant fields (Price, 2012), but for our toy problem, a naïve formulation suffices. We then derive a learning target analogous to the previous one as

$$\nabla_i \delta_i = \sum_{j \in \mathcal{N}_i} a_j \nabla_i W_{ij}. \tag{62}$$

This task is, again, a single message-passing step; however, this time, the convolutional filter function to be learned has to have odd symmetry instead of even symmetry for the kernel interpolation.

For both toy problems, we evaluate the performance of a single message-passing step continuous convolutional network with no activation functions being used, as we want to investigate how the basis functions can work as interpolators for the SPH interpolations. In addition to the CConv approach, we consider more generic GNNs where we still utilize only a single message-passing step. Otherwise, the training setup is identical to the one in test case I.

### B.3 TEST-CASE II: TWO-DIMENSIONAL WEAKLY-COMPRESSIBLE SPH

The focus of our second test case is quantifiability within a CFD-oriented context. Existing test cases often focus on the visual appearance, e.g., collisions of random blobs with challenging to-quantify behavior. In contrast, our test case setup can be adjusted and has clearly quantifiable behavior.

We use a closed two-dimensional domain $\Omega \in [-1, 1]^2$ with a no-slip boundary condition and a random initial velocity field to set up our test case. However, using a random per-particle velocity would not be very useful as this would lead to an arbitrary amount of divergence in the first timestep that can readily lead to problems for numerical solvers. Accordingly, the random-velocity field needs to be divergence-free at initialization; however, this is easier said than done.

The initial particle configuration for all of our simulations is a regular spaced grid of $64 \times 64$ particles such that the summation density of all particles at $t = 0$ is equal to the rest density. This sampling means that there exists a regular grid with cell centers identical to the particle positions, which makes the initialization more straightforward. The initial velocity field is then computed using curl noise (Bridson et al., 2007) sampled on the grid and then resampled back to the particle set.

The idea behind curl noise is that the gradient of the curl of a random potential field can be used to create a velocity field that, by definition, is divergence-free. The initial potential field is sampled based on a two-dimensional Octave noise field, using Perlin-noise, as this smooth, relatively low-frequency noise leads to large structures that can be easily simulated using direct numerical solvers. However, this potential field would still create issues in the actual simulation as it does not respect the boundary conditions. As we utilize a no-slip boundary condition, the most straightforward way is to ensure that the potential field is constant at the boundary, e.g., $0$, such that the curl is also $\mathbf{0}$. Based on the suggestions by Bridson et al. (2007), this can be ensured by linearly blending the potential field to be $0$ at the boundary and using a gradient along the boundary normals to *blend* the potential field from its actual value with $0$ based on the boundary distance.

The next step is to compute the curl of the potential field, which is done on the grid representation of the simulation domain using a first-order central difference scheme. This curl is then directly

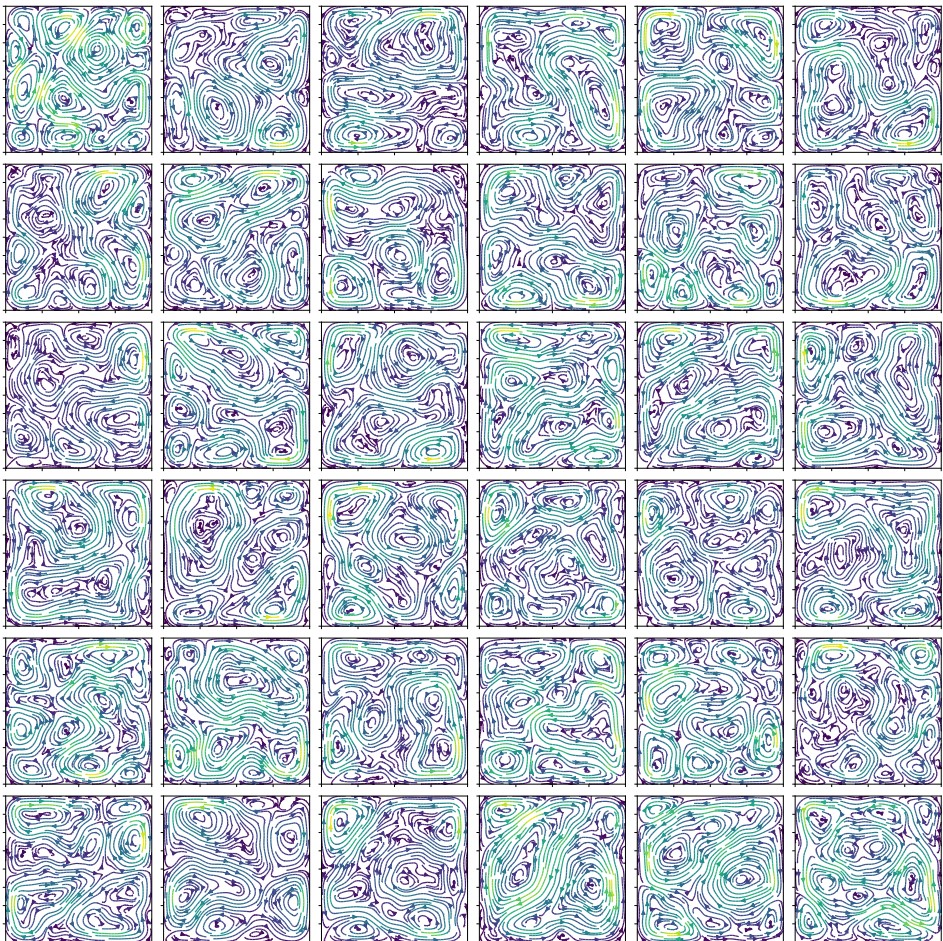

Figure 14: Visualization of the initial conditions of the 36 initial conditions used for the training dataset for test case II. The simulations here are run using a weakly compressible SPH model with an initially divergence-free velocity field with solid boundary conditions outside the shown areas. Color mapping indicates velocity magnitude, and streamlines indicate the flowfield at $t = 0$ computed by mapping the particle velocities to a regularly sampled grid using SPH interpolants.

mapped to the particles, where we utilize an SPH gradient interpolation to evaluate the initial velocities of all particles. Note that it is crucial to compute the gradient using an SPH gradient operator as, otherwise, the divergence of the velocity field is not quite zero.

We can then generate our dataset. To make the test cases more interesting, we use a lower frequency potential field, i.e., the test simulations have larger structures that are not seen as such during training. Overall, we generate 32 training samples, see Fig. 14, and 4 testing samples, see Fig. 15.

The simulations are then performed using a $\delta$-SPH method (Marrone et al., 2011) with a Runge-Kutta time integration scheme of fourth order (Antuono et al., 2012). Each simulation contains 2048 timesteps where the learning task is to compute the position of a particle for $s = 4$ timesteps at once. Note that the underlying $\delta$-SPH simulation uses a so-called continuum density formulation, i.e., the density of any given particle is not the result of an SPH interpolation, as in test case I, but is integrated over time using the continuity equation. However, we only provide the velocity of particles as features, making the learning task ambiguous and more complex, thus making it more interesting. Furthermore, due to a combination of the multi-step prediction task, the RK-4 time integration, and the particle scaling, the receptive field of the particles in the simulation covers the entire domain, i.e., each particle influences all other particles; however, we utilize the architecture of Ummenhofer et al. (2019), which only uses 4 rounds of message-passing. As such, the problem is

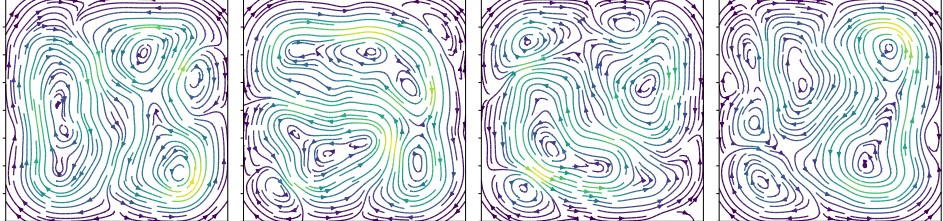

Figure 15: Visualization of the initial conditions of the 4 initial conditions used for the test dataset for test case II. The simulations use a weakly compressible SPH model with an initially divergence-free velocity field with solid boundary conditions outside the shown areas. Color mapping indicates velocity magnitude, and streamlines indicate the flowfield at $t = 0$ computed by mapping the particle velocities to a regularly sampled grid using SPH interpolants. Note that the frequencies are lower than the training samples to better evaluate generalization.

made even more difficult as the networks are not provided with all seemingly necessary information. All of these limitations make the learning task here exceedingly difficult as the network has (a) not all necessary features, (b) a receptive field that is too small, and (c) the testing samples are out-of-band.

A further complication arises from the nature of the simulation. Vortices in SPH can be challenging to simulate as they often lead to the formation of *holes*, i.e., regions of the simulation domain with no particles. While we avoid this during the simulation through particle shifting (Rastelli et al., 2022), the network must also learn this correction on top of the physics update. However, this is made more challenging as the dataset only considers already corrected particle positions, i.e., the network never sees a correction explicitly and, thus, has to implicitly learn this behavior.

We compute various metrics to evaluate this dataset and focus on metrics calculated on a regular grid that spans the closed simulation domain. This regular grid data can be computed by performing an SPH interpolation for all grid centers, e.g., for velocity and density. The regularized data has the advantage of not overly punishing particle swaps, i.e., two particles may be located at each other's location in relation to the ground truth, leading to a significant per-particle error even though the underlying velocity field is accurately predicted. We compute the $L_2$ of the density, velocity, and divergence field on the regularized data, where the latter is calculated using an SPH gradient interpolation on the cell centers. Finally, we also compute a Power Spectrum Density (PSD) on the grid velocity field to compare the overall structures, irrespective of their spatial location.

For training, we utilize 20 epochs, each consisting of 1000 weight updates, with an initial learning rate of $10^{-3}$ that is halved after every five epochs. We start the training with an initial rollout length of 1, which is increased every second epoch by 1, up to a maximum unroll length during training of 10. Evaluations are computed on all test samples from the dataset for frames $[s, 512 + s, 1024 + s, 2175 + s]$ for an inference length of 64, where $s$ is the prediction distance chosen as 16. During training, we use a batch size of 2 and augment all samples by including a random jitter for the particle positions, set to be normally distributed with a standard deviation of $0.01h$, and a random rotation (Brandstetter et al., 2022a). Note that we do not apply noise during the unroll while training and only use the augmentation on the first step, similar to Prantl et al. (2022).

### B.4 TEST-CASE III: TWO-DIMENSIONAL INCOMPRESSIBLE SPH

Our other test cases so far consisted only of particle distributions where particle neighborhoods were always fully occupied. However, one of the significant advantages of SPH-based simulations is the ability to handle deforming free surfaces where particle neighborhoods are not fully occupied. Accordingly, it is essential that our methods are also applicable in simulations involving free surfaces. A common task for this problem is the collision of two or more blobs of liquid in free space, and while these simulations can be visually impressive, they are often not very challenging in their dynamics. Our dataset aims to be a more challenging variation of such test cases by increasing the difficulty of the physical setup and using a more accurate solver.

Generating the initial conditions in this test case is much more straightforward than the prior two test cases as we only generate two spheres of liquid, with a larger one located at the origin and a smaller

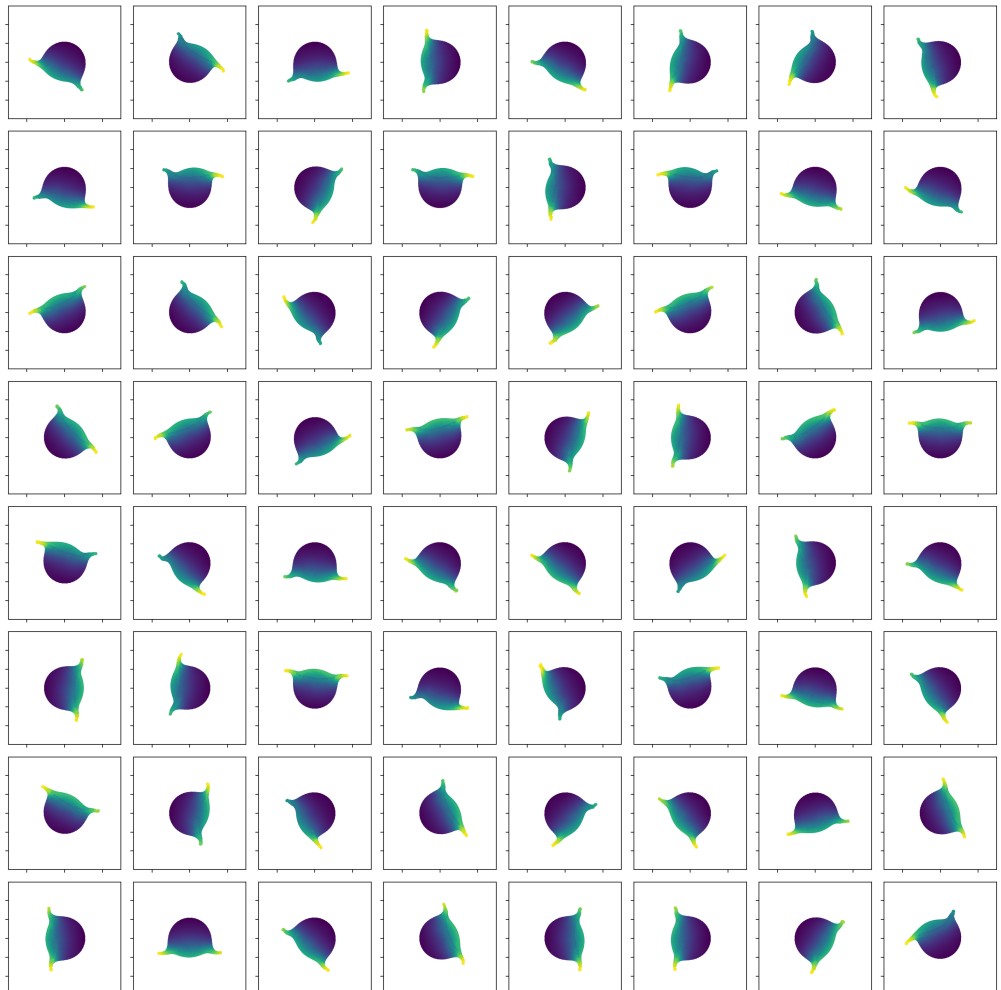

Figure 16: Visualization of the training dataset for test case III showing all 64 constellations. Each initial condition is chosen by randomly placing a smaller sphere around a larger sphere with random offsets, causing generally similar but different behavior. Color mapping indicates particle velocity from low (blue) to high (yellow) after 32 timesteps.

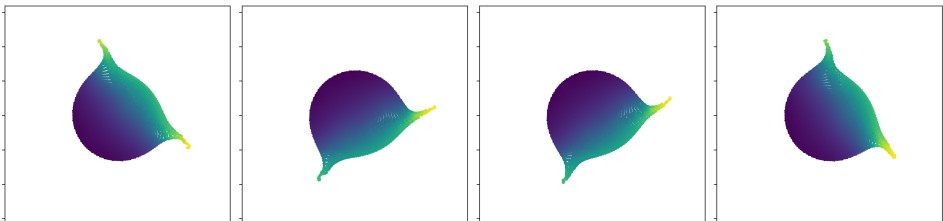

Figure 17: Visualization of the testing dataset for test case III showing all 4 constellations. Each initial condition is chosen by randomly placing a smaller sphere around a larger sphere with random offsets, causing generally similar but different behavior. Color mapping indicates particle velocity from low (blue) to high (yellow) after 32 timesteps. Training samples are chosen as in-band, relative to the training samples, but with unseen rotations and offsets.

one randomly placed orbiting the larger one. We then give the orbiting sphere an initial velocity towards the origin and apply an offset in an orthogonal direction of the velocity to offset the impact location to generate more variety. We then move the smaller sphere to collide with the larger sphere

after a small number of timesteps, where this number of timesteps is consistent across all samples. Based on this setup, we now generate 68 simulations, with 4 random simulations, see Fig. 17, used as in-band testing samples and the remaining 64 used for training, see Fig. 16.

As the underlying solver we utilize a divergence-free SPH solver set to $0.001\%$ maximum density error and $0.01\%$ maximum divergence error, resulting in, on average, 64 substeps per simulation timestep. We base this solver on a summation density approach and use a simple kinematic viscosity formulation with a low viscosity coefficient to stabilize the simulation. Time integration is done using an RK-4 scheme with diffusion freezing. Note that no boundaries exist in this simulation, i.e., no boundary treatment is necessary, and no such provisions are included in the network.

We then add a potential gravity field that is based on the distance $d$ and direction $\mathbf{r}$ of a particle relative to the origin, i.e., $|\mathbf{x}|$ and $\mathbf{x}$, respectively as

$$\frac{d\mathbf{v}}{dt} = -\frac{1}{2}g^2\mathbf{x}\,|\mathbf{x}|^2\,, \tag{63}$$

for some gravitational constant $g$. This gravitation is applied to the current velocity as an advection velocity, see Sec. 2.1, where the network task is to predict a particle's position change for a single timestep. Note that in contrast to the prior scenarios, we train a single timestep prediction here as the scenario is already challenging as is. We then train all networks for 20 epochs, each consisting of 1000 weight updates with a batch size of 4. We start the training with an initial rollout length of 1 and increase the rollout every second epoch by 1. Finally, we start with an initial learning rate of $10^{-3}$ and decrease it every 100 iterations to reach $10^{-5}$ at the end of training.

### B.5    TEST-CASE IV: THREE-DIMENSIONAL TOY PROBLEMS

To expand our evaluations to three dimensions, we chose to create a simple setup to verify the findings from one and two dimensions. Accordingly, this dataset is designed to serve as a straight forward test case for SPH kernel and gradient interpolations, similar to the problems in test case I, but in a three-dimensional space. Generating the data was performed by creating a regular grid of $16^3 = 4096$ particles on a regular grid spanning the unit cube $[-1, 1]^3$, with an initial volume per particle of $v = 8/4096$ and a support radius such that each particle has 32 neighbors in this initial constellation under periodic boundary conditions. For the dataset we apply a random three dimensional periodic Perlin noise to the volume of each particle by multiplying the particle volume with a random scaling in $[-1, 1]$. While this does result in negative volumes, this is mathematically not an issue as the SPH interpolant and gradient work regardless of particle volume, albeit they only make physical sense with positive masses. We then add an additional normal distributed offset $\mathcal{N}(0, 0.05h)$ to each particle position to add more variation to the dataset. This results in a configuration as shown in Fig. 18, where we generate 1024 such configurations for training and 4 for testing. For each of these configurations we then compute the SPH density and the gradient of the density using a naïve gradient formulation, analogous to the toy problems for test case I.

In the training setup, we utilize 4000 weight updates, i.e., approximately 4 epochs, with a learning rate set initially as $10^{-2}$ and ending at $10^{-4}$ with a reduction in learning rate every 25 weight updates. We perform no data augmentation for this toy problem and utilize as features either the particle density $\rho_i$, evaluated using a standard SPH interpolant, or a normalized volume $\hat{V}_i = \frac{V_i}{h^3}$, chosen such that if a window function identical to the ground truth kernel function was used, the ideal network weights would be 1 for a linear basis.

## C    ABLATION STUDIES / EVALUATION

This section will discuss the various ablation studies we performed as part of our research and summarize the respective results. We will also provide more in-depth data for all the experiments discussed in the paper, both numerically and visually. See Table 1 for an overview of all experiments.

### C.1    TOY-PROBLEMS

Our first set of evaluations focused on test case I, see Appendix B.1, where we wanted to investigate how different basis functions could learn a simple symmetric and antisymmetric task. In addition

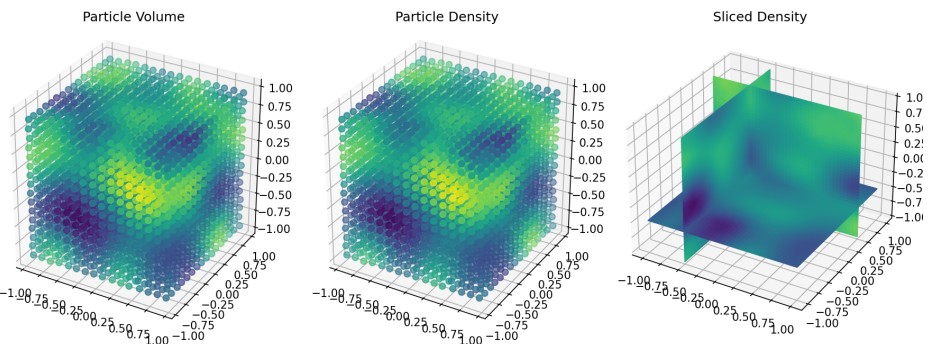

Figure 18: Visualization of the setup for test case IV showing initial particle volume (left), resulting particle density (middle) and a slice through the density field within the simulation domain. Color mapping indicates, respectively, volume, density and density, with purple indicating low values and yellow high values.

to the evaluations shown in the main paper, we provide additional results for the basis function approaches in the form of tables and an additional ablation study for MLP-based approaches. The latter primarily serves to verify that our implementations of the MLP-based approaches can learn *something* and how they relate in performance to the basis function approaches. It is important to note, however, that a direct comparison in this case of basis function approaches and MLP on a parameter count basis would not be a fair comparison as the inductive biases built into basis function approaches would result in parameter counts that are orders of magnitude different in a test case specifically in-tune with the inductive biases. Section C.1.1 will discuss the results for learning the kernel function, and Section C.1.2 will discuss the results for learning the kernel gradient. Numerical results are provided in Table 2 and 3.

### C.1.1 SPH KERNEL INTERPOLATION

Within SPH, kernel interpolations are an important and central aspect, and hence, we investigate learning these interpolations as a fundamental and central task. To investigate the ability of different basis functions and network architectures to learn an SPH kernel interpolation, we set up a simple toy problem, as discussed in Appendix B.1. We now performed two separate ablation studies, one focused on basis convolutions and one focused on MLP-based approaches.

**Basis Convolutions**: Continuous Convolutions using basis functions are inherently built around inductive biases that SPH exhibits, and, accordingly, they should perform relatively well in this setup. To perform our evaluations, we evaluate 10 different basis terms for a single message-passing step architecture with no activation functions or normalization. The goal here is to determine which basis functions best work to approximate an SPH kernel, and accordingly, the expectation would be that symmetric and smooth methods should perform ideally. Note that we used no window function as using a window function, especially if it is identical to the SPH kernel function, would make the learning problem trivial.

**Baseline Methods**: We evaluated a few baseline methods in this problem, i.e., *LinCConv*, *SplineConv*, *DMCF* and also a naïve network build using nearest neighbor interpolation. We observed that the best-performing basis in this case was *LinCConv* as this approach yielded an $L_2$ error of $2.5 \cdot 10^{-5}$, which is lower than either *SplineConv* ($5.13 \cdot 10^{-4}$) and *Nearest Neighbor* ($4.3 \cdot 10^{-4}$), while *DMCF* was not able to learn anything in this task due to its inherent antisymmetric nature. Furthermore, we observed that the number of basis terms significantly impacted the results, i.e., using only 4 basis terms resulted in a noticeably worse result than using 32 terms due to the inherent lack of smoothness for these basis functions.

**Fourier Methods**: Using a Fourier-basis resulted in a much better result than the baseline methods; however, it becomes apparent that increasing the parameter count yielded a worse network performance. This effect is primarily due to the nature of a Fourier Series, as higher term counts do not

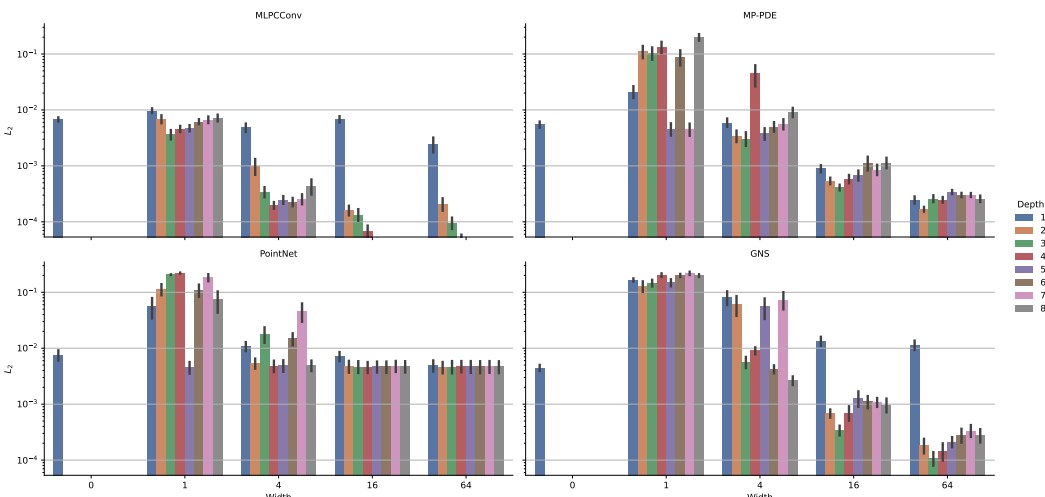

Figure 19: This figure evaluates the correlation between the hidden layer architecture of MLP-based graph networks and the ability of a single-layer message-passing network to learn an SPH kernel function for various basis functions. The x-axes correspond to the number of features per hidden layer, and the y-axis corresponds to the $L_2$ error evaluated using the test dataset from test case I for $4$ different frames $t = [0, 0.512, 1.024, 2.048]$, with color indicating the number of hidden layers. The plots show the behavior of the MLPCConv (top left), Message Passing PDE (top right), PointNet (bottom left), and GNS (bottom right) network architectures.

improve the smoothness of the convolution but include higher and higher frequency terms. As the underlying kernel function here is relatively low in frequency, including high-frequency terms requires the network to learn that they have zero contribution, which is difficult to achieve. This is supported by the fact that the Fourier series itself, with two terms by harmonic, shows an apparent behavior of decreasing in performance for every two terms added, i.e., every time a new harmonic is added, the performance decreases, while the Fourier series only consisting of even symmetry terms decreases in performance for every term added. Finally, the Fourier series consisting only of odd symmetry terms was not able to learn only a very coarse result in this task due to its inherent anti-symmetry; however, the result is much better than the *DMCF* approach as the first term is a constant term that is still symmetric.

**Chebyshev Basis**: The Chebyshev basis variant performed reasonably well in this case but did not show much improvement as more terms were included; however, it still managed to outperform *LinCConv* for most parameter counts due to the inherent smoothness.

**MLP-based methods**: Considering the results for the MLP-based GNNs, see Figure 19, we can make several observations. Firstly, the *PointNet* approach fails to learn anything meaningful for any architecture, which is somewhat expected as this architecture does not consider the interconnectivity of the particles and is only provided the position and area of each particle, which is not enough information to reconstruct an SPH kernel interpolation. Secondly, both *GNS* and *MP-PDE* performed somewhat comparably with a clear trend that highlights that increasing the number of hidden layers beyond two does not provide much benefit while increasing the number of neurons per hidden layer shows a clear and noticeable impact on the results. This is in line with the recommendations of Sanchez-Gonzalez et al. (2020) to use a hidden architecture that is shallow, i.e., with $2$ hidden layers, and wide, i.e., with $128$ neurons per layer. Finally, *MLPCConv* performed better than all other MLP-based approaches primarily due to its inductive bias of representing convolutions.

Table 2: This table shows the results of an ablation study to find the optimal number of basis terms for learning a purely symmetric SPH kernel interpolation. The results were obtained as the $L_2$ loss for a simple SPH kernel interpolation evaluated on the test dataset of test case I at 4 different timepoints for 4 different network initialization seeds. Each column represents a different number of basis terms, with the best behavior for each number of basis terms highlighted in bold.

| n Basis | 1 | 2 | 3 | 4 | 5 | 6 | 7 | 8 | 16 | 32 |
|---|---|---|---|---|---|---|---|---|---|---|
| Nearest Neighbor | $\mathbf{7.07 \cdot 10^{-3}}$ | $7.06 \cdot 10^{-3}$ | $3.38 \cdot 10^{-3}$ | $1.81 \cdot 10^{-3}$ | $1.79 \cdot 10^{-3}$ | $1.45 \cdot 10^{-3}$ | $1.20 \cdot 10^{-3}$ | $1.73 \cdot 10^{-3}$ | $1.16 \cdot 10^{-3}$ | $4.31 \cdot 10^{-4}$ |
| LinCConv | $7.08 \cdot 10^{-3}$ | $7.08 \cdot 10^{-3}$ | $2.64 \cdot 10^{-3}$ | $1.65 \cdot 10^{-3}$ | $8.47 \cdot 10^{-4}$ | $8.40 \cdot 10^{-4}$ | $3.33 \cdot 10^{-4}$ | $2.73 \cdot 10^{-4}$ | $2.50 \cdot 10^{-5}$ | $8.40 \cdot 10^{-5}$ |
| DMCF | $4.01$ | $4.01$ | $4.01$ | $4.01$ | $4.01$ | $4.01$ | $4.01$ | $4.01$ | $4.01$ | $4.01$ |
| SplineConv | $7.08 \cdot 10^{-3}$ | $7.07 \cdot 10^{-3}$ | $6.71 \cdot 10^{-3}$ | $5.94 \cdot 10^{-3}$ | $5.02 \cdot 10^{-3}$ | $4.16 \cdot 10^{-3}$ | $3.69 \cdot 10^{-3}$ | $2.66 \cdot 10^{-3}$ | $5.22 \cdot 10^{-4}$ | $5.13 \cdot 10^{-4}$ |
| SFBC | $7.08 \cdot 10^{-3}$ | $\mathbf{1.61 \cdot 10^{-8}}$ | $\mathbf{1.61 \cdot 10^{-8}}$ | $\mathbf{8.56 \cdot 10^{-8}}$ | $\mathbf{8.56 \cdot 10^{-8}}$ | $\mathbf{4.00 \cdot 10^{-7}}$ | $\mathbf{4.00 \cdot 10^{-7}}$ | $\mathbf{7.15 \cdot 10^{-7}}$ | $\mathbf{1.30 \cdot 10^{-5}}$ | $8.06 \cdot 10^{-3}$ |
| Fourier (even) | $7.08 \cdot 10^{-3}$ | $\mathbf{1.61 \cdot 10^{-8}}$ | $8.67 \cdot 10^{-8}$ | $3.16 \cdot 10^{-7}$ | $9.40 \cdot 10^{-7}$ | $2.81 \cdot 10^{-6}$ | $3.93 \cdot 10^{-6}$ | $9.55 \cdot 10^{-6}$ | $1.80 \cdot 10^{-5}$ | $\mathbf{6.20 \cdot 10^{-5}}$ |
| Fourier (odd) | $\mathbf{7.07 \cdot 10^{-3}}$ | $7.07 \cdot 10^{-3}$ | $7.07 \cdot 10^{-3}$ | $7.07 \cdot 10^{-3}$ | $7.07 \cdot 10^{-3}$ | $7.07 \cdot 10^{-3}$ | $7.07 \cdot 10^{-3}$ | $7.08 \cdot 10^{-3}$ | $7.07 \cdot 10^{-3}$ | $7.07 \cdot 10^{-3}$ |
| Chebyshev | $7.08 \cdot 10^{-3}$ | $7.07 \cdot 10^{-3}$ | $3.65 \cdot 10^{-5}$ | $3.64 \cdot 10^{-5}$ | $2.46 \cdot 10^{-5}$ | $2.43 \cdot 10^{-5}$ | $1.09 \cdot 10^{-4}$ | $1.00 \cdot 10^{-4}$ | $2.26 \cdot 10^{-4}$ | $6.90 \cdot 10^{-5}$ |

Table 3: This table shows the results of an ablation study to find the optimal number of basis terms for learning a purely antisymmetric SPH gradient interpolation. The results were obtained as the $L_2$ loss for a simple SPH gradient interpolation evaluated on the test dataset of test case I at 4 different timepoints for 4 different network initialization seeds. Each column represents a different number of basis terms, with the best behavior for each number of basis terms highlighted in bold.

| n Basis | 1 | 2 | 3 | 4 | 5 | 6 | 7 | 8 | 16 | 32 |
|---|---|---|---|---|---|---|---|---|---|---|
| LinCConv | $1.00 \cdot 10^{-1}$ | $1.47 \cdot 10^{-1}$ | $1.51 \cdot 10^{-1}$ | $1.24 \cdot 10^{-1}$ | $1.30 \cdot 10^{-1}$ | $1.19 \cdot 10^{-1}$ | $1.32 \cdot 10^{-1}$ | $1.23 \cdot 10^{-1}$ | $1.14 \cdot 10^{-1}$ | $9.30 \cdot 10^{-2}$ |
| DMCF | $\mathbf{9.82 \cdot 10^{-2}}$ | $\mathbf{2.07 \cdot 10^{-3}}$ | $\mathbf{8.36 \cdot 10^{-4}}$ | $9.02 \cdot 10^{-4}$ | $5.28 \cdot 10^{-4}$ | $\mathbf{2.34 \cdot 10^{-4}}$ | $\mathbf{2.42 \cdot 10^{-4}}$ | $7.51 \cdot 10^{-4}$ | $1.86 \cdot 10^{-3}$ | $3.50 \cdot 10^{-4}$ |
| SplineConv | $1.00 \cdot 10^{-1}$ | $1.00 \cdot 10^{-1}$ | $1.06 \cdot 10^{-1}$ | $1.20 \cdot 10^{-1}$ | $1.16 \cdot 10^{-1}$ | $1.07 \cdot 10^{-1}$ | $1.04 \cdot 10^{-1}$ | $1.07 \cdot 10^{-1}$ | $9.73 \cdot 10^{-2}$ | $9.17 \cdot 10^{-2}$ |
| SFBC | $1.00 \cdot 10^{-1}$ | $4.13 \cdot 10^{-3}$ | $4.13 \cdot 10^{-3}$ | $1.42 \cdot 10^{-3}$ | $1.48 \cdot 10^{-3}$ | $1.50 \cdot 10^{-4}$ | $4.37 \cdot 10^{-4}$ | $4.63 \cdot 10^{-4}$ | $4.99 \cdot 10^{-4}$ | $6.63 \cdot 10^{-3}$ |
| Fourier (even) | $1.00 \cdot 10^{-1}$ | $1.00 \cdot 10^{-1}$ | $1.00 \cdot 10^{-1}$ | $1.00 \cdot 10^{-1}$ | $1.01 \cdot 10^{-1}$ | $1.01 \cdot 10^{-1}$ | $1.01 \cdot 10^{-1}$ | $1.01 \cdot 10^{-1}$ | $1.01 \cdot 10^{-1}$ | $1.14 \cdot 10^{-1}$ |
| Fourier (odd) | $1.00 \cdot 10^{-1}$ | $4.13 \cdot 10^{-3}$ | $1.44 \cdot 10^{-3}$ | $\mathbf{1.09 \cdot 10^{-4}}$ | $\mathbf{1.38 \cdot 10^{-4}}$ | $2.38 \cdot 10^{-4}$ | $2.88 \cdot 10^{-4}$ | $\mathbf{3.15 \cdot 10^{-4}}$ | $\mathbf{2.57 \cdot 10^{-4}}$ | $\mathbf{2.06 \cdot 10^{-4}}$ |
| Chebyshev | $1.00 \cdot 10^{-1}$ | $9.89 \cdot 10^{-2}$ | $9.89 \cdot 10^{-2}$ | $9.53 \cdot 10^{-3}$ | $2.02 \cdot 10^{-1}$ | $1.93 \cdot 10^{-1}$ | $3.11 \cdot 10^{-2}$ | $2.81 \cdot 10^{-2}$ | $5.40 \cdot 10^{-2}$ | $8.66 \cdot 10^{-2}$ |

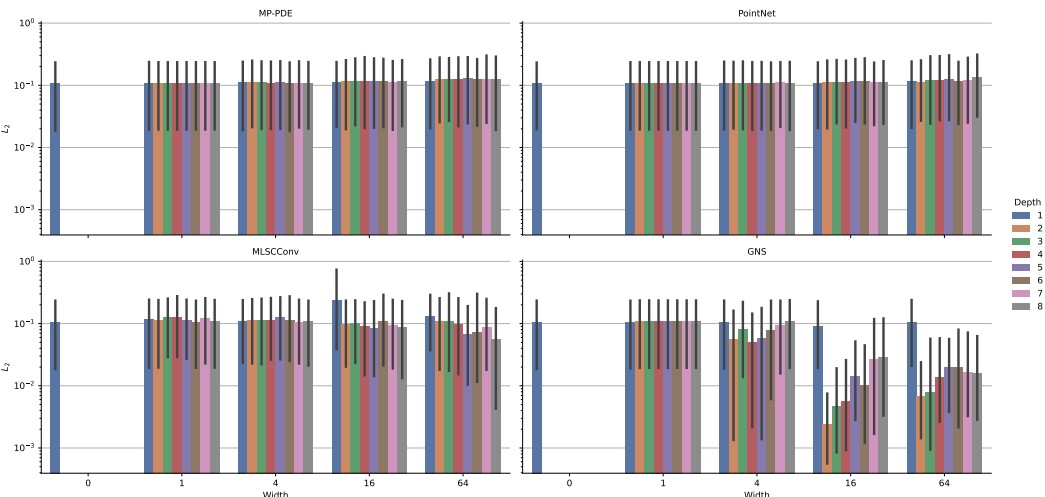

Figure 20: This figure evaluates the correlation between the hidden layer architecture of MLP-based graph networks and the ability of a single-layer message-passing network to learn an SPH kernel gradient for various basis functions. The x-axes correspond to the number of features per hidden layer, and the y-axis corresponds to the $L_2$ error evaluated using the test dataset from test case I for 4 different frames $t = [0, 0.512, 1.024, 2.048]$, with color indicating the number of hidden layers. The plots show the behavior of the Message Passing PDE (top left), PointNet (top right), MLPCConv (bottom left), and GNS (bottom right) network architectures.

### C.1.2 SPH KERNEL GRADIENT INTERPOLATION

In addition to kernel interpolations, gradient interpolations are another vital aspect of SPH simulation and, accordingly, if a method is not able to learn these terms on their own, then the method might not be well suited to learn an overall SPH simulation as, e.g., physical forces generally rely on such gradient terms. Overall, the setup is equivalent to the first toy problem.

**Baseline Methods**: Compared to the first toy problem, the results here, see Fig. 2 and Table 3, are notably different. On the one hand, for the kernel interpolation, all baseline methods were able to learn something; in this case, both *LinCConv* and *SplineConv* were not able to learn any reasonable approximation of the gradient interpolation. On the other hand, *DMCF*, which was unable to learn anything in the prior task, can perform as well as the Fourier-based approaches due to the inherent antisymmetry of this basis. These results indicate that learning to represent an antisymmetric interaction is very challenging without an inductive bias to help the network. Note that the performance of *DMCF* did not improve with very high numbers of base terms, i.e., the best performance was achieved with 6 terms, indicating that smoothness in this problem is not as necessary as for the first toy problem.

**Chebyshev Basis**: While this basis performed well for the first toy problem, it cannot learn anything meaningful in this task, even though this method exhibits inherent symmetries. However, it is essential to note that the fourth term is antisymmetric and clearly performs better than using one more or fewer term, which still indicates that including antisymmetry is important.

**Fourier Methods**: In this toy problem, the Fourier-based methods still performed better than all other compared results; however, the difference between the Fourier methods and other methods is not nearly as significant. A crucial observation is that some behaviors observed for the previous toy problem, especially the stepped change in performance, are still exhibited here but in an inverse relationship relative to before. As such, we can clearly observe that the Fourier basis improves notably as higher harmonics are included and that the performance peaks after including the third harmonic terms. Another important observation is that the oddly symmetric Fourier basis does not degrade in performance as notably as the complete Fourier basis.

**MLP-based Methods**: While for the previous toy problem *GNS*, *MP-PDE* and *MLPCConv* were able to learn the task, in this problem only *MP-PDE* shows promising results. This clearly highlights

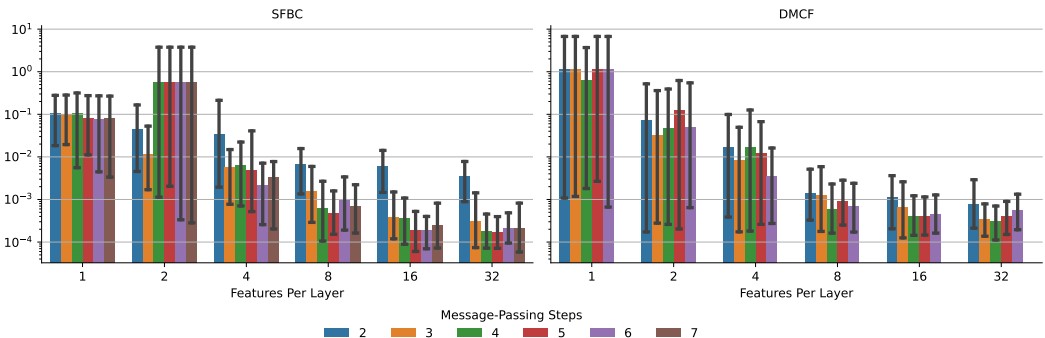

Figure 21: This figure shows the relationship of features per layer (x-axis) and message passing steps (color coded) to the ability to learn a single-step physics update. The $L_2$ error is evaluated using the test dataset from test case I for $4$ different frames $t = [0, 0.512, 1.024, 2.048]$ and across $4$ random network initializations, errors bars indicating the lower 5-th to upper 95-th percentile. The *DMCF* approach is shown on the left, and our proposed Fourier basis approach is shown on the right.

that this problem is much more challenging to learn, especially without the inclusion of several inductive biases. While *MLPCConv* did not perform well, it exhibited some learning behavior for larger architectures. Overall, only *GNS* was able to perform well as the inclusion of several inductive biases, i.e., providing physical quantities to all MLP operations, did not aid the *MP-PDE* approach but rather added further additional terms that made the task more challenigng to learn. This highlights an important avenue of future research for basis convolutions as the usage of vertex MLPs with physical quantities as additional inputs might also be helpful in those architectures; however, this is beyond the scope of our current research.

## C.2 Test-Case I

Learning the physics update, i.e., training an NNTI, is significantly more challenging than learning a simple SPH kernel or kernel gradient. The underlying simulation also requires multiple steps, e.g., to compute particle density and pressure forces, and, accordingly, we utilize a multiple message-passing architecture here. We are now interested in how different methods perform, especially with regard to network size, and if there is a clear and significant difference between methods that incorporate symmetry and those that do not.

**Network Scaling**: Regarding network scaling, we varied the number of message-passing steps and the number of features per layer; see Figure 3. On the one hand, the results indicate that the number of message-passing steps, once the number of steps is at least 3, does not significantly impact performance. However, for some feature per layer sizes, there is still a trend, e.g., for $4$ features per layer. On the other hand, the number of features per layer has a much more pronounced impact on the network's performance and appears to be the driving force of the network performance. Overall, we still see the expected behavior insofar as larger networks perform better than smaller ones. Accordingly, we focus the further discussions on a network with $6$ message-passing steps and $32$ features per layer. We also investigated the number of basis terms and found that, like the antisymmetric case, $6$ terms perform optimally.

**Baseline Methods**: Compared to the toy problems, the only baseline method that performs reasonably well is *DMCF*, which is also the only baseline method that performed well for the antisymmetric case and the only baseline method that includes antisymmetric terms. This result supports our prior observations, i.e., that antisymmetry is an essential property for convolutions and that most baseline methods struggle with learning this aspect.

**Chebyshev Basis**: The Chebyshev-based convolution, in this case, performs worse than all other methods and, notably, performs worse with larger networks. This generally poor performance is primarily due to the difficulty in training these networks, i.e., the initialization significantly influences the final network performance, and most initialization does not lead to a well-converging network. Accordingly, further investigations of the initialization of these methods are an important part of

Table 4: This figure shows the quantitative results for an ablation study of basis terms in test case II. All entries are computed by initializing 4 networks for each case at all 4 testing samples from the dataset at 4 different time points and then performing 64 inference steps while computing the density, velocity, divergence, and PSD errors on a grid resampling of the particle quantities and finally computing the mean across the inference period. Bold indicates the lowest value per row.

| Configuration | Window: Müller | | | | Window: None | | | |
| Basis | Dens. ↓ | Vel. ↓ | Div. ↓ | PSD ↓ | Dens. ↓ | Vel. ↓ | Div. ↓ | PSD ↓ |
|---|---|---|---|---|---|---|---|---|
| Nearest Neighbor | 0.040 | 0.126 | 2.31 | 56 | 0.170 | 0.261 | 5.13 | 148 |
| LinCCConv | 0.041 | 0.130 | 3.44 | 45 | 0.089 | 0.147 | 2.32 | 80 |
| DMCF | 0.052 | 0.124 | 2.75 | 52 | 0.422 | 0.496 | 8.62 | 285 |
| SplineConv | 0.043 | 0.129 | 2.47 | 55 | 0.258 | 0.285 | 2.41 | 152 |
| SFBC | **0.031** | **0.106** | 2.46 | **34** | **0.035** | **0.098** | **1.29** | **41** |
| Chebyshev | 0.051 | 0.135 | 3.23 | 73 | 0.468 | 0.332 | 4.20 | 123 |
| Chebyshev (2nd) | 0.058 | 0.142 | 3.48 | 75 | 0.418 | 0.296 | 3.55 | 106 |
| Gaussian Kernel | 0.312 | 0.377 | 5.81 | 203 | 0.491 | 0.326 | 3.60 | 104 |
| Müller Kernel | 0.683 | 0.637 | 5.81 | 354 | 0.493 | 0.344 | 3.96 | 126 |
| Wendland-2 | 0.057 | 0.133 | 2.53 | 62 | 0.489 | 0.329 | 3.71 | 106 |
| Quartic Spline | 0.045 | 0.151 | 2.20 | 82 | 0.397 | 0.417 | 3.99 | 259 |
| Spiky | 0.113 | 0.165 | **1.49** | 74 | 0.170 | 0.199 | 2.42 | 111 |
| Bump RBF | 0.480 | 0.433 | 2.49 | 125 | 0.530 | 0.432 | 3.48 | 170 |

future research, especially for physical simulations. Note that these observations align with prior observations in pattern recognition, e.g., by He et al. (2022).

**Fourier Methods**: In this scenario, the complete Fourier Series approach performed better than all other methods and outperforms *DMCF* across a broad range of network sizes, with a more significant gap for smaller network sizes. Interestingly, the oddly symmetrical Fourier Series performs better than all methods except for *DMCF*, but still significantly worse than the complete Fourier Series. This indicates that simply using an antisymmetric basis may work for some basis functions, including symmetric terms, which can significantly improve the overall network performance.

**MLP-based**: Finally, for the MLP-based GNNs, we saw performance comparable to most baselines, e.g., *LinCCConv*, but notably worse performance than the antisymmetric basis convolutions. This, again, indicates a significant boost to the learning capabilities of a network by including useful inductive biases at the core of the method. Furthermore, these networks required significantly more parameters for these networks, as discussed in Section 4.2. An important final note is the performance of *PointNet*. While *PointNet* did not perform well for either toy problem, it performs surprisingly well for this problem and even outperforms *MLPCConv*, even though it does not consider graph connectivity explicitly. This indicates that while this problem is challenging, basing a guess for the new particle velocity on the current position and velocity, due to the smooth nature of the simulation, can already give reasonable performance.

## C.3 TEST-CASE II

While the main paper already discussed this test case in-depth, the results shown here provide a more quantifiable foundation for our conclusions. In this section, we will discuss all the ablation studies and provide numerical results in tables and bar charts, as well as visualizations of the inference behavior. The latter serves as a helpful visualization of the difference of the basis functions on top of the quantifiable metrics. Finally, we will expand on some evaluations, e.g., we include a separate section regarding network initialization stability and how different basis functions perform when evaluated using different time points in the test simulation to verify that our results are representative.

### C.3.1 BASIS FUNCTION EVALUATION

Basis functions are an essential part of our proposed basis convolution methodology, and accordingly, evaluating a broad set of potential basis functions is a crucial part of the evaluation. As our

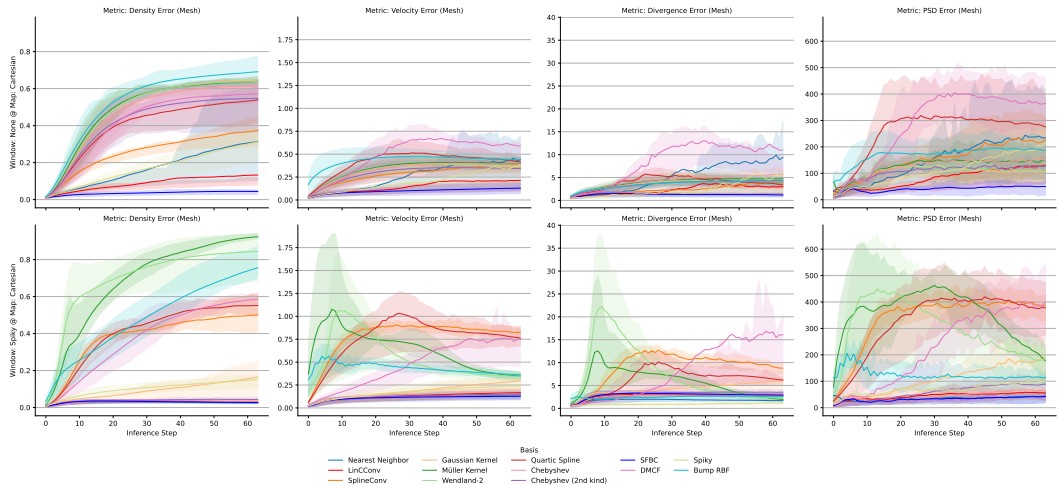

Figure 22: This figure shows the inference stability for an ablation study of basis terms in test case II. The lines are evaluated by initializing the network at all 4 testing samples from the dataset at 4 different time points and then performing 64 inference steps while computing the (f.l.t.r.) density, velocity, divergence, and PSD errors on a grid resampling of the particle quantities, with error bars indicating lower 5-th to upper 95-th percentile. Note that 4 network initialization seeds are used here as well. The color here indicates different basis terms, with the top row showing behavior for using no window function and the bottom row showing the results when using the *Spiky* window function.

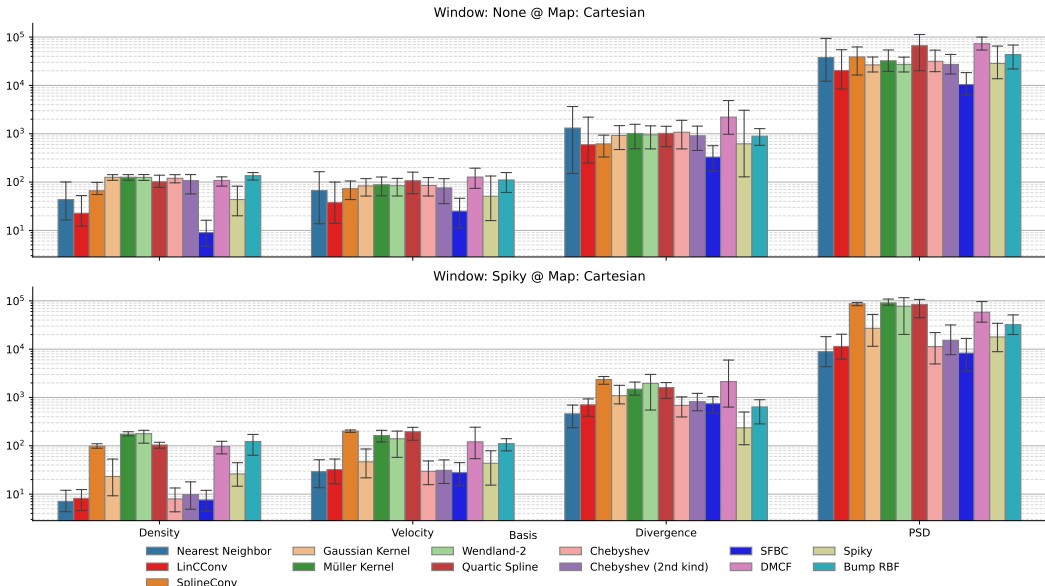

Figure 23: This figure shows the quantitative results for an ablation study of basis terms in test case II. All entries are computed by initializing 4 networks for each case at all 4 testing samples from the dataset at 4 different time points and then performing 64 inference steps while computing the (f.l.t.r.) density, velocity, divergence, and PSD errors on a grid resampling of the particle quantities and finally computing the behavior across the inference period. The color indicates the basis term, and error bars indicate the lower 5-th to upper 95-th percentile. The top row indicates the results of not using a window function, and the bottom row indicates the results of using the *Spiky* window.

formulation is generic and can be used to implement a broad range of methods, we focus on the most intuitively interesting methods that represent several classes of methods, see Appendix A.4. The first group of methods is piece-wise bases, which include *LinCConv*, an important prior baseline to compare against, and a naïve *Nearest Neighbor* based approach as a useful baseline. The second group of methods is Gaussian bases consisting of *SplineConv*, using a cubic B-Spline basis, a quartic B-spline basis, the polynomial Müller Kernel, the SPH-based Wendland-2 Kernel, and a Gaussian, where we chose the methods to represent a broad set of choices common in SPH literature. The third group of methods are generic Radial Basis Functions, which include the *Bump* RBF and the *Spiky* Kernel, neither of which is Gaussian-shaped. The fourth and final group are bases with inherent symmetries and include our *Fourier* basis, Chebyshev bases of first and second kind, and the *DMCF* basis. For our evaluations here we consider the inference behavior, see Fig. 22, as well as the average performance over an inference length of 64 steps both numerically, see Table 4, and regarding their distributions, see Fig. 23.

**Piece-Wise Bases**: Regarding piece-wise bases, we see a notable difference between nearest neighbor and linear interpolation; however, this difference is heavily reliant on the choice of window function. When not using a window function, linear interpolation exhibits a lower error in all metrics compared to nearest neighbor, which is strongly influenced by the nearest neighbor basis becoming unstable for some experiments. However, when using the *Müller* window function, the nearest neighbor basis exhibits a lower error than the linear basis for all metrics besides the PSD error. Notably, all errors for the nearest neighbor interpolation with a window function are below the values for the linear basis without a window function. Overall, this indicates that nearest neighbor interpolation is useful due to its simplicity but relies heavily on external biases to perform stably. In contrast, linear interpolation performs reasonably well even without a window function.

**Gaussian Bases**: Regarding these basis functions, the most crucial observation is that none of these methods work without a window function, i.e., they all exhibit density errors that are an order of magnitude worse than the best method. However, some of these methods perform comparably well without a window function, while others do not. Interestingly, both B-Spline bases and the very similar Wendland-2 basis perform well, while the polynomial *Müller* basis and Gaussian kernels do not. This difference indicates that similar to SPH literature (Dehnen & Aly, 2012), there are significant differences between kernel functions that might not be apparent from their shapes, i.e., the *Müller* basis is very similar in shape to the cubic B-Spline but works notably worse. Furthermore, these results indicate that the compactness of the basis function is still a valuable property, as the non-compact Gaussian basis does not work well.

**Generic RBFs**: Regarding generic basis functions, we did not see strong behavior in any case, which indicates that these functions are not ideal bases for our formulation. However, this is hardly surprising as RBF interpolation does not solely adjust the weights for each basis function but instead shifts the centroids and modifies the shapes of the basis terms, neither of which we include in our network. These additions pose an interesting direction for future work; however, optimizing them as part of a neural network may be challenging as these terms are non-linear.

**Symmetric Bases**: For symmetric without a window function, we find that only our proposed Fourier-based approach remains stable while all other options quickly become unstable. This is not very surprising as, on the one hand, the Chebyshev basis is not compact, and, as seen for the Gaussian basis, compactness appears to be a useful property. On the other hand, the Fourier basis contains terms that decay towards zero at the outside edges. Furthermore, with a window function, all of these bases become stable and show behavior comparable to all other methods, with our proposed Fourier-based network performing better than all other bases. Finally, while the divergence error only increases for the Fourier-based method, all other symmetric bases exhibit a lower error; however, do note that none of the other bases were stable without a window function.

**Conclusions**: Overall, we could clearly observe behavior that is consistent for different classes of functions where our proposed network architecture outperforms all other bases in all but a single metric. Moreover, these evaluations indicate that other base terms, such as nearest neighbor interpolation, can still be useful even though they appear less capable than more complex bases, such as cubic B-Splines. Finally, regarding the inference behavior, we observed that methods were either stable, i.e., they remained at a low error the entire inference period or steadily got worse over time with no method randomly becoming unstable late into the inference period.

Table 5: This figure shows the quantitative results for an ablation study of window functions in test case II. All entries are computed by initializing 4 networks for each case at all 4 testing samples from the dataset at 4 different time points and then performing 64 inference steps while computing the density, velocity, divergence, and PSD errors on a grid resampling of the particle quantities and finally computing the mean across the inference period. Bold indicates the lowest value per row.

| Basis

window | LinCConv | | | | Ours | | | |
|---|---|---|---|---|---|---|---|---|
| | Dens. ↓ | Vel. ↓ | Div. ↓ | PSD ↓ | Dens. ↓ | Vel. ↓ | Div. ↓ | PSD ↓ |
| None | 0.080 | 0.137 | **2.11** | 67 | 0.038 | 0.103 | **1.24** | 44 |
| Müller | 0.041 | 0.130 | 3.44 | 45 | 0.031 | 0.106 | 2.46 | 34 |
| Spiky Kernel | 0.056 | 0.180 | 4.81 | 87 | 0.031 | 0.113 | 3.06 | **29** |
| Cubic Spline | 0.038 | 0.135 | 3.11 | 54 | 0.032 | 0.113 | 3.11 | 31 |
| Quartic Spline | 0.044 | 0.150 | 3.53 | 63 | 0.034 | 0.117 | 3.36 | 35 |
| $1 - x$ | **0.037** | **0.119** | 2.31 | **43** | 0.026 | 0.097 | 2.04 | 32 |
| $1 - x^2$ | 0.063 | 0.147 | 3.13 | 67 | **0.025** | **0.095** | 1.87 | 31 |

### C.3.2   ABLATION STUDY: WINDOW FUNCTIONS

Window functions are a strong inductive bias in CConv approaches built on the ideas of SPH. These window functions aim to (a) ensure compactness, (b) ensure smoothness, and (c) as a bias towards the underlying simulation. Compactness is, generally, ensured by defining window functions such that they are equal to 0 for $|q| = 1$, e.g., $f(q) = 1 - |q|$, smoothness is, generally, enforced by using higher order polynomials, e.g.,., cubic B-splines, and the bias is achieved by using SPH kernels as window functions. Accordingly, it makes sense to evaluate the strength of these individual motivations by evaluating a set of basis functions that exhibit varying levels of smoothness and similarity to the underlying SPH simulation, which, for this problem, is the cubic B-Spline kernel.

**LinCConv**: For this approach, we observe that the best-performing window function, overall, is a simple linear window, which only ensures compactness but does not impose any further restrictions on smoothness. The second best kernel function is the cubic B-Spline kernel, indicating that the third motivation, e.g., similarity to the underlying SPH kernel, is also an important effect. Still, this additional bias might introduce other effects that are not desirable. Finally, the other kernel functions, especially the smoother quartic B-Spline kernel, perform worse than the cubic B-Spline. A crucial observation here, however, is that the best method regarding the divergence error is not using any window function, which clearly indicates that using a window function is not a universally beneficial choice and, instead, opens up the ability to fine-tune the network based on the task.

**Fourier**: Contrary to the *LinCConv* approach, the results for our proposed basis are much closer to each other, e.g., the deviation for the density is $\pm 0.06$ for our approach while for *LinCConv* the spread is $\pm 0.21$, indicating that the method is inherently more capable as it outperforms the *LinCConv* base regarding all metrics with all basis functions. Furthermore, barring the divergence error, the worst choice of window function for each metric for our proposed basis still performs as well, or better, than *LinCConv* with the respective best choice. For our proposed basis, we also observe that the linear window performs very close to the best window function regarding all metrics except for divergence, while not using a window function still results in the lowest divergence.

Based on the results observed here, we conclude that not using a window function is an important choice to consider, as not using a window function leads to the lowest divergence error in all cases; see also Appendix C.3.1. Furthermore, simple choices such as a linear window can outperform more complex and *intuitive* window functions when choosing an actual basis function. In contrast, some SPH kernels, such as the *Spiky* window, do not perform as well as others.

### C.3.3   ABLATION STUDY: COORDINATE MAPPINGS

Coordinate systems play an essential role in many systems and can encode useful information through their definitions. For example, SPH is built on rotationally symmetric kernel functions, i.e., functions that solely depend on the distance and not the angle of two particles. Accordingly, using polar coordinates is useful in defining SPH interactions, and using the same coordinate sys-

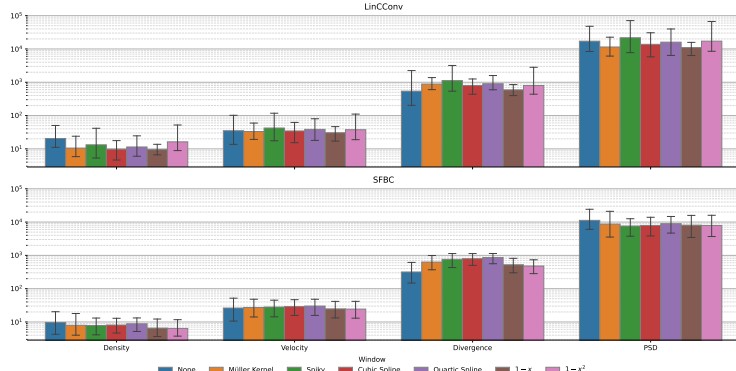

Figure 24: This figure shows the quantitative results for an ablation study of window functions in test case II. All entries are computed by initializing 4 networks for each case at all 4 testing samples from the dataset at 4 different time points and then performing 64 inference steps while computing the (f.l.t.r.) density, velocity, divergence, and PSD errors on a grid resampling of the particle quantities and finally computing the behavior across the inference period. The color indicates window function, and error bars indicate lower 5-th to upper 95-th percentile. The top row indicates results for the *LinCConv* basis, and the bottom row indicates results of using our Fourier basis.

Table 6: This figure shows the quantitative results for an ablation study of coordinate mappings in test case II. All entries are computed by initializing 4 networks for each case at all 4 testing samples from the dataset at 4 different time points and then performing 64 inference steps while computing the density, velocity, divergence, and PSD errors on a grid resampling of the particle quantities and finally computing the mean across the inference period. Bold indicates the lowest value per row.

| Map | Basis | Window: None | | | | Window:Spiky | | | |
| | | Dens. ↓ | Vel. ↓ | Div. ↓ | PSD ↓ | Dens. ↓ | Vel. ↓ | Div. ↓ | PSD ↓ |
| --- | --- | --- | --- | --- | --- | --- | --- | --- | --- |
| Cart. | LinCConv | 0.099 | 0.159 | 2.88 | 85 | 0.035 | 0.129 | 3.16 | 49 |
| | NN | 0.166 | 0.291 | 6.78 | 150 | 0.026 | 0.107 | **1.49** | 31 |
| | SFBC | 0.033 | 0.097 | 1.17 | **35** | 0.033 | 0.114 | 3.19 | **29** |
| Pol. | LinCConv | **0.028** | **0.095** | 1.91 | 28 | 0.026 | **0.105** | 2.33 | 30 |
| | NN | 0.125 | 0.206 | 3.81 | 115 | **0.025** | 0.108 | 1.50 | 33 |
| | SFBC | 0.059 | 0.162 | **0.99** | 62 | 0.051 | 0.172 | 1.63 | 49 |
| Pres. | LinCConv | 0.281 | 0.312 | 6.36 | 168 | 0.043 | 0.138 | 3.13 | 64 |
| | NN | 0.172 | 0.257 | 5.14 | 120 | 0.027 | 0.110 | 1.64 | 32 |
| | SFBC | 0.036 | 0.099 | 1.35 | 44 | 0.034 | 0.119 | 3.31 | 36 |

tem for a neural network that is to learn an SPH simulator would also make sense. Furthermore, coordinate systems can be used to make the use of coefficients more efficient, as proposed by Ummenhofer et al. (2019), as using a regular grid in higher dimensions is not ideal for spherically limited interactions as some weights will have reduced, or even no, influence. However, as we focus our evaluations on two-dimensional systems, there is merit in re-evaluating these biases on our specific problems. Accordingly, we evaluate the influence of choosing no coordinate mapping, polar coordinates or a volume-preserving mapping for the *LinCConv* approach, our proposed approach and a nearest neighbor interpolation approach as a baseline for no window function and the *Spiky* window function, see Fig. 25 and Table 6.

**LinCConv**: For the *LinCConv* approach, we saw the best results without a window function for the polar coordinate mapping and the worst outcomes for the volume preserving mapping, while when using a window function, the results are only marginally different. These results indicate a clear benefit to changing the coordinate mapping in some cases, i.e., without a window function. Still, when using a window function, the differences are negligible. While this is somewhat contrary to

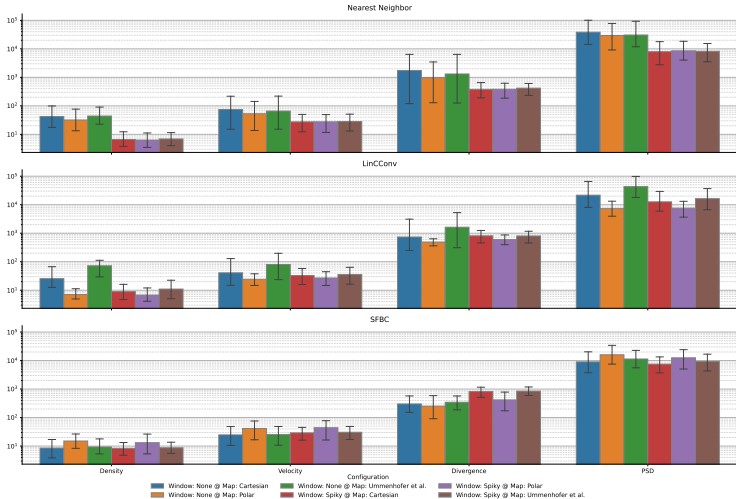

Figure 25: This figure shows the quantitative results for an ablation study of coordinate mappings in test case II. All entries are computed by initializing 4 networks for each case at all 4 testing samples from the dataset at 4 different time points and then performing 64 inference steps while computing the (f.l.t.r.) density, velocity, divergence, and PSD errors on a grid resampling of the particle quantities and finally computing the behavior across the inference period. Color here indicates coordinate mapping and window function combinations and error bars indicating lower 5-th to upper 95-th percentile. From top to bottom, the subplots represent using a Nearest Neighbor Basis, *LinCConv* and *SFBC*.

Table 7: This figure shows the quantitative results for an ablation study of Fourier terms in test case II. All entries are computed by initializing 4 networks for each case at all 4 testing samples from the dataset at 4 different time points and then performing 64 inference steps while computing the density, velocity, divergence, and PSD errors on a grid resampling of the particle quantities and finally computing the mean across the inference period. Bold indicates the lowest value per row.

| Configuration | Window: None | | | | Window: Spiky | | | |
| --- | --- | --- | --- | --- | --- | --- | --- | --- |
| | Density | Vel. | Div. | PSD | Density | Vel. | Div. | PSD |
| Basis | | | | | | | | |
| SFBC | 0.046 | **0.118** | **1.65** | 57 | 0.029 | 0.109 | 2.90 | 28 |
| Fourier (4-Terms) | 0.072 | 0.126 | 2.33 | 67 | **0.026** | **0.105** | **2.46** | **27** |
| Fourier (5-Terms) | **0.041** | 0.125 | 2.56 | **55** | 0.028 | 0.106 | 2.64 | 30 |
| Fourier (even) | 0.373 | 0.369 | 5.50 | 212 | 0.308 | 0.336 | 4.97 | 177 |
| Fourier (odd) | 0.095 | 0.188 | 3.81 | 87 | 0.032 | 0.113 | 3.10 | 37 |
| Fourier (odd) + x | 0.829 | 0.523 | 3.87 | NaN | 0.328 | 0.278 | 2.96 | 101 |
| Fourier (odd) + sgn(x) | 0.896 | 0.561 | 5.46 | NaN | 0.431 | 0.306 | 3.39 | 104 |

prior work, the volume-preserving mapping is mostly intended for three-dimensional applications and not two-dimensional tasks.

**Fourier Basis**: Similar to observations regarding the window function, see Appendix C.3.2, there is only a very small difference between different choices of coordinate mappings. This indicates that (a) the influence of different hyperparameters is very low for our proposed basis, and (b) ideal performance can often be achieved while removing inductive biases from the network.

Overall, as we saw very little difference with a window function and for our method in general, we propose to remove this inductive bias in most cases from network architectures but still include it as it can be helpful in corner cases, e.g., without window functions for *LinCConv*.

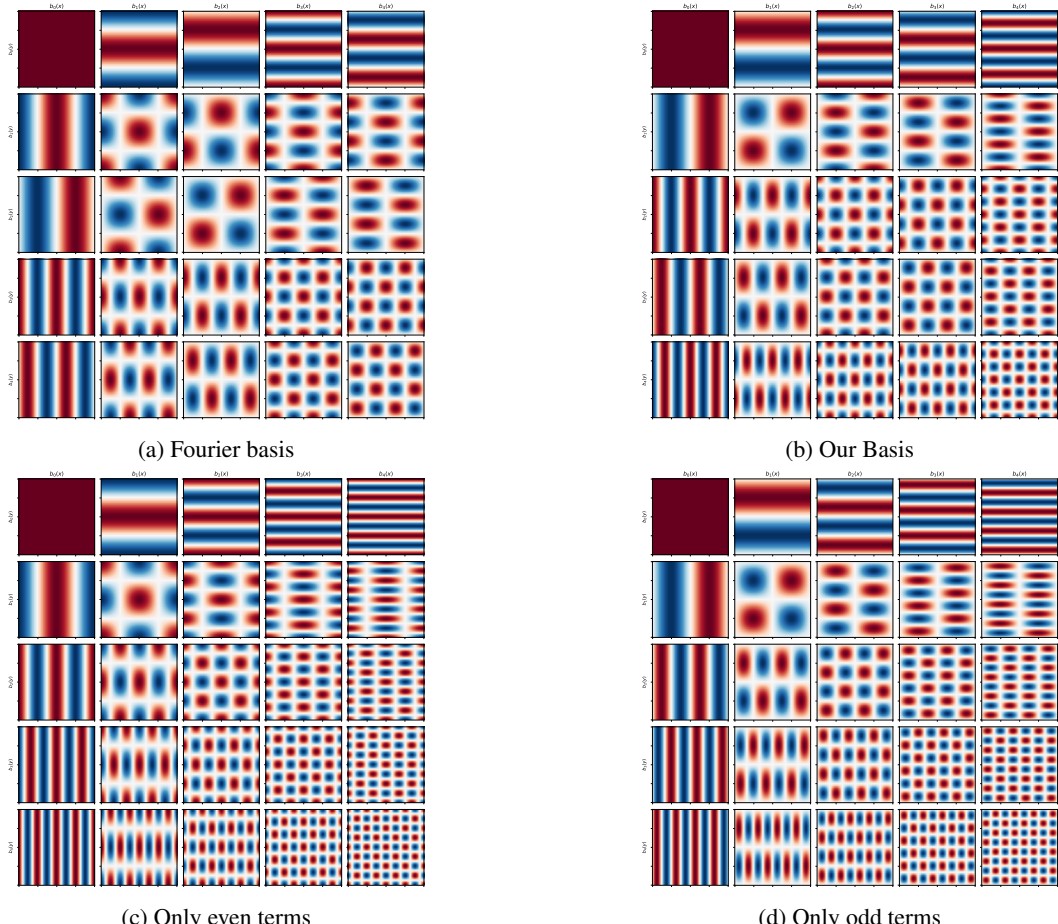

$$
\begin{array}{cc}
\text{(a) Fourier basis} & \text{(b) Our Basis} \\
\text{(c) Only even terms} & \text{(d) Only odd terms}
\end{array}
$$

Figure 26: Visualization of some Fourier basis setups we evaluated as part of our ablation studies.

### C.3.4 ABLATION STUDY: FOURIER TERMS

Fourier Series have a long and rich history of over 200 years of research, but at the core, a Fourier series is a sequence of alternative sine and cosine terms of increasing frequencies with an additional constant function added on top. As such, a Fourier series can be defined as

$$
b_i^{\text{Fourier}}(p) = \begin{cases}
1, & i = 0, \\
\frac{1}{\sqrt{\pi}} \cos\left(\pi \left[\lfloor \frac{i-1}{2} \rfloor + 1\right] p\right), & i \text{ odd}, \\
\frac{1}{\sqrt{\pi}} \sin\left(\pi \left[\lfloor \frac{i-1}{2} \rfloor + 1\right] p\right), & i \text{ even}.
\end{cases}
\tag{64}
$$

However, while for an infinite Fourier series, equally many cosine and sine terms exist, this is not necessarily true for a finite sequence of terms of a Fourier Series. For an odd number $n$ of terms the constant term and $(n-1)/2$ harmonics of a sine and cosine term each can be utilized. However, a common choice in baseline methods, e.g., *LinCConv* (Ummenhofer et al., 2019), is using only four terms, which is even and, accordingly, some term needs to be excluded. As representative choices, we investigate either excluding the second harmonic sine term (an odd symmetry term) or the first harmonic cosine term (an even symmetry term). Furthermore, we investigate four more series that consists of (a) only even symmetry terms, (b) only odd symmetry terms with an additional constant term, (c) only odd symmetry terms with the constant term replaced by $p$, and (d) only odd symmetry terms with the constant term replaced by $\text{sgn}(p)$. Note that many more choices exist, e.g., an odd symmetry series with no offset term at all, but we chose to limit ourselves to this set of 8 series.

**Complete Series**: Considering the definition from Eqn. 64, we evaluate this series with 4 and 5 terms, see Fig. 26a, where the 4 term variant drops the second harmonic sine term. As such, we

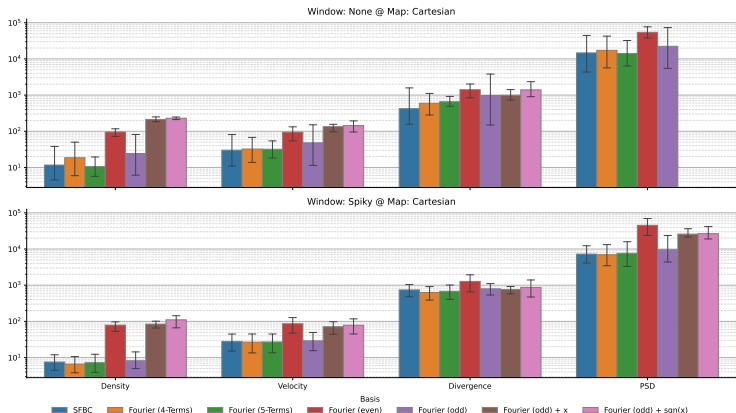

Figure 27: This figure shows the quantitative results for an ablation study of Fourier terms in test case II. All entries are computed by initializing 4 networks for each case at all 4 testing samples from the dataset at 4 different time points and then performing 64 inference steps while computing the (f.l.t.r.) density, velocity, divergence, and PSD errors on a grid resampling of the particle quantities and finally computing the behavior across the inference period. The color indicates Fourier term combinations and error bars indicate lower 5-th to upper 95-th percentile. The top row indicates the results of not using a window function, and the bottom row indicates the results of using the *Spiky* window function.

would expect the 5 term variant to perform notably better regarding symmetric quantities, such as density error, but comparable for antisymmetric quantities. Considering the results in Table 7 and Fig. 27, we observed the expected behavior when not using a window function, whereas without a window function, the results are much closer to another.

**Ours**: As antisymmetry is an essential aspect of physical systems, including antisymmetric terms should improve the performance, and their behavior should dominate the overall learned behavior. Accordingly, removing the first harmonic even symmetry term and trading this off with a second harmonic symmetric term is an interesting choice; see Fig. 26a. For a four-term basis, the two-dimensional series consists of 8 symmetric and 8 antisymmetric terms, whereas the alternative choice consists of 10 symmetric and 6 antisymmetric terms. Using this series, we observe a significantly improved behavior regarding all quantities without a window function, whereas without a window function, this choice of terms performs worse in all quantities. A crucial observation here is that the divergence error of this formulation increases by 75% when using a window function, whereas the divergence error only increases by 6% for the alternate formulation. However, as this formulation leads to the lowest overall divergence and better behavior without a window function, we used this function as the basis for our approach.

**Symmetric Series**: Using only even terms, i.e., the constant term and cosine terms of increasing harmonics, see Fig. 26c leads to 16 symmetric and 0 antisymmetric terms, whereas using the constant term and sine terms of increasing harmonics, see Fig. 26d leads to 10 symmetric and 6 antisymmetric terms. Note that with our separable basis formulation, it is not easily possible to only construct antisymmetric terms. Regarding the even variant and the modified odd variants, we saw no stable behavior regardless of whether a window function was used. This indicates that neither basis is a useful choice overall. In contrast, the results regarding the even basis indicate that using a purely symmetric basis is not a useful choice for learning physical behavior. However, for the odd series with the constant term, we saw reasonable behavior with and without a window function; however, the results were worse than using a modified Fourier Series. This is primarily due to the antisymmetric terms, see Fig. 26 being only for the sole $x$ and $y$ terms, i.e., none of the mixed terms are antisymmetric. This indicates that having antisymmetry alone is insufficient and that having a reasonable ratio of antisymmetric to symmetric terms is essential.

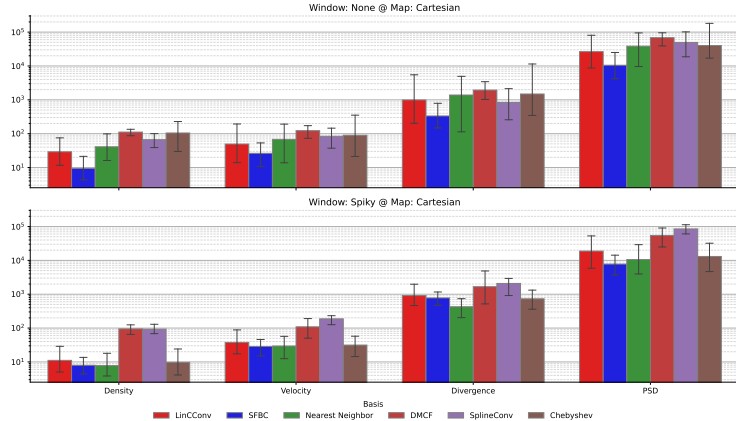

Figure 28: This figure shows the quantitative results for an ablation study of network initialization stability in test case II. All entries are computed by initializing 32 networks for each case at all 4 testing samples from the dataset at 4 different time points and then performing 64 inference steps while computing the (f.l.t.r.) density, velocity, divergence, and PSD errors on a grid resampling of the particle quantities and finally computing the behavior across the inference period. The color indicates the basis term, and error bars indicate the lower 5-th to upper 95-th percentile. The top row indicates the results of not using a window function, and the bottom row indicates the results of using the *Spiky* window function.

Table 8: This figure shows the quantitative results for an ablation study regarding network initialization stability in test case II. All entries are computed by initializing 32 networks for each case at all 4 testing samples from the dataset at 4 different time points and then performing 64 inference steps while computing the density, velocity, divergence, and PSD errors on a grid resampling of the particle quantities and finally computing the mean across the inference period.

| Configuration | Window: None | | | | Window: Spiky | | | |
| | Density | Vel. | Div. | PSD | Density | Vel. | Div. | PSD |
| Basis | | | | | | | | |
|---|---|---|---|---|---|---|---|---|
| LinCConv | 0.114 | 0.193 | 3.86 | 103 | 0.043 | 0.147 | 3.57 | 74 |
| SFBC | **0.037** | **0.103** | **1.30** | **41** | 0.031 | **0.112** | 3.04 | **30** |
| Nearest Neighbor | 0.160 | 0.266 | 5.42 | 150 | **0.030** | 0.115 | **1.67** | 41 |
| DMCF | 0.435 | 0.479 | 7.58 | 268 | 0.370 | 0.426 | 6.52 | 213 |
| SplineConv | 0.260 | 0.327 | 3.27 | 194 | 0.364 | 0.732 | 8.14 | 332 |
| Chebyshev | 0.409 | 0.351 | 5.80 | 157 | 0.038 | 0.123 | 2.86 | 51 |

### C.3.5    ABLATION STUDY: NETWORK INITIALIZATION

Network initialization and training stability are important aspects in machine learning tasks and are especially important for repeatability. To investigate the influence of network initialization and our training setup, we performed two separate ablation studies where we first evaluated 32 different initialization seeds for a set of core methods. Secondly, we investigate if our choice of test frames influences the result, i.e., if the performance of NNTIs is significantly different when applied starting at timestep 0 of a simulation or much later in the simulation and if by choice of this starting point, our results were biased towards specific methods.

**Random Initialization**: Considering random initialization, see Fig. 28 and Table 8, we did not observe any significant differences between this broad set of evaluations and the narrower choice of seeds in the respective ablation studies. While this does indicate that our training setup and initialization process results in a repeatable and fair evaluation, we can still observe that the spread of some methods, especially the *Chebyshev* basis, is much greater than the other methods, indicating that some base functions are more difficult to initialization. We already observed a similar behavior before, e.g., in test case I, where the Chebyshev bases did not perform as well as many other methods

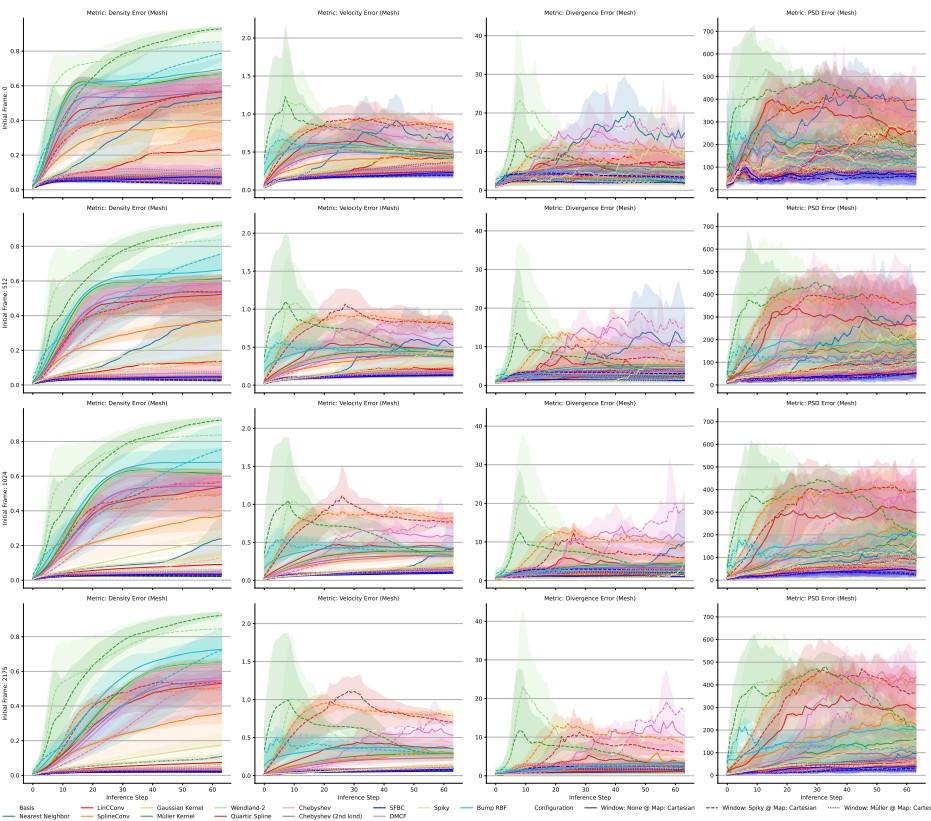

Figure 29: This figure shows the quantitative results for an ablation study of network initialization stability in test case II. All entries are computed by initializing 4 networks for each case at all 4 testing samples from the dataset at a single time point and then performing 64 inference steps while computing the (f.l.t.r.) density, velocity, divergence, and PSD errors on a grid resampling of the particle quantities and finally computing the behavior across the inference period. The color indicates the basis term, and error bars indicate the lower 5-th to upper 95-th percentile. Each row indicates a different initial timestamp in the simulation where the inference was started.

in most cases, partially due to the initialization. In the future, we hope to address the initialization of Chebyshev-based convolutional networks in more depth, but this is beyond our current scope.

**Testing Frame Dependence**: When considering the initial frame choice, see Fig. 29, we did not observe a strong dependence of the initial frame choice to the performance of the network, i.e., if a network performed poorly, it performed poorly regardless of the initial frame. However, a crucial observation to be made here is how the errors increase differently based on initial frame choice, e.g., when considering the density error, the increase of error per inference step is much lower the further into the simulation the inference is started. This behavior is likely due to the underlying SPH data, especially during initialization. As mentioned before, see Appendix B.3, particles in this test case are initialized on a rectangular grid of particles with even spacing, whereas particle distributions become disordered and chaotic during the simulation. Accordingly, few training samples include the regular particle sampling from the beginning of simulations, while most of the training samples are during the more random stages of the simulation. Note that this particle disorder is a general SPH problem often addressed by particle shifting (Rastelli et al., 2022), a process that aims to regularize particle distributions during a simulation to improve numerical accuracy. Regarding the inference behavior, we observed that several methods very quickly diverge away from the ground truth solution but that the error metrics stabilize after some time, where this stabilization happens significantly after the training rollout period. This kind of behavior could indicate that the methods find new stable equilibriums that change depending on the basis function and may be due to the inherent numerical

Table 9: This table shows an ablation study performed in test case III to find the optimal number of basis terms. These numbers were computed using 4 differently initialized networks, training them to overfit to a single training sample and evaluating the point-wise distance based solely on this single training sample. The columns indicate the number of basis terms, with lower values being better.

| n Configuration | 4 | 6 | 8 |
|---|---|---|---|
| LinCCConv @ Spiky | 0.054923 | 0.050374 | 0.049219 |
| SFBC @ None | 0.052832 | 0.050328 | 0.050279 |
| SFBC @ Spiky | **0.045858** | **0.042443** | **0.040794** |

Table 10: This table shows the result of an ablation study performed on test case 3 in an overfitting setup to find an optimal network layout and show the point-wise distance for our proposed Fourier-based architecture with no window function for different network layouts. Note that all values are premultiplied by $10^2$ for legibility, with the lowest value per column being boldened.

| Steps Features | 1 | 2 | 3 | 4 | 5 | 6 | 7 | 8 | 16 | 32 |
|---|---|---|---|---|---|---|---|---|---|---|
| 1 | 6.07 | 5.88 | 5.83 | 5.77 | 5.99 | 6.02 | 6.07 | 6.15 | 6.37 | 6.61 |
| 2 | 5.99 | 5.76 | 5.57 | 5.45 | 5.37 | 5.31 | 5.24 | 5.21 | 5.07 | 4.97 |
| 4 | 5.77 | 5.35 | 5.22 | 5.09 | 5.06 | 4.92 | 4.90 | 4.83 | 4.70 | 4.92 |
| 8 | 5.63 | 5.18 | 4.93 | 4.85 | 4.71 | 4.66 | 4.61 | 4.48 | 4.35 | 4.54 |
| 16 | 5.42 | 4.90 | 4.65 | 4.46 | 4.37 | 4.32 | 4.31 | 4.30 | **4.27** | 4.42 |
| 32 | 5.34 | 4.75 | 4.53 | 4.40 | **4.22** | **4.24** | **4.20** | **4.20** | 4.31 | **4.36** |
| 64 | **5.17** | **4.55** | **4.33** | **4.24** | 4.25 | 4.27 | 4.94 | 4.38 | 4.30 | 5.27 |

properties of each basis function. However, investigating this complex relationship is beyond the scope of our work but an interesting direction for future, more theoretical, research.

## C.4 TEST-CASE III

This section will provide additional details regarding the architecural ablations for the final test case. The main goal of these results is to show how the network can be scaled for better performance and quantify the differences. Finally, we provide results for methods not discussed in the main paper, e.g., *DMCF* without a window function to provide additional context.

### C.4.1 ABLATION STUDY: BASIS TERM COUNT

Analogously, to test case I, we investigated the influence of the number of basis terms on the performance of the networks. To simplify this ablation study, we focused solely on evaluating *LinCCConv* and our Fourier-based approach, with and without a window. Furthermore, we limited the training to a simple overfitting case as we were only concerned with the learning abilities in this problem, not the generalization capabilities. To set up this overfitting test, we chose a random frame from the training set and trained a neural network with 4 message-passing steps and 32 features per layer with an otherwise identical training setup to the other evaluations.

In this evaluation, we saw a minor improvement when increasing the base terms for the *LinCCConv* approach improved by 9% from 4 to 6 base terms and by 2% from 6 to 8 terms. For our Fourier-based approach without a window function, performance only increased by 4% from 4 to 6 base terms but only by 0.1% from 6 to 8 terms, see Table 11. Accordingly, we used 6 base terms for all further evaluations. A crucial observation here is that 6 base terms also was the choice for optimal behavior in the one-dimensional compressible test case, which indicates that this choice is an overall beneficial choice for improving the learning behavior of basis convolutions.

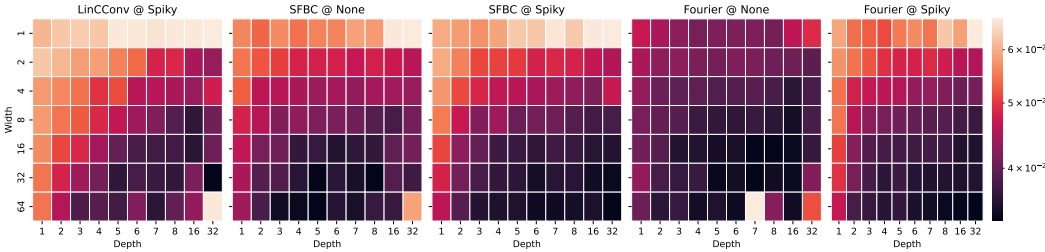

Figure 30: This figure shows the result of an ablation study performed on test case 3 in an overfitting setup to find an optimal network layout. From left to right, the subplots represent using the *LinCConv* approach with a window function, our modified Fourier basis without and with a window function, and using an *proper* Fourier basis without and with a window function. The x-axes represent the number of message-passing steps, and the y-axes the number of features per layer. The color indicates the point-wise distance error from high (white) to low (black), with lower being better.

### C.4.2    ABLATION STUDY: NETWORK LAYOUT

The following ablation study we performed was regarding the network layout, similar to the ablation study performed in test case I; however, we used the same overfitting setup as for the prior ablation study due to the complexity of this training task. For this ablation study, we evaluated three basis functions, i.e., *LinCConv*, a complete Fourier Series, and our modified Fourier series from test case II with a removed first harmonic cosine term, as well as with *no* window and with a window for the Fourier based methods, see Table 10 and Fig. 30. Note that these results were obtained using a single initialization seed, and, accordingly, there might be some notable outliers in the data that are not representative of the overall behavior; however, the general trends still hold.

**LinCConv**: For the *LinCConv* approach, we saw a clear and expected trend, i.e., improving the number of message-passing steps and increasing the features per layer improves performance. Contrary to the results in the one-dimensional case, however, increasing the number of message-passing steps to more than two still significantly improves performance. This difference is primarily due to the inherently increased complexity of the problem, i.e., the underlying physical system has a much larger receptive field and requires significantly more SPH interpolants per simulation step. Consequently, increasing both has a notable benefit for the network's performance.

**Fourier Methods**: Regarding the Fourier-based methods, we observed a similar behavior as for *LinCConv*; however, the overall performance of the Fourier-based networks is notably better at all sizes of network, especially when not using a window function. Furthermore, for this problem, using a complete Fourier Series for small network architectures significantly improves over the modified Fourier Series, indicating that including the symmetric first harmonic term is beneficial. However, as the performance is only marginally different for larger networks, we continued using this Series.

These ablations show that a network with 6 message-passing steps and 32 features per layer, provides reasonable performance for all tested methods without requiring excessive parameter counts. The similarities with test case I indicate that there are general trends in learning behaviors that are worth considering when setting up a neural network for a given learning task.

### C.4.3    BASIS FUNCTION EVALUATION

As the final ablation study, we considered the influence of different basis functions for test case III regarding inference performance. To do this, we trained the respective networks on the entire test case III dataset with an incremental rollout of up to ten timesteps during training, a batch size of four, and a decreasing learning rate. As basis functions, we chose the five most important baselines and proposed configurations, i.e., *LinCConv*, *DMCF* and *Nearest Neighbor* as well as Fourier and Chebyshev Series based networks. For the comparisons, we chose the point-wise distance, i.e., the mean distance of each particle in the prediction relative to the closest ground truth particle, regarding temporal inference behavior, see Fig. 31, and as an average over an inference length of 96 timesteps.

Table 11: Quantiative results for the different basis functions evaluated in test case 3 when learning the physics update. Each value is computed for 4 different network initializations evaluated on all 4 testing samples for 4 different timesteps using an inference length of 96 steps and then computing the mean point-wise distance across the inference period. Lower values are better, and the lowest value per column is boldened.

| window Basis | Müller | None | Spiky |
|---|---|---|---|
| LinCConv | $\mathbf{2.739 \cdot 10^{-3}}$ | $3.047 \cdot 10^{-3}$ | $9.014 \cdot 10^{-3}$ |
| Chebyshev | $2.871 \cdot 10^{-3}$ | $3.508 \cdot 10^{-3}$ | $\mathbf{2.941 \cdot 10^{-3}}$ |
| DMCF | $2.899 \cdot 10^{-3}$ | $3.618$ | $5.534 \cdot 10^{-1}$ |
| Nearest Neighbor | $3.221 \cdot 10^{-3}$ | $3.298 \cdot 10^{-3}$ | $6.197 \cdot 10^{-3}$ |
| SFBC | $2.850 \cdot 10^{-3}$ | $\mathbf{2.876 \cdot 10^{-3}}$ | $3.224 \cdot 10^{-3}$ |

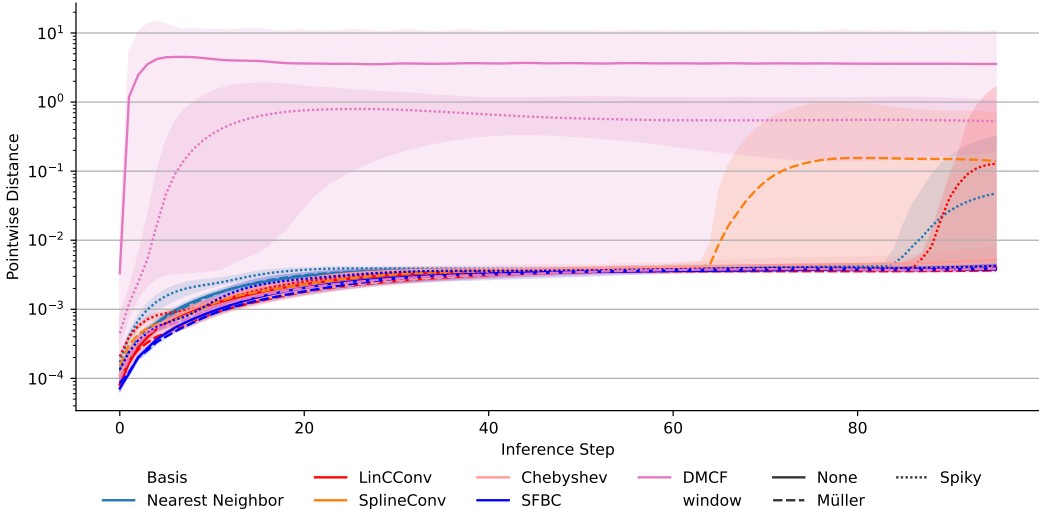

Figure 31: This figure shows the inference stability for test case 3 when learning the physics update. Each line represents 4 different network initializations evaluated on all 4 testing samples for 4 different timesteps using an inference length of 96 steps and computing the point-wise distance to the ground truth data after every inference step. The color indicates the basis function, style indicates the window function and error bars indicate the lower 5-th to upper 95-th percentile.

**LinCConv**: Contrary to the prior test cases, the basic approach by Ummenhofer et al. (2019) performed better than all other approaches, on average; however, during the first 40 inference steps, the performance is slightly worse than our proposed Fourier-based approach. An important observation here is the apparent strong influence of the window function on the performance of the basis function, i.e., changing from the *Spiky* window to the *Müller* window improved performance by a factor of 3.3, mostly due to instabilities during longer inference periods, see Fig. 31.

**DMCF**: The approach by Prantl et al. (2022) performed very close to the other baselines when the *Müller* window was used; however, when using the *Spiky* window as suggested, the performance decreased by two orders of magnitude. Furthermore, not using a window function in this case did not lead to stable prediction behavior for any sees. These observations highlight that while the inductive bias of antisymmetry can be helpful in many cases, it does not guarantee stable behavior and clearly demonstrates how this approach is susceptible to hyperparameter changes.

**Nearest Neighbor**: This basis function performs somewhat similar to *LinCConv* as the choice of window function notably impacts the performance, but overall, performance is pretty comparable.

**Fourier-based**: While our proposed Fourier-based approach outperformed all other bases in most other comparisons, even regarding the overfitting behavior in this test case, there is no significant

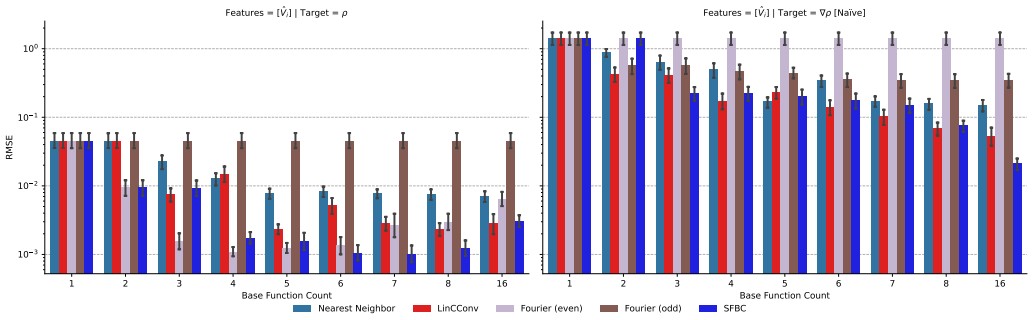

Figure 32: Results of the ablation study for the density (left) and density gradient (right) toy problems in test case IV. Base function color coded.

benefit in this test case. However, our proposed technique still performs better than all other approaches without a window function and, regarding average behavior, is only marginally worse than *LinCCConv*. Furthermore, during the first 40 inference steps, our proposed technique outperforms *LinCCConv*, highlighting that there still is a benefit to choosing our proposed technique.

**Chebyshev**: While the Chebyshev basis struggled in many prior test cases, in this case, it demonstrates performance in line with all other approaches. Furthermore, it shows the best behavior with the *Spiky* window function, which also exhibits better performance than not using a window function for this approach. This clearly highlights that while this method is useful, its performance is still dependent on hyperparameter choices such as window functions.

## C.5    TEST CASE IV

This section provides the results of our evaluations in the three dimensional problems in test case IV regarding SPH kernel and gradient interpolation. Section C.5.1 will discuss results for single-layer networks, setup analogous to Sections C.2, and Section C.5.2 will discuss results regarding three and four-layer networks to evaluate overparametrization, i.e., how the networks learn for a task that requires fewer message passing steps in the ground truth than the network uses.

### C.5.1    SINGLE-LAYER NETWORKS

In this toy problem we want to evaluate the ability of different base functions to learn a simple SPH kernel and gradient interpolation using a single-layer network setup with no bias, analogous to the one-dimensional toy problem, see Fig. 32.

**Kernel Interpolation**: Comparing the results with the results from test case I, see Fig. 2, we observe a significant overlap in behavior. For small numbers of terms, i.e., 1 term per axis, all basis functions perform equally poorly with an improvement as the number of terms increases, except for the odd Fourier basis. While this was expected in the one dimensional test case, the three dimensional basis of an odd Fourier basis contains many terms that are symmetric, however, none of the terms along a cardinal direction are symmetric, which prevents the network from learning any reasonable behavior. Similarly, the even Fourier basis performs better than the full Fourier basis for 3 terms, due to the inclusion of more symmetric terms, and, analogous to the one dimensional problem, the full Fourier basis outperforms the even basis for larger numbers of terms, e.g., for $n = 8$. Furthermore, *LinCCConv* performs better than the nearest neighbor basis for most configurations, whilst both are significantly outperformed by SFBC. However, it is important to note that the overall achieved loss terms here are much higher than in the one-dimensional case, i.e., the lowest loss here is on the order of magnitude of $10^{-3}$, whereas in the one-dimensional case the lowest loss was on the order of magnitude of $10^{-8}$.

**Gradient Interpolation**: Comparing the results with the results from test case I, see Fig. 2, we can also observe a significant overlap in behavior. However, whilst for the one-dimensional case the odd Fourier basis performed better than all other methods, in this case it performs much worse than the full Fourier basis. This, similar to the kernel interpolation task, is due to the three dimensional odd Fourier basis containing a significant number of symmetric product terms and is not a

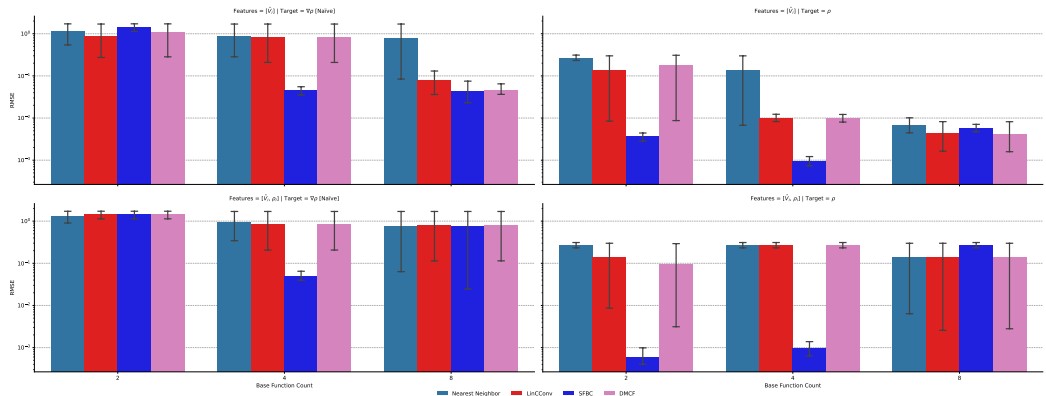

Figure 33: Results of the ablation study for the gradient (left) and density (right) toy problems in test case IV for a network with 2 hidden layers using 32 features each. Base function color coded.

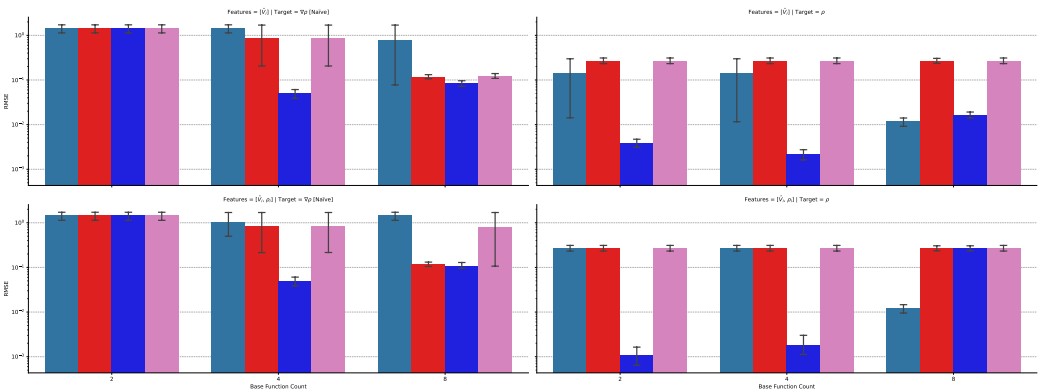

Figure 34: Results of the ablation study for the gradient (left) and density (right) toy problems in test case IV for a network with 3 hidden layers using 32 features each. Base function color coded.

purely antisymmetric basis. Furthermore, while SFBC outperforms LinCConv for some parameters, the difference is much smaller than for the one-dimensional case and the overall loss terms are significantly worse than for the one-dimensional case for a comparable network setup.

Overall, we can observe a significant overlap between the one-dimensional and three-dimensional toy problems, indicating that the insights can be transferred readily across dimensionalities. Furthermore, similar to the two-dimensional evaluations, separable bases cannot be fully antisymmetric, leading to a notably worse performance for purely antisymmetric tasks, whereas symmetric tasks are still straight forward to learn.

### C.5.2 MULTI-LAYER NETWORKS

Thus far we only considered the performance of networks in cases in which the perceptual field, and number of message passing steps, between the ground truth and the graph network were identical, for the toy problems, or cases in which the network is significantly reduced compared to the simulation, for all other problems. However, it is useful to evaluate how the networks perform if they are overparametrized in this regard, i.e., what kind of behavior can be observed if the graph network performs more message-passing steps than necessary. To evaluate this behavior we utilize the same setup as for the single layer toy problem, see Appendix C.5.1, but utilize a network set up with 2, see Fig. 33, and 3, see Fig. 34, additional layers using 32 features each. As network inputs we consider two variants using either just the normalized particle volume or the normalized particle volume and particle density, where in the latter case the particle density input is not required to compute the ground truth and serves to highlight behavior in case an unneeded feature is provided, i.e., how well a network can reject a false signal.

**Two-Layer Network**: For the Kernel interpolation we see a significant drop in performance compared to the single Layer case where for two and four base function terms only *SFBC* was able to learn the correct behavior, and reached a similar loss to the single-layer setup. For eight base function terms all networks perform comparably well, but with a loss that is notably worse than for the single-layer setup. For the gradient interpolation we observed a notably lower error for four base terms for *SFBC* (by approximately an order of magnitude), showing that this problem might be more straight forward to learn with a deeper network with non-linearities, through activation functions, than with a single-layer linear setup. However, this only holds for *SFBC* for four basis terms as no other method was capable of learning any useful behavior. For eight base function terms *LinCConv*, *SFBC* and *DMCF* all out performed the single-layer performance, whereas the *Nearest Neighbor* base function did not learn any meaningful output.

**Three-Layer Network**: The overall behavior observed here is very similar to the two-layer network regarding *SFBC*, i.e., the performance in all cases is as good, or better, than for the single-layer network. However, while for the two-layer setup *LinCConv* and was still able to learn plausible behavior for four and eight base terms, for the three-layer setup *LinCConv* did not learn any meaningful behavior for the kernel interpolation.

**Additional Inputs**: When we add the particle density as an unneeded input to the network can clearly observe that for the kernel interpolation only *SFBC* is able to learn the correct behavior (for two and four basis terms), whereas for eight basis terms only the *Nearet Neighbor* method learns any meaningful output. For the gradient interpolation we furthermore observed that no configuration was able to learn the gradient for a two-layer setup and eight terms, whilst for a three-layer setup and eight terms only *DMCF* changed from learning to not reliably learning (as indicated by the large error bar) the problem task. For four terms only *SFBC* was able to learn any meaningful behavior, consistent with not having the additional input.

Based on these evaluations we can observe that *SFBC* is more resilient to less-than-ideal learning setups that utilizes either suboptimal, i.e., too deep, network architectures or utilize unneeded network inputs. While some methods, e.g., *LinCCOnv* show some resilience to these changes, they are not as resilient. Finally, while the three-layer setup with density as an additional feature is not a, mathematically, optimal setup for learning the density, *SFBC* showed a lower loss for both two and four basis terms by up to an order of magnitude.

## C.6 COMPUTATIONAL PERFORMANCE

So far we only considered the performance of all models regarding their learning efficacy without discussing computational requirements, which are of vital importance for applications in scientific machine learning. To evaluate the computational performance of our method, and baselines, we first constructed a set of dummy data points, i.e., 4096 particles in 1D, $64^2 = 4096$ particles in 2D sampled on the unit square and $16^3 = 4096$ particles in 3D sampled on the unit cube. We then computed a support radius for all particles such that each particle, under periodic boundary conditions, has 32 neighbors and added a random offset to the particle positions. Based on this set of particles we then sampled a random feature per particle using a unit normal distribution, $\mathcal{N}(0, 1)$, and a respective random ground truth per particle using a unit normal distribution as well. For each set of hyperparameters we then constructed a respective neural network and fed the dummy features in, performed a forward pass, compute a loss against the ground truth and performed a backward pass and updated the weights of the network. For each such weight update we computed the computational cost of the forward and backwards pass, as well as the entire weight update process. We also evaluated these updates for a single-layer architecture and an architecture with one and two hidden layers of 32 features each and used 64 measurements per hyperparameter set (totaling $401,600$ timed weight updates in total across all constellations). All measurements were done on a system with an Nvidia RTX A5000 GPU with 24 GiB of VRAM and an Intel Xeon 6242R CPU with 754 GiB of RAM. We then evaluated (a) the computational scaling for different basis functions, numbers of basis terms and dimensionality, (b) the computational cost of coordinate mapping and (c) the computational cost of using a window function.

To investigate the computational scaling we evaluated a set of basis functions in one, two and three dimensions using identity, polar and preserving coordinate mappings and using the *Müller* window function, as well as using no window function, see Fig. 35 and 36 for the results for the forward

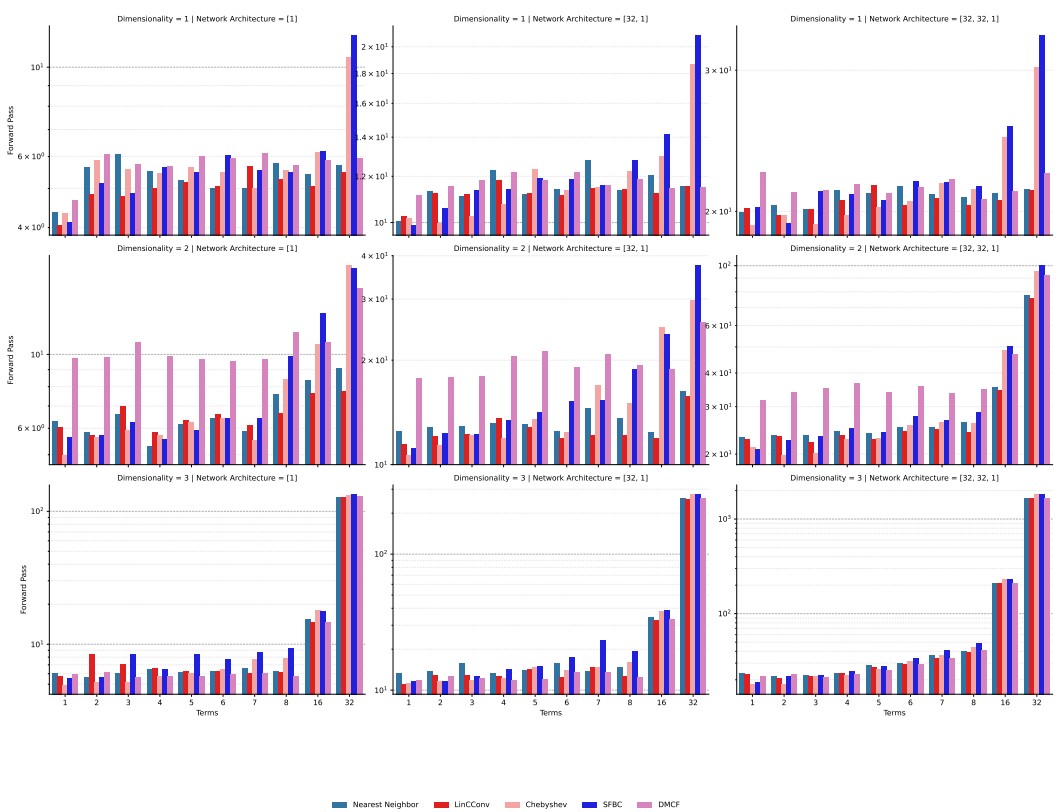

Figure 35: Performance evaluation of the forward pass of various basis functions and approaches using an identity coordinate mapping and no window function in one, two, and three spatial dimensions. Basis function color mapped and row indicating dimensionality, shown for a network using a single layer (left), a single hidden layer with 32 features (middle), and two hidden layers with 32 features each (right). All measurements in milliseconds.

and backward passes. Considering the forward and backward passes, overall, we observed no significant difference in overall behavior but did measure a slightly faster backward pass than forward pass, which is the expected behavior as the backwards pass also involves the same convolutions, but transposed and *in reverse*, but can reuse some results from the forward pass. Consequently, we will not consider the passes separately for all remaining discussions. Regarding the aggregate results, see Fig. 38 left, we see very small difference in performance in all dimensions between the basis, with the only significant outlier being the *DMCF* method in two dimensions. Otherwise, all aggregate values are within $\pm 6\%$, indicating only a small difference in performance between methods. Looking at the influence of basis terms on computational performance, see Fig. 36, we can observe that most configurations for eight and fewer base terms perform very similarly, i.e., within measurement accuracy for most measurements, with the only outlier being the *DMCF* method. For larger networks, i.e., in three dimensions, and for the network with two hidden layers, we see an increase based on the number of basis terms that is approximately cubic, i.e., $\mathcal{O}(n^3)$, which is expected as the number of weights increases with $\mathcal{O}(n^3)$. However, it is important to note that for smaller numbers of base terms, e.g., 8 and fewer the scaling is sublinear due to the GPU being underutilized for smaller term counts. Similarly, we observe that in one dimension only *SFBC* and *Chebyshev* show a significant increase in computational requirements for large numbers of base terms as they are the only bases that require a measurable increase as all other methods are bottle-necked in performance based on other parts of the network.

Regarding computational requirements of the coordinate mapping, see Fig. 37, we see a clear but marginal increase in computational requirements in two dimensions and three dimensions for all numbers of basis terms. Considering the aggregate results, see Fig. 38 middle, the overall differ-

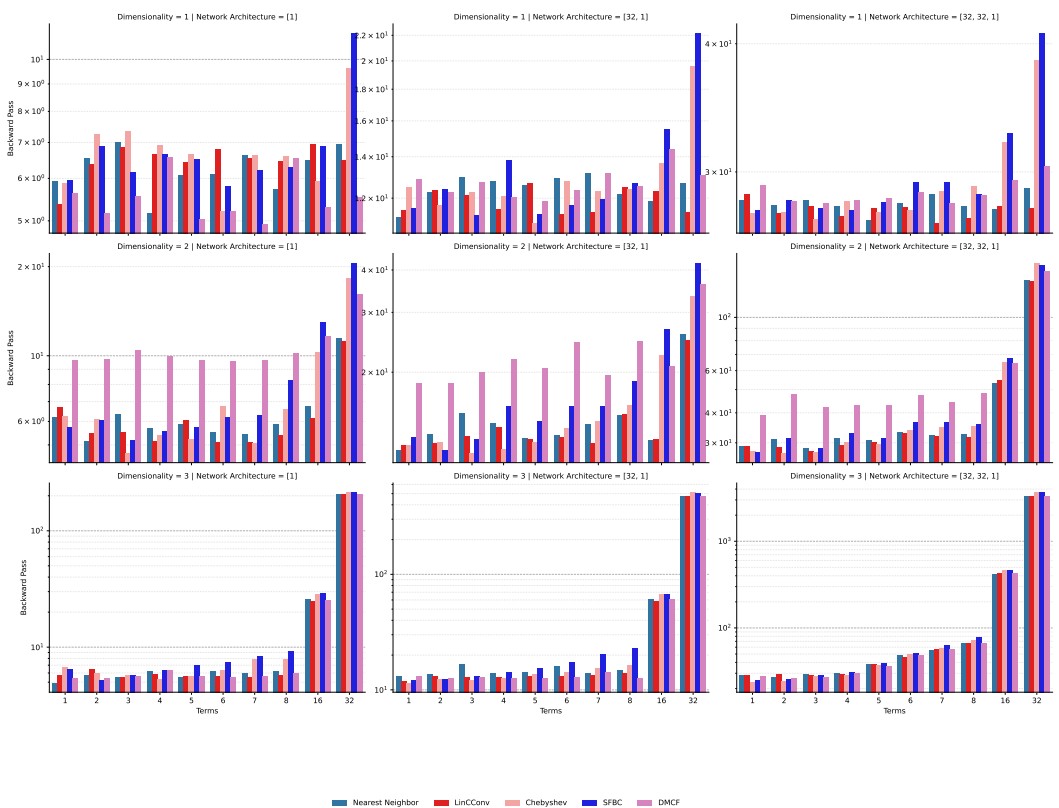

Figure 36: Performance evaluation of the backward pass of various basis functions and approaches using an identity coordinate mapping and no window function in one, two, and three spatial dimensions. Basis function color mapped and row indicating dimensionality, shown for a network using a single layer (left), a single hidden layer with 32 features (middle), and two hidden layers with 32 features each (right). All measurements in milliseconds.

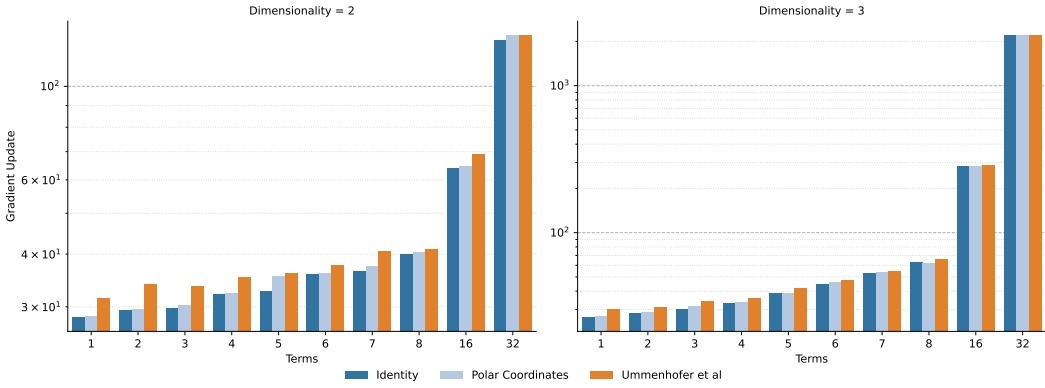

Figure 37: Performance evaluation of a weight update for different coordinate mappings for SFBC and no window function in two and three spatial dimensions. Coordinate Mapping color mapped, results aggregate over all three network architectures, and both window functions tested. All measurements in milliseconds.

ence in computational cost between different coordinate mappings is relatively low at 7% in two dimensions and 0.4% in three dimensions.

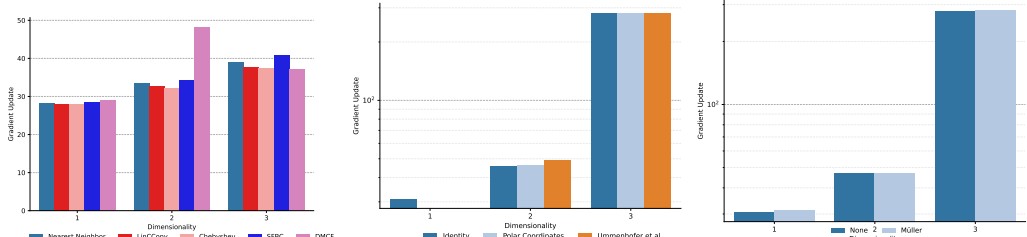

Figure 38: Performance evaluation of a weight update as aggregate results across all relevant hyper-parameters for one, two, and three dimensions (indicated on the x-axis). Color mapping indicates the basis function (left), coordinate mapping (middle), and window function (right). Note that for one dimension, we only evaluated an identity coordinate mapping. All measurements in milliseconds.

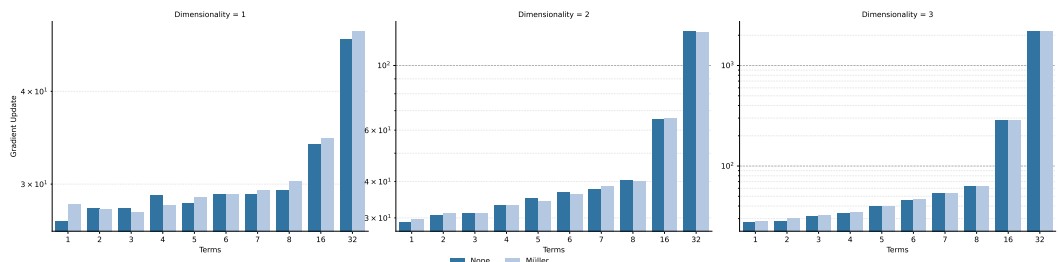

Figure 39: Performance evaluation of a weight update for different window functions for SFBC and an identity coordinate mapping in two and three spatial dimensions. Window function color mapped, values aggregated over all coordinate mappings and architectures. All measurements in milliseconds.

Regarding computational requirements of window functions, see Fig. 39, we observed that for one dimensional networks there is a slight, but consistent, overhead imposed by the window function. However, this increase is relatively small and disappears for two and three dimensions, resulting in a mean difference of performance of at most $\pm 0.7\%$, with two and three-dimensional networks showing a difference of less than $\pm 0.3\%$.

Based on these results, there is no notable overhead of using our *SFBC* approach, compared to *LinCConv*, for any network architecture used in our other evaluations. Furthermore, there is no significant overhead imposed by using either a coordinate mapping or window function.

## D    CONCLUDING THOUGHTS

In our appendix, we provided a broad range of data regarding many different ablation studies to highlight the strengths and weaknesses of several methods. Overall, our proposed Symmetric Fourier Basis Convolution (SFBC) approach performs either ideally or close to ideally across a broad range of problems and scenarios regardless of the hyperparameters chosen, indicating that our method is a versatile and useful method overall. We hope that our evaluations and datasets inspire future research and help with developing novel and exciting solutions to these challenging physical problems.

