# OpenReview forum: "Symmetric Basis Convolutions for Learning Lagrangian Fluid Mechanics"
_ICLR.cc/2024/Conference — ICLR 2024 poster_

### Official Review · Reviewer_Acuo · 2023-11-01

**Soundness:** 3 good
**Presentation:** 3 good
**Contribution:** 3 good
**Rating:** 6
**Confidence:** 3

**Summary:**

- The paper proposes a generalized formulation of continuous convolution approaches using separable basis functions. The coefficients of the basis functions are learned from the training data.
- By using the Fourier series as the basis functions with both even and odd symmetry, the symmetries are built in the in the convolutions which are beneficial for accurately learning Lagrangian flow physics.
- The method is evaluated on multiple cases:
    - The Symmetric Fourier Basis Convolution approach is compared against against other CConv approaches in a toy problem.
    - It’s compared against multiple MLP-based GNNs methods, using a compressible one-dimensional problem
    - It’s evaluated on a two-dimensional closed domain simulation, which shows the proposed method can predict over a long time frame with inference stability. In this 2D case, the paper also explores the influence of window functions, Fourier terms, and coordinate mappings.
    - The free-surfaces is also evaluated, using a fluid blob collision scenario, to investigate how different basis terms perform with partially occupied support domains.

**Strengths:**

- The proposed method of using Fourier basis functions provides an inherently symmetric and smooth continuous convolution approach, which shows good results on several testing cases.
- The paper provides additional details, analysis and ablation studies in the appendix to better show case the method and the comparison with other continuous convolutional approaches.

**Weaknesses:**

- Most real-world applications of fluid simulation are in 3D cases. It would be good if the paper has some 3D examples or discussions regarding how to extend the method to 3D cases.

**Questions:**

- When applying this method to 3D scenarios, is there a concern that the weight matrix (W) might contain too many parameters, and intermediate matrix B (in Eq. 20) might become too large, causing challenges for training?
- When comparing the proposed method with related works, it might be beneficial to include runtime performance metrics as well, which can include information such as the number of iterations required for training, the time taken for each training iteration, and the time take for generating one prediction.

---

> ### Author Response · Authors · 2023-11-16
> **Response to questions and summary of relevant changes**
>
> We appreciate your attention to the computational aspects of our work and have revised the paper to include a discussion of computational performance and a three-dimensional test case. In this response, we will provide additional context to your specific questions and weaknesses mentioned. Furthermore, we have uploaded a new revision of the paper, including the feedback of all reviewers with changes highlighted in purple.
>
> _Regarding three-dimensional systems_: We have added an additional three-dimensional test case in Appendix Sec. C.5, analogous to test case I in Sec. C.1.1 and Appendix Sec. C.1.2, see Section 4.1 of the main paper for a discussion of the results. This additional test case demonstrates how insights from one and two dimensions, e.g., regarding using symmetric and antisymmetric bases, also transfer to three dimensions and how our proposed SFBC approach is also more resilient to non-ideal training setups.
>
> _Regarding the intermediate Matrix B_: The size of the intermediate matrix is certainly a concern and is a potential limitation even in two spatial dimensions for larger particle counts as the size scales with n x u x v x w (for n edges and three spatial dimensions using u,v and w weights per dimension). However, in our backend, which will be published alongside the paper, we perform this computation in a batched format by splitting the set of edges into separate batches of size b. This limits the size of the intermediate matrix to b x u x v x w. However, this would still require storing potentially large matrices for backpropagation, which we avoid by recomputing the components of the outer products using a custom backprop operation. Accordingly, we only need to store the relative positions and input features for backpropagation, which allows us to train our network, even in three dimensions, using very low memory requirements, e.g., less than 4GByte. Note that some implementations of this Einsum operation can perform the operation without storing explicit intermediates, e.g., Nvidia’s Cutlass library. We have expanded the discussion Appendix Sec. A.1 to better reflect this.
>
> _Regarding computational evaluations_:  We have added a runtime performance evaluation in Appendix Sec. C.6, which evaluates a broad set of hyperparameter choices, including basis functions, coordinate mappings, and network architectures. In this evaluation, we found that, on average, the performance difference between different basis functions is very low, i.e., less than 7% between methods (see Figure 38), with a similar scaling of computational costs between methods based on the number of basis terms, i.e., O(n^d) for d dimensions and n basis terms (see Figure 36). We also found a very low overhead for coordinate mappings and window functions. Overall, the performance overhead of using SFBC over LinCConv is small for the networks we evaluated. We briefly discuss these results in Section 4 of the main paper.
>
> Furthermore, we trained our networks with a fixed number of iterations per training and, thus, a comparable overall computational cost. Based on our initialization tests, see Appendix Sec. C.3.5, we consider this training strategy to perform well for our given tasks. Naturally, network initialization substantially impacts training, and we have made best efforts to provide all networks with equally good initializations. However, we see improved methods for initialization of the different convolutional architectures as an important component to be improved in future work.  We have highlighted this as a limitation in the Conclusions.

---

### Official Review · Reviewer_T2aE · 2023-11-02

**Soundness:** 4 excellent
**Presentation:** 3 good
**Contribution:** 3 good
**Rating:** 6
**Confidence:** 5

**Summary:**

The authors propose a general formulation for continuous convolutions using separable basis functions and evaluated on 3 different datasets. They demonstrate that even and odd symmetry are the critical components for success. With the proposed Fourier-based network, the inductive bias, like the window function, is no longer necessary.

**Strengths:**

Quality: the paper is rich in detail, more than 40 pages in total and give sufficient background and relevant math for understanding the problem.

Originality: The author proposed an SFBC approach that works better than other CConv-based methods. The author also shows that with the proposed structure, the window function is not necessary.

Significance: The proposed Fouries-basis network is part of a larger group of symmetric methods that opens up future research.

Clarity: The content is self-contained and easy to follow.

**Weaknesses:**

Some details need to be clarified. See questions. Moreover, there are no movies for the learned dynamics to check for temporal coherence. The unique contribution compared with previous research needs to be elaborated.

**Questions:**

1. The result of the proposed separable basis is demonstrated on the 1D and 2D tests. Since the method uses the outer product, will it cause issues with the scalability to higher dimensions?

2. What is the computational overhead compared with other baseline models?

3. What is the limitation of the current methods?

4. The author cites Fey 2018 works several times. In the cited work, the high dimensional spline basis is also the product of the 1d spline basis, sharing similar properties of the proposed separable basis. Can you elaborate on the difference between the proposed one and Fey's method?

---

> ### Author Response · Authors · 2023-11-16
> **Response to questions and summary of relevant changes**
>
> We appreciate the feedback provided and have revised the paper accordingly. In this response, we will address the questions posited and the weaknesses mentioned in your review. Furthermore, we have uploaded a new revision of the paper, including the feedback of all reviewers with changes highlighted in purple.
>
> _Regarding temporal unrolling behavior_: We apologize for the omission and agree that it is a good idea to provide sequences that show the temporal behavior of the different methods and will include examples alongside the source code and data repositories upon publication. You can find an example of the temporal unrolling behavior for LinCConv with and without a window function and an example of our SFBC approach with and without a window function in the supplementary material. These gifs clearly highlight the difference in temporal behavior, i.e., how our Fourier-based approach yields a significantly lower and more temporally stable behavior than LinCConv
>
> _Regarding the outer product_: Using an outer product can certainly be a limitation in performance as the number of weights of a filter scales with n^d (n basis terms, d dimensions); however, our approach shares this scaling behavior with regular convolutions and with prior work, e.g., Fey et al. [2018] and Ummenhofer et al. [2019]. Based on prior work and our own results in 3D, see Appendix Sec. C.5, this is not yet a limitation in 3D simulations, which are commonly used in fluid mechanics and other simulation problems. It could, however, become a limiting factor for convolutions in higher dimensions.
>
> _Regarding computational overhead_: We have included a runtime evaluation in one, two, and three dimensions for different architectures and hyperparameters in Appendix Sec. C.6 (Figures 36 and 38) . This evaluation shows that for most basis term counts, especially in three dimensions, the computational requirements are very similar between the different basis functions, i.e., they are within 7% of each other. For one dimensional networks, we were able to observe a worse performance of our proposed approach and the Chebyshev basis, as the other basis functions are limited in computational performance by other parts of the network, but only for 16 and 32 base terms. For larger networks, we also observed the expected scaling of O(n^d), with n being the number of basis terms and for d dimensions. We added a reference to these results in Section 4.
>
> _Regarding differentiation from prior work_: Fey's method and similarly LinCConv by Ummenhofer both used a separable basis, with the latter not being aware that their Linear Interpolation is separable. We expanded this concept to a generic formulation of separable bases and showed how having a generic framework to construct continuous convolution layers. This allows for readily choosing the best basis function for a specific task by investigating bases with different inherent properties, e.g., symmetries and smoothness.
>
> Note that with our framework a much broader set of convolutional filters can be explored that are, for example, rotationally symmetric instead of just axisymmetric. Such a filter can be constructed, e.g., by using a spherical coordinate mapping with a single constant filter for the angular component and a different basis, e.g., a Chebyshev base, for the radial components. We have found promising initial results in this direction, but this opens up a much broader hyperparameter space that is beyond our current scope and computational resources. We have added this as an outlook in the Conclusions in Sec. 5. As an example of using such a combination consider the following example of using a combination of a linear basis for the radial component and a cubic spline basis for the angular component: https://i.imgur.com/iaZCnxy.png (uploaded as an anonymous user so double blind reviewing is preserved properly) .
>
> _Regarding Limitations_: We have revised the conclusion section of the paper to highlight the limitations of our proposed method. These limitations include (a) investigating other physical systems and simulations outside of SPH, (b) exploring more complex architectures based on continuous convolutions, and (c) a lack of exploring a large space of hyperparameter choices that seem potentially relevant for future work, e.g., using dissimilar bases within a single network. Our aim was to provide a framework and a thorough investigation of how continuous convolution kernels can be built, and correspondingly, these topics are very interesting directions for follow-up work. We plan to explore some of these avenues and hope that our submission and source code will likewise inspire follow-up work by others.

---

### Official Review · Reviewer_akhW · 2023-11-05

**Soundness:** 4 excellent
**Presentation:** 3 good
**Contribution:** 3 good
**Rating:** 5
**Confidence:** 4

**Summary:**

The study introduces a novel approach employing continuous convolution with symmetric Fourier basis functions, effectively leveraging the problem's inherent biases. The research demonstrates the superiority of using symmetric/anti-symmetric basis functions over the previously suggested explicit weight-tying mechanism. Additionally, the paper offers comprehensive ablation studies on various hyper-parameters, providing in-depth insights into their impact on the outcomes.

**Strengths:**

The research presents a method for acquiring symmetric/anti-symmetric basis functions successfully applied to selected problems. The study conducts a detailed ablation study to compare the proposed technique with related methods and the choice of hyperparameters. The authors also additionally present a novel dataset that can be used for the aforementioned problem.

**Weaknesses:**

1. The contribution of the work requires a clearer elucidation. Although the work introduces a symmetric Fourier Basis, it does not explicitly define the analytical form of the basis functions being utilized or distinguish them from Fourier sine and cosine series. Additionally, essential questions pertaining to the rationale behind the new proposed technique remain unaddressed (please refer to the listed questions).

2. I find it puzzling that the study did not incorporate "WaterRamps" and "Liquid3d" from prior research, considering their complexity. Utilizing these challenging scenarios would have been more suitable for demonstrating the model's effectiveness.

3. Improvements in some cases seem marginal.

**Questions:**

1. Regarding the proposed technique for obtaining symmetric/anti-symmetric basis functions (Eqn 9 and 10), it is essential to clarify whether these equations pertain to symmetry along the x-axis, y-axis, or a combined symmetry along both axes [x,y].
 Additionally, what is the assumed domain of the basis functions $b_x$ and $b_y$? Furthermore, it is crucial to determine whether a given set of orthonormal basis functions retains their orthonormality after applying this technique.

2. How does having more parameters than DMCF help here? Because even if you are learning more coefficients, the basis functions are constrained. So, in terms of expressivity, wouldn't the two methods be the same?

3. Did you directly use the even and odd Fourier Basis for symmetric and anti-symmetric basis functions (after proper modification of the 0 frequency)? If not, what is the explicit equation of the SFB? And if so, Do you think, with proper regularization, filters with Fourier basis will be able to learn proper symmetric and anti-symmetric kernel?

4. In Figure 2, we can see that with a high number of basis functions, Fourier even (or odd) is outperforming DMCF. What might be the reason for it?

---

> ### Author Response · Authors · 2023-11-16
> **Response to questions and summary of relevant changes**
>
> We appreciate your evaluation of our submission, highlighting some important aspects we have addressed in the revised manuscript. In addition to the explicit questions, this response also addresses the choice of the datasets mentioned as a weakness. Furthermore, we have uploaded a new revision of the paper, including the feedback of all reviewers with changes highlighted in purple.
>
> _Regarding the datasets_: We agree that the Liquid3D and Waterramps datasets are useful datasets for machine learning in general, but we chose to generate our own datasets built on a more numerically quantifiable approach. For example, consider the Liquid3D dataset with random shapes colliding in random directions and the error being evaluated using particle distances. While this evaluation is useful from a data science perspective, it is not obvious how the error being evaluated relates to the actual fluid mechanics or the viewer's perception. However, when constructing the datasets, primarily for test cases I, II, and IV, we intentionally chose settings with narrow scopes that have clearly defined and expected behavior. Instead of evaluating particle distances, we can directly evaluate the resulting flow field regarding density, velocity and divergence. We have added this explanation to Appendix B.
>
> _Regarding the basis questions_: When talking about anti-symmetry beyond one-dimensional cases, we refer to antisymmetry with respect to the origin, i.e., the classical physical antisymmetry of f(\bf{x}) = -f(-\bf{x}). The basis functions we use are defined over the domain [-1,1]^d, i.e., the unit cube in 3D, where the relative distance of particles, i.e., the edge distances, are first scaled inversely by the particle support radius and then mapped using an optional coordinate mapping, analogous to the work of Ummenhofer et al. [2019] regarding the orthogonality of the combined basis, we have included proof of this property in Appendix A.1.
>
> _Regarding the DMCF approach_: Prior to submission, we consulted with the original authors of the DMCF approach and decided to implement their approach as described in the paper based on these discussions. The DMCF approach used the Open3D backend for their convolutions and placed the antisymmetric constraint on top of this backend. Accordingly, this required several workarounds, i.e., the explicit mirroring of weights creates a disconnect between computed gradients, which assume the weights to be real, and the actual gradients. For our implementation, we used our own pytorch-geometric-based backend that will be published alongside the paper, and, accordingly, we could implement the antisymmetry requirement directly in the backend. In our implementation, the learned filter is still constrained to be antisymmetric (see Eq, (44)), and all weights contribute to the learned filter, whereas the original DMCF implementation was constrained to be antisymmetric but only half of all weights contributed to the learned filter. We have clarified this in Section 3.
>
> _Regarding the Fourier basis_: We did use the bases directly. For an explicit equation of SFB, see Eq. (53) in Appendix Sec. A.4. We believe that the results for the purely antisymmetric and symmetric terms in 1D (and now 3D) show that the combined basis can already learn purely antisymmetric and symmetric problems with the current learning setup, albeit at a higher error than a purely even or odd basis shown in Figure 2 (which is the expected behavior). Furthermore, the odd Fourier basis only utilizes antisymmetric terms in 1D and is significantly smoother than the DMCF approach, resulting in the lower observed error in Figure 2.
>
> Ummenhofer et al. [2019] - Benjamin Ummenhofer, Lukas Prantl, Nils Thuerey, and Vladlen Koltun. Lagrangian fluid simulation with continuous convolutions. In International Conference on Learning Representations, 2019

---

> > ### Comment · Reviewer_akhW · 2023-12-03
> > **Response to the Authors**
> >
> > I thank the authors for the clarifications. I prefer to keep my score.

---

### Official Review · Reviewer_Ac8w · 2023-11-08

**Soundness:** 3 good
**Presentation:** 3 good
**Contribution:** 3 good
**Rating:** 6
**Confidence:** 3

**Summary:**

This paper introduces Symmetric Fourier Basis Convolutions (SFBC), a new method for learning Lagrangian fluid simulations using graph neural networks. The key idea is to construct convolutional filters using Fourier series as separable basis functions, incorporating both even and odd symmetries. Through extensive experiments on three fluid simulation datasets, the authors demonstrate that SFBC outperforms prior methods like LinCConv and DMCF in terms of accuracy and stability. A major contribution is showing that previously used inductive biases like window functions are no longer needed with the Fourier basis. The paper provides a generalized framework for continuous convolutions, rigorous evaluation of design choices, and highlights the benefits of symmetry and smoothness in the convolutional bases.

**Strengths:**

1. The paper aims to tackle an important and challenging problem in ML for physics - modeling Lagrangian fluid mechanics.

2. Leveraging ideas like symmetry, smoothness, and Fourier bases to inject useful inductive biases into graph networks is logically sound and extends prior work nicely. In addition, it is nice to unify symmetry and antisymmetry under the same framework.

3. The extensive experimental methodology covering diverse design choices and evaluations on three distinct test cases is a major strength. The baselines are strong enough and the evaluation is rigorous. While the theoretical novelty is incremental, the engineering rigor and state-of-the-art results are valuable and significant contributions.

**Weaknesses:**

1. The theoretical novelty is somewhat limited as the core concepts like symmetric bases are adapted from prior work in other domains. Additional theoretical analysis could further strengthen the approach.

2. The evaluations are extensive but restricted to a specific type of fluid simulation problems. It is hard to extract insights that can generalize to a broader scope. Testing generalization on more diverse physical systems could better establish applicability.

[1] Li, Z., Huang, D. Z., Liu, B., & Anandkumar, A. (2022). Fourier neural operator with learned deformations for pdes on general geometries. arXiv preprint arXiv:2207.05209.
[2] Pfaff, T., Fortunato, M., Sanchez-Gonzalez, A., & Battaglia, P. W. (2020). Learning mesh-based simulation with graph networks. arXiv preprint arXiv:2010.03409.

**Questions:**

N/A

---

> ### Author Response · Authors · 2023-11-16
> **Summary of changes based on your feedback**
>
> We appreciate the feedback provided and have revised our paper to address the weaknesses mentioned. While there were no specific questions, we would still like to address and clarify the weaknesses you pointed out. Furthermore, we have uploaded a new revision of the paper, including the feedback of all reviewers with changes highlighted in purple.
>
> _Regarding theoretical foundations_: We have added a short proof that shows that the separable basis construction of our work preserves orthogonality, i.e., if two orthogonal bases b_i and b_j are chosen, then the resulting two-dimensional basis is orthogonal in 2D, and analogously in 3D, see Appendix sec. A.1.
>
> _Regarding the evaluations_: We have added an additional test case that is constructed analogously to test case I but performs the evaluations in 3D instead of 1D, which highlights how some insights, e.g., how symmetric bases help with learning even symmetric functions, remain consistent across dimensions, see Section 4.1 in the main paper and Appendix C.5.
>
> _Regarding more diverse physical systems_: Our paper currently uses a compressible, an explicit weakly compressible, and an implicit incompressible simulation approach with very different requirements for numerical solvers and their respective stability. This provides a broad basis of evaluation for our network, albeit admittedly limited to particle-based hydrodynamics. We believe that exploring other physical systems and areas of application will be an exciting and promising direction for future work.

---

> > ### Comment · Reviewer_Ac8w · 2023-11-23
> >
> > I appreciate the comments from the authors. Since I have already acknowledged the contributions of the paper. I will keep my score as it is.

---

### Author Response · Authors · 2023-11-16
**Rebuttal overview, summary of changes**

We would like to thank all the reviewers for their feedback and questions. We have uploaded a revised version of the paper that addresses the highlighted shortcomings with changes to the text highlighted in purple. We have also added a supplementary video that includes visualizations of the temporal unrolling behavior for some key network configurations for test case II.

Overall, we made the following changes:
- We added an additional test case that utilizes a synthetic three-dimensional setup to evaluate how insights gained from one and two dimensions relate to three dimensions (test case IV see Appendix B.5 for the experiment description and C.5 for the ablation studies on this test case)
- We added a discussion of this new test case in Section 4.1 in the main paper
- For test case IV (see Appendix C.5.1) we evaluate a toy problem analogous to test case I for single-layer networks to evaluate insights into symmetries in higher dimensions. These results indicate a good transfer of insights across dimensions.
- For test case IV (see Appendix C.5.2) we also evaluate how a non-ideal network setup, i.e., a network with a receptive field larger than for the simulation and with unnecessary inputs, impacts the ability of different base functions to learn the correct behavior. These results indicate a much greater resilience of the SFBC approach over, e.g., LinCConv.
- We have added a runtime performance evaluation in one, two, and three dimensions (see Appendix C.6) to evaluate the overhead imposed by SFBC compared to the most relevant baselines and how different hyperparameter choices, e.g., coordinate mapping, impact runtime performance. This evaluation shows that the computational overhead of SFBC over LinCConv is 0.5%, 8.5% and 12%, in one, two and three dimensions, respectively, see Fig. 38 and Section 4.
- We have added a short proof that the tensor product of two orthogonal bases b_x and b_y is orthogonal itself in Appendix A.1
- We have expanded the discussion of our implementation in Appendix A.1 to include how we handle the computation of the Einsum operation at the core of our method to reduce and limit intermediate memory consumption, which enables our method to train three-dimensional networks on GPUs with 4GByte of VRAM.

We have included the results from these insights, especially regarding three-dimensions and runtime performance, at the corresponding places in the main manuscript.

---

### Comment · Area_Chair_4VhG · 2023-11-21
**Reviewers: Please respond to authors or update review**

Dear Reviewers,

The discussion phase will end tomorrow.  Could you kindly respond to the authors rebuttal letting them know if they have addressed your concerns  and update your review as appropriate? Thank you.

-AC

---

### Meta-Review · Area_Chair_4VhG · 2023-12-11

**Metareview:**

**Summary** This paper proposes Symmetric Fourier Basis Convolutions (SFBC), a method for learning Lagrangian fluid mechanics.  SFBC uses continuous convolutions parameterized using separable basis functions.  Although the method provides a general formulation over different bases, Fourier basis functions are found to work best.   The model is evaluated over four different fluid simulations.

**Metareview** This paper addresses an important problem, Lagrangian fluid simulation, and provides a novel and logical method based on reasonable inductive biases.  Although some elements of the method appeared in previous work, the authors provide a general framework and strong implementation.  An important contribution is showing window functions are no longer required for good performance.  The evaluation is rigorous and extensive and shows good performance in accuracy and generalization, with evaluation over several datasets, ablations, and design comparisons.  To improve the paper, the authors should clarify the technical contribution relative to previous work.

**Justification For Why Not Higher Score:**

- clarity of technical contribution and the novelty relative to previous work
- significance of empirical improvements

**Justification For Why Not Lower Score:**

- novel method with effective implementation
- extensive and rigorous evaluation

---

### Decision · Program_Chairs · 2024-01-16

Accept (poster)